# Posterior Contraction Rates for Matérn Gaussian Processes on Riemannian Manifolds

**Paul Rosa**
University of Oxford

**Viacheslav Borovitskiy**
ETH Zürich

**Alexander Terenin**
University of Cambridge
and Cornell University

**Judith Rousseau**
University of Oxford

## Abstract

Gaussian processes are used in many machine learning applications that rely on uncertainty quantification. Recently, computational tools for working with these models in geometric settings, such as when inputs lie on a Riemannian manifold, have been developed. This raises the question: can these intrinsic models be shown theoretically to lead to better performance, compared to simply embedding all relevant quantities into $\mathbb{R}^d$ and using the restriction of an ordinary Euclidean Gaussian process? To study this, we prove optimal contraction rates for intrinsic Matérn Gaussian processes defined on compact Riemannian manifolds. We also prove analogous rates for extrinsic processes using trace and extension theorems between manifold and ambient Sobolev spaces: somewhat surprisingly, the rates obtained turn out to coincide with those of the intrinsic processes, provided that their smoothness parameters are matched appropriately. We illustrate these rates empirically on a number of examples, which, mirroring prior work, show that intrinsic processes can achieve better performance in practice. Therefore, our work shows that finer-grained analyses are needed to distinguish between different levels of data-efficiency of geometric Gaussian processes, particularly in settings which involve small data set sizes and non-asymptotic behavior.

## 1   Introduction

Gaussian processes provide a powerful way to quantify uncertainty about unknown regression functions via the formulation of Bayesian learning. Motivated by applications in the physical and engineering sciences, a number of recent papers [11, 9, 10, 37] have studied how to extend this model class to spaces with geometric structure, in particular Riemannian manifolds including important special cases such as spheres and Grassmannians [4], hyperbolic spaces and spaces of positive definite matrices [5], as well as general manifolds approximated numerically by a mesh [11].

These Riemannian Gaussian process models are starting to be applied for statistical modeling, and decision-making settings such as Bayesian optimization. For example, in a robotics setting, Jaquier et al. [27] has shown that using Gaussian processes with the correct geometric structure allows one to learn quantities such as the orientation of a robotic arm with less data compared to baselines. The same model class has also been used by Coveney et al. [15] to perform Gaussian process regression on a manifold which models the geometry of a human heart for downstream applications in medicine.

Given these promising empirical results, it is important to understand whether these learning algorithms have good theoretical properties, as well as their limitations. Within the Bayesian framework, a natural way to quantify data-efficiency and generalization error is to posit a data-generating mech-

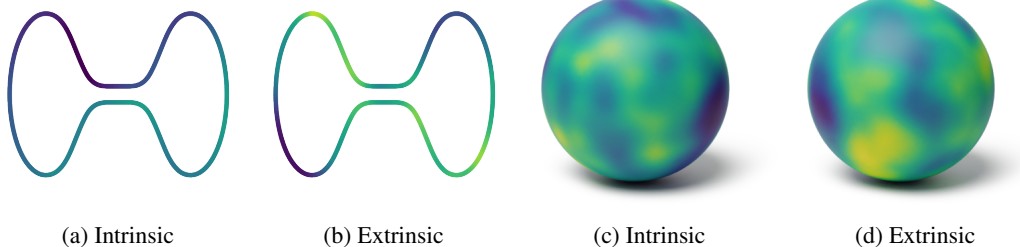

| (a) Intrinsic | (b) Extrinsic | (c) Intrinsic | (d) Extrinsic |

Figure 1: Samples from different Matérn Gaussian processes on different manifolds, namely a one-dimensional dumbbell-shaped manifold and a two-dimensional sphere. Notice that the values across the dumbbell's bottleneck can be very different for the intrinsic process in (a), despite being very close in the ambient Euclidean distance and in contrast to the situation for the extrinsic model in (b). On the other hand, there is little qualitative difference between (c) and (d), since the embedding produces a reasonably-good global approximation to geodesic distances on the sphere.

anism model and study if—and how fast—the posterior distribution concentrates around the true regression function as the number of observations goes to infinity.

Within the Riemannian setting, it is natural to compare *intrinsic* methods, which are formulated directly on the manifold of interest, with *extrinsic* ones, which require one to embed the manifold within a higher-dimensional Euclidean space. For example, the two-dimensional sphere can be embedded into the Euclidean space $\mathbb{R}^3$: intrinsic Gaussian processes model functions on the sphere while extrinsic ones model functions on $\mathbb{R}^3$, which are then restricted to the sphere. Are the former more efficient than the latter? Since embeddings—even isometric ones—at best only preserve distances locally, they can induce spurious dependencies, as points can be close in the ambient space but far away with respect to the intrinsic geodesic distance: this is illustrated in Figure 1. In cases where embeddings significantly alter distances, one can expect intrinsic models to perform better, and it is therefore interesting to quantify such differences.

In other settings, the manifold on which the data lies can be unknown, which makes using intrinsic methods directly no longer possible. There, one would like to understand how well extrinsic methods can be expected to perform. According to the *manifold hypothesis* [18], it is common for perceptual data such as text and images to concentrate on a lower-dimensional submanifold within, for instance, pixel space or sequence space. It is therefore also interesting to investigate how Gaussian process models—which, being kernel-based, are simpler than for instance deep neural networks—perform in such scenarios, at least in the asymptotic regime.

In this work, we develop geometric analogs of the Gaussian process posterior contraction theorems of van der Vaart and van Zanten [56]. More specifically, we derive posterior contraction rates for three main geometric model classes: (1) the intrinsic Riemannian Matérn Gaussian processes, (2) truncated versions of the intrinsic Riemannian Matérn Gaussian processes, which are used in practice to avoid infinite sums, and (3) the extrinsic Euclidean Matérn Gaussian processes under the assumption that the data lies on a compact Riemannian manifold. In all cases, we focus on IID randomly-sampled input points—commonly referred to as *random design* in the literature—and contraction in the sense of the $L^2(p_0)$ distance, defined in Section 2. We focus on *compact* Riemannian manifolds: this allows one to define Matérn Gaussian processes through their Karhunen–Loève expansions, which requires a discrete spectrum for the Laplace-Beltrami operator—see for instance Borovitskiy et al. [11] and Chavel [13], Chapter 1—and is a common setting in statistics [39].

**Contributions.** We show that all three classes of Gaussian processes lead to optimal procedures, in the minimax sense, as long as the smoothness parameter of the kernel is aligned with the regularity of the unknown function. While this result is natural—though non trivial—in the case of intrinsic Matérn processes, it is rather remarkable that it also holds for extrinsic ones. This means that in order to understand their differences better, finite-sample considerations are necessary. We therefore present experiments that compute the worst case errors numerically. These experiments highlight that intrinsic models are capable of achieving better performance in the small-data regime. We conclude with a discussion of why these results—which might at first seem counterintuitive—are very natural

when viewed from an appropriate mathematical perspective: they suggest that optimality is perhaps best seen as a basic property or an important guarantee that any sensible model should satisfy.

## 2 Background

*Gaussian process regression* is a Bayesian approach to regression where the modeling assumptions are $y_i = f(x_i) + \varepsilon_i$, with $\varepsilon_i \sim \mathrm{N}(0, \sigma_\varepsilon^2)$, $x_i \in X$, and $f$ is assigned a Gaussian process prior. A *Gaussian process* is a random function $f : X \to \mathbb{R}$ for which all finite-dimensional marginal distributions are multivariate Gaussian. The distribution of such a process is uniquely determined by its *mean function* $m(\cdot) = \mathbb{E}(f(\cdot))$ and *covariance kernel* $k(\cdot, \cdot') = \mathrm{Cov}(f(\cdot), f(\cdot'))$, hence we write $f \sim \mathrm{GP}(m, k)$.

For Gaussian process regression, the posterior distribution given the data is also a Gaussian process with probability kernel $\Pi(\cdot \mid \boldsymbol{x}, \boldsymbol{y}) = \mathrm{GP}(m_{\Pi(\cdot \mid \boldsymbol{x}, \boldsymbol{y})}, k_{\Pi(\cdot \mid \boldsymbol{x}, \boldsymbol{y})})$, see Rasmussen and Williams [41],

$$m_{\Pi(\cdot \mid \boldsymbol{x}, \boldsymbol{y})}(\cdot) = \mathbf{K}_{(\cdot)\boldsymbol{x}}(\mathbf{K}_{\boldsymbol{x}\boldsymbol{x}} + \sigma_\varepsilon^2 \mathbf{I})^{-1}\boldsymbol{y}, \tag{1}$$

$$k_{\Pi(\cdot \mid \boldsymbol{x}, \boldsymbol{y})}(\cdot, \cdot') = \mathbf{K}_{(\cdot, \cdot')} - \mathbf{K}_{(\cdot)\boldsymbol{x}}(\mathbf{K}_{\boldsymbol{x}\boldsymbol{x}} + \sigma_\varepsilon^2 \mathbf{I})^{-1}\mathbf{K}_{\boldsymbol{x}(\cdot')}. \tag{2}$$

These quantities describe how incorporating data updates the information contained within the Gaussian process. We will be interested studying the case where $X$ is a Riemannian manifold, but first review the existing theory on the asymptotic behaviour of the posterior when $X = [0, 1]^d$.

### 2.1 Posterior Contraction Rates

Posterior contraction results describe how the posterior distribution concentrates around the true data generating process, as the number of observations increases, so that it eventually uncovers the true data-generating mechanism. The area of *posterior asymptotics* is concerned with understanding conditions under which this does or does not occur, with questions of *posterior contraction rates*—how fast such convergence occurs—being of key interest. At present, there is a well-developed literature on posterior contraction rates, see Ghosal and van der Vaart [20] for a review.

In the context of Gaussian process regression with *random design*, which is the focus of this paper, the true data generating process is assumed to be of the form

$$y_i \mid x_i \sim \mathrm{N}(f_0(x_i), \sigma_\varepsilon^2) \qquad\qquad x_i \sim p_0 \tag{3}$$

where $f_0 \in \mathcal{F} \subseteq \mathbb{R}^X$, a class of real-valued functions, and $\mathrm{N}(\mu, \sigma^2)$ denotes the Gaussian with moments $\mu, \sigma^2$. Note that, in this particular variant, these equations exactly mirror those of the Gaussian process model's likelihood, including the use of the same noise variance $\sigma_\varepsilon^2$ in both cases: in this paper, we focus on the particular case where $\sigma_\varepsilon$ is known in advance. This setting is restrictive, one can extend to an unknown $\sigma_\varepsilon > 0$ using techniques that are not specific to our geometric setting: for instance, the approach of [55] allows to handle an unknown $\sigma_\varepsilon$ if one assumes an upper and lower bound on it and keep the same contraction rates. In practice, more general priors, including ones that do not assume an upper or lower bound on $\sigma_\varepsilon$, can be used, such as a conjugate one like in Banerjee [6]—these can also be analyzed to obtain contraction rates, albeit with additional considerations. The generalization error for prediction in such models is strongly related to the *weighted $L^2$ loss* given by

$$\|f - f_0\|_{L^2(p_0)} = \left(\int_X |f(x) - f_0(x)|^2 \, \mathrm{d}p_0(x)\right)^{1/2} \tag{4}$$

which is arguably the natural way of measuring discrepancy between $f$ and $f_0$, given the fact that the covariates $x_i$ are sampled from $p_0$. The posterior contraction rate is then defined as

$$\mathbb{E}_{\boldsymbol{x}, \boldsymbol{y}} \, \mathbb{E}_{f \sim \Pi(\cdot \mid \boldsymbol{x}, \boldsymbol{y})} \|f - f_0\|_{L^2(p_0)}^2 \tag{5}$$

where $\mathbb{E}_{f \sim \Pi(\cdot \mid \boldsymbol{x}, \boldsymbol{y})}(\cdot)$ denotes expectation under the posterior distribution while $\mathbb{E}_{\boldsymbol{x}, \boldsymbol{y}}(\cdot)$ denotes expectation under the true data generating process.[1] In the case of covariates distributed on $[0, 1]^d$, posterior contraction rates have been derived under Matérn Gaussian process priors [47] in van der Vaart and van Zanten [56], who showed the following result.

---

[1]Note that other notions of posterior contraction can be found in the literature, see Ghosal and van der Vaart [20] and Rousseau [42] for examples that are slightly weaker than the definition we work with.

**Result 1** (Theorem 2 of van der Vaart and van Zanten [56]). *In the Bayesian regression model, let $f$ be a mean-zero Matérn Gaussian process prior on $\mathbb{R}^d$ with amplitude $\sigma_f^2$, length scale $\kappa$, and smoothness $\nu > d/2$. Assume that the true data generating process is given by* (3), *where $p_0$ has a Lebesgue density on $X = [0,1]^d$ which is bounded from below and above by $0 < c_{p_0} < C_{p_0} < \infty$, respectively. Let $f_0 \in H^\beta \cap \mathcal{CH}^\beta$ with $\beta > d/2$, where $H^\beta$ and $\mathcal{CH}^\beta$ the Sobolev and Hölder spaces, respectively. Then there exists a constant $C > 0$, which does not depend on $n$ but does depend on $d$, $\sigma_f^2$, $\nu$, $\kappa$, $\beta$, $p_0$, $\sigma_\varepsilon^2$, $\|f_0\|_{H^\beta(\mathcal{M})}$, and $\|f_0\|_{\mathcal{CH}^\beta(\mathcal{M})}$, such that*

$$\mathbb{E}_{\boldsymbol{x},\boldsymbol{y}} \, \mathbb{E}_{f \sim \Pi(\cdot|\boldsymbol{x},\boldsymbol{y})} \|f - f_0\|_{L^2(p_0)}^2 \leq Cn^{-\frac{2\min(\beta,\nu)}{2\nu+d}} \tag{6}$$

*and, moreover, the posterior mean satisfies*

$$\mathbb{E}_{\boldsymbol{x},\boldsymbol{y}} \, \|m_{\Pi(\cdot|\boldsymbol{x},\boldsymbol{y})} - f_0\|_{L^2(p_0)}^2 \leq Cn^{-\frac{2\min(\beta,\nu)}{2\nu+d}}. \tag{7}$$

Note that $m_{\Pi(\cdot|\boldsymbol{x},\boldsymbol{y})}$ is the Bayes estimator [52] of $f$ associated to the weighted $L^2$ loss and that the second inequality above is a direct consequence of the first. Therefore the posterior contraction rate implies the same convergence rate for $m_{\Pi(\cdot|\boldsymbol{x},\boldsymbol{y})}$. The best rate is attained when $\beta = \nu$: that is, when true smoothness and prior smoothness match—which is known to be minimax optimal in the problem of estimating $f_0$: see Tsybakov [52]. In this paper, we extend this result to the manifold setting.

## 2.2 Related Work and Current State of Affairs

The formalization of posterior contraction rates of Bayesian procedures dates back to the work of Schwartz [46] and Le Cam [29], but has been extensively developed since the seminal paper of Ghosal et al. [19] for various sampling and prior models, see for instance [20, 42] for reviews. This includes, in particular, work on Gaussian process priors [54, 56, 57, 43, 49]. Most of the results in the literature, however, assume Euclidean data: as a consequence, contraction properties of Bayesian models under manifold assumptions are still poorly understood, with exception of some recent developments in both density estimation [7, 8, 60] and regression [63, 60].

The results closest to ours are those of Yang and Dunson [63] and Castillo et al. [12]. In the former, the authors use an extrinsic length-scale-mixture of squared exponential Gaussian processes to achieve optimal contraction rates with respect to the weighted $L^2$ norm, using a completely different proof technique compared to us, and their results are restricted to $f_0$ having Hölder smoothness of order less than or equal to two. On the other hand Castillo et al. [12] consider, as an intrinsic process on the manifold, a hierarchical Gaussian process based on its heat kernel and provide posterior contraction rates. For the Matérn class, Li et al. [30] presents results which characterize the asymptotic behavior of kernel hyperparameters: our work complements these results by studying contraction of the Gaussian process itself toward the unknown ground-truth function. One can also study analogous discrete problems: Dunson et al. [17] and Sanz-Alonso and Yang [45] present posterior contraction rates for a specific graph Gaussian process model in a semi-supervised setting. In the next section, we present our results on Matérn processes, defined either by restriction of an ambient process or by an intrinsic construction, and discuss their implications.

## 3 Posterior Contraction Rates on Compact Riemannian Manifolds

We now study posterior contraction rates for Matérn Gaussian processes on manifolds, which are arguably the most-widely-used Gaussian process priors in both the Euclidean and Riemannian settings. We begin by more precisely describing our geometric setting before stating our key results and discussing their implications. From now on, we write $X = \mathcal{M}$, to emphasize that the covariate space is a manifold.

**Assumption 2.** *Assume that $\mathcal{M} \subset \mathbb{R}^D$ is a smooth, compact submanifold (without boundary) of dimension $d < D$ equipped with the standard Riemannian volume measure $\mu$.*

We denote $|\mathcal{M}| = \int_{\mathcal{M}} d\mu(x)$ for volume of $\mathcal{M}$. With this geometric setting defined, we will need to describe regularity assumptions in terms of functional spaces on the manifold $\mathcal{M}$. We work with Hölder spaces $\mathcal{CH}^\gamma(\mathcal{M})$, defined using charts via the usual Euclidean Hölder spaces, the Sobolev spaces $H^s(\mathcal{M})$, and Besov spaces $B^s_{\infty,\infty}(\mathcal{M})$ which are one of the ways of generalizing

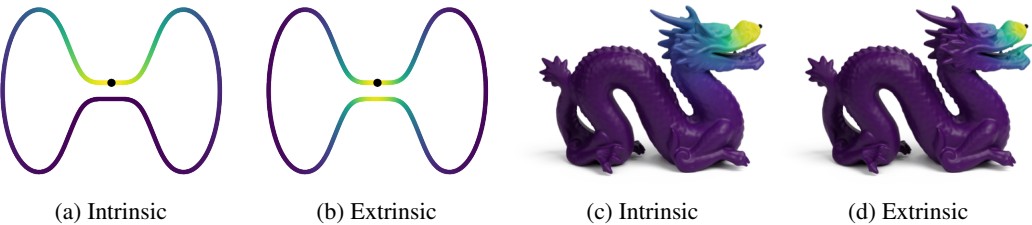

| (a) Intrinsic | (b) Extrinsic | (c) Intrinsic | (d) Extrinsic |

Figure 2: Different Matérn kernels $k(\bullet, x)$ on different manifolds.

the Euclidean Hölder spaces of smooth functions to manifolds. We follow Coulhon et al. [14] and Castillo et al. [12], and define these spaces using the Laplace-Beltrami operator on $\mathcal{M}$ in Appendix A.

Recall that the data-generating process is given by (3), with $f_0$ as the true regression function and $p_0$ as the distribution of the covariates.

**Assumption 3.** *Assume that $p_0$ is absolutely continuous with respect to $\mu$, and that its density, denoted by $p_0$, satisfies $c \leq p_0 \leq C$ for $0 < c, C < \infty$. Assume the regression function $f_0$ satisfies $f_0 \in H^\beta(\mathcal{M}) \cap B^\beta_{\infty,\infty}(\mathcal{M})$ for some $\beta > d/2$, and that $\sigma^2_\varepsilon > 0$ is fixed and known.*

This setting can be extended to handle unknown variance $\sigma_\varepsilon$ by putting a prior on $\sigma_\varepsilon$, following the strategy of Salomond [44] and Naulet and Barat [36]. Since we are focused primarily on the impact of the manifold, we do not pursue this here. With the setting fully defined, we proceed to develop posterior contraction results for different types of Matérn Gaussian process priors: intrinsic, intrinsic truncated and extrinsic.

## 3.1 Intrinsic Matérn Gaussian Processes

We now introduce the first geometric Gaussian process prior under study—the Riemannian Matérn kernel of Whittle [62], Lindgren et al. [33], and Borovitskiy et al. [11]. This process was originally defined using stochastic partial differential equations: here, we present it by its Karhunen–Loève expansion, to facilitate comparisons with its truncated analogs presented in Section 3.2.

**Definition 4** (Intrinsic Matérn prior). *Let $\nu > 0$, and let $(\lambda_j, f_j)_{j \geq 0}$ be the eigenvalues and orthonormal eigenfunctions of the Laplace–Beltrami operator on $\mathcal{M}$, in increasing order. Define the intrinsic* RIEMANNIAN MATÉRN GAUSSIAN PROCESS *through its Karhunen–Loève expansion to be*

$$f(\cdot) = \frac{\sigma^2_f}{C_{\nu,\kappa}} \sum_{j=1}^\infty \left( \frac{2\nu}{\kappa^2} + \lambda_j \right)^{-\frac{\nu+d/2}{2}} \xi_j f_j(\cdot) \qquad \xi_j \sim \mathrm{N}(0,1) \qquad (8)$$

*where $\nu, \kappa, \sigma^2_f$ are positive parameters and $C_{\nu,\kappa}$ is the normalization constant, chosen such that $\frac{1}{|\mathcal{M}|} \int_M \mathrm{Var}(f(x)) \, \mathrm{d}\mu(x) = \sigma^2_f$, where* $\mathrm{Var}$ *denotes the variance.*

The covariance kernels of these processes are visualized in Figure 2. With this prior, and the setting defined in Section 3, we are ready to present our first result: this model attains the desired optimal posterior contraction rate as soon as the regularity of the ground-truth function matches the regularity of the Gaussian process, as described by the parameter $\nu$.

**Theorem 5.** *Let $f$ be a Riemannian Matérn Gaussian process prior of Definition 4 with smoothness parameter $\nu > d/2$ and let $f_0$ satisfy Assumption 3. Then there is a $C > 0$ such that*

$$\mathbb{E}_{\boldsymbol{x},\boldsymbol{y}} \, \mathbb{E}_{f \sim \Pi(\cdot|\boldsymbol{x},\boldsymbol{y})} \|f - f_0\|^2_{L^2(p_0)} \leq C n^{-\frac{2\min(\beta,\nu)}{2\nu+d}}. \qquad (9)$$

All proofs are given in Appendix B. Our proof follows the general approach of van der Vaart and van Zanten [56], by first proving a contraction rate with respect to the distance $n^{-1/2}\|f(\boldsymbol{x}) - f_0(\boldsymbol{x})\|_{\mathbb{R}^n}$ at input locations $\boldsymbol{x}$, and then extending the result to the true $L^2$-distance by applying a suitable concentration inequality. The first part is obtained by studying the *concentration function*, which is known to be the key quantity to control in order to derive contraction rates of Gaussian process priors—see Ghosal and van der Vaart [20] and van der Vaart and van Zanten [57] for an overview.

Given our regularity assumptions on $f_0$, the most difficult part lies in controlling the small-ball probabilities $\Pi\big[\|f\|_{\mathcal{C}(\mathcal{M})} < \varepsilon\big]$: we handle this by using results relating this quantity with the entropy of an RKHS unit ball with respect to the uniform norm. Since our process' RKHS is related to the Sobolev space $H^{\nu+d/2}(\mathcal{M})$ which admits a description in terms of charts, we apply results on the entropy of Sobolev balls in the Euclidean space to conclude the first part. Finally, to extend the rate to the true $L^2(p_0)$ norm, following van der Vaart and van Zanten [56], we prove a Hölder-type property for manifold Matérn processes, and apply Bernstein's inequality. Together, this gives the claim.

This result is good news for the intrinsic Matérn model: it tells us that asymptotically it incorporates the data as efficiently as possible at least in terms of posterior contraction rates, given that its regularity matches the regularity of $f_0$. An inspection of the proof shows that the constant $C > 0$ can be seen to depend on $d, \sigma_f^2, \nu, \kappa, \beta, p_0, \sigma_\varepsilon^2, \|f_0\|_{H^\beta(\mathcal{M})}, \|f_0\|_{B_{\infty\infty}^\beta(\mathcal{M})}$, and $\|f_0\|_{\mathcal{CH}^\beta(\mathcal{M})}$. Theorem 5 extends to the case where the norm is raised to any power $q > 1$ rather than the second power, with the right-hand side raised to the same power: see Appendix B for details. We now consider variants of this prior that can be implemented in practice.

## 3.2 Truncated Matérn Gaussian Processes

The Riemannian Matérn prior's covariance kernel cannot in general be computed exactly, since Definition 4 involves an infinite sum. Arguably the simplest way to implement these processes numerically is to truncate the respective infinite series in the Karhunen–Loève expansion by taking the first $J$ terms, which is also optimal in an $L^2(\mathcal{M})$-sense.

Note that the truncated prior is a randomly-weighted finite sum of Laplace–Beltrami eigenfunctions, which have different smoothness properties compared to the original prior: the truncated prior takes its values in $\mathcal{C}^\infty(\mathcal{M})$ since the eigenfunctions of $\mathcal{M}$ are smooth—see for instance De Vito et al. [16]. Nevertheless, if the truncation level is allowed to grow as the sample size increases, then the regularity of the process degenerates and one gets a function with essentially-finite regularity in the limit.

Truncated random basis expansions have been studied extensively in the Bayesian literature in the Euclidean setting—see for instance Arbel et al. [2] and Yoo et al. [64] or Ghosal and van der Vaart [20], Chapter 11 for examples with priors based on wavelet expansions. It is known that truncating the expansion at a high enough level usually allows one to retain optimality. Instead of truncating deterministically, it is also possible to put a prior on the truncation level and resort to MCMC computations which would then select the optimal number of basis functions adaptively, at the expense of a more computationally intensive method—this is done, for instance, in van der Meulen et al. [53] in the context of drift estimation for diffusion processes. Random truncation has been proven to lead in many contexts to adaptive posterior contraction rates, meaning that although the prior does not depend on the smoothness $\beta$ of $f_0$, the posterior contraction rate—up to possible $\ln n$ terms—is of order $n^{-\beta/(2\beta+d)}$: see for instance Arbel et al. [2] and Rousseau and Szabo [43].

By analogy of the Euclidean case with its random Fourier feature approximations [40], we can call the truncated version of Definition 4 the *manifold Fourier feature* model, for which we now present our result.

**Theorem 6.** *Let $f$ be a Riemannian Matérn Gaussian process prior on $\mathcal{M}$ with smoothness parameter $\nu > d/2$, modified to truncate the infinite sum to at least $J_n \geq cn^{\frac{d(\min(1,\nu/\beta))}{2\nu+d}}$ terms, and let $f_0$ satisfy Assumption 3. Then there is a $C > 0$ such that*

$$\mathbb{E}_{\boldsymbol{x},\boldsymbol{y}} \, \mathbb{E}_{f\sim\Pi(\cdot|\boldsymbol{x},\boldsymbol{y})} \|f - f_0\|_{L^2(p_0)}^2 \leq Cn^{-\frac{2\min(\beta,\nu)}{2\nu+d}}. \tag{10}$$

The proof is essentially-the-same as the non-truncated Matérn, but involves tracking dependence of the inequalities on the truncation level $J_n$, which implicitly defines a sequence of priors rather than a single fixed prior.

This result is excellent news for the intrinsic models: it means that they inherit the optimality properties of the limiting one, even in the absence of the infinite sum—in spite of the fact that the corresponding finite-truncation prior places its probability on $\mathcal{C}^\infty(\mathcal{M})$. Again, the constant $C > 0$ can be seen to depend on $d, \sigma_f^2, \nu, \kappa, \beta, p_0, \sigma_\varepsilon^2, \|f_0\|_{H^\beta(\mathcal{M})}, \|f_0\|_{B_{\infty\infty}^\beta(\mathcal{M})}$, and $\|f_0\|_{\mathcal{CH}^\beta(\mathcal{M})}$. This concludes our results for the intrinsic Riemannian Matérn priors. We now study what happens if, instead of working with a geometrically-formulated model, we simply embed everything into $\mathbb{R}^d$ and formulate our models there.

### 3.3 Extrinsic Matérn Gaussian Processes

The results of the preceding sections provide good reason to be excited about the intrinsic Riemannian Matérn prior: the rates it obtains match the usual minimax rates seen for the Euclidean Matérn prior and Euclidean data, provided that we match the smoothness $\nu$ with the regularity of $f_0$. Another possibility is to consider an extrinsic Gaussian process, that is, a Gaussian process defined over an ambient space. This has been considered by Yang and Dunson [63] for instance for the square-exponential process, in an adaptive setting where one does not assume that the regularity $\beta$ of $f_0$ is explicitly known, but where $\beta \leq 2$. In this section we prove a non-adaptive analog of this result for the Matérn process.

**Definition 7** (Extrinsic Matérn prior). *Assume that the manifold $\mathcal{M}$ is isometrically embedded in the Euclidean space $\mathbb{R}^D$, such that we can regard $\mathcal{M}$ as a subset of $\mathbb{R}^D$. Consider the Gaussian process with zero mean and kernel given by restricting onto $\mathcal{M}$ the standard Euclidean Matérn kernel*

$$k_{\nu,\kappa,\sigma_f^2}(x,x') = \sigma_f^2 \frac{2^{1-\nu}}{\Gamma(\nu)} \left( \sqrt{2\nu} \frac{\|x-x'\|_{\mathbb{R}^D}}{\kappa} \right)^\nu K_\nu \left( \sqrt{2\nu} \frac{\|x-x'\|_{\mathbb{R}^D}}{\kappa} \right) \tag{11}$$

*where $\sigma_f, \kappa, \nu > 0$ and $K_\nu$ is the modified Bessel function of the second kind [22].*

Since the extrinsic Matérn process is defined in a completely agnostic way with respect to the manifold geometry, we would expect it to be less performant when $\mathcal{M}$ is known. However, it turns out that the extrinsic Matérn process converges at the same rate as the intrinsic one, as given in the following claim.

**Theorem 8.** *Let $f$ be a mean-zero extrinsic Matérn Gaussian process prior with smoothness parameter $\nu > d/2$ on $\mathcal{M}$, and let $f_0$ satisfy Assumption 3. Then for some $C > 0$ we have*

$$\mathbb{E}_{\boldsymbol{x},\boldsymbol{y}}\, \mathbb{E}_{f\sim\Pi(\cdot|\boldsymbol{x},\boldsymbol{y})}\|f - f_0\|^2_{L^2(p_0)} \leq Cn^{-\frac{2\min(\beta,\nu)}{2\nu+d}}. \tag{12}$$

Theorem 8 is a surprising result because the optimal rates in this setting only require the knowledge of the regularity $\beta$, but not the knowledge of the manifold or the intrinsic dimension. More precisely, the prior is not designed to be an adaptive prior, since it is a fixed Gaussian process, but it surprisingly adapts to the dimension of the manifold, and thus to the manifold.

The proof is also based on control of concentration functions. The main difference is that, although the ambient process has a well known RKHS—the Sobolev space $H^{s+D/2}(\mathbb{R}^D)$—the restricted process has a non-explicit RKHS, which necessitates further analysis. We tackle this issue by using results from Große and Schneider [24] relating manifold and ambient Sobolev spaces by linear bounded trace and extension operators, and from Yang and Dunson [63] describing a general link between the RKHS of an ambient process and its restriction. This allows us to show that the restricted process has an RKHS that is actually norm-equivalent to the Sobolev space $H^{\nu+d/2}(\mathcal{M})$, which allows us to conclude the result in the same manner as in the intrinsic case.

As consequence, our argument applies *mutatis mutandis* in any setting where suitable trace and extension theorems apply, with the Riemannian Matérn case corresponding to the usual Sobolev results. In particular, our arguments therefore apply directly to other processes possessing similar RKHSs, such as for instance various kernels defined on the sphere—see e.g. Wendland [61], Chapter 17 and Hubbert et al. [26]. The constant $C > 0$ can be seen to depend on $d, D, \sigma_f^2, \nu, \kappa, \beta, p_0, \sigma_\varepsilon^2$, $\|f_0\|_{H^\beta(\mathcal{M})}, \|f_0\|_{B^\beta_{\infty\infty}(\mathcal{M})}, \|f_0\|_{\mathcal{CH}^\beta(\mathcal{M})}$—notice that here $C$ depends implicitly on $D$ because of the presence of trace and extension operator continuity constants. We now proceed to understand the significance of the overall results.

### 3.4 Summary of Results

As a consequence of our previous results, fixing a single common data generating distribution determined by $p_0, f_0$, under suitable conditions the intrinsic Matérn process, its truncated version, and the extrinsic Matérn process all possess the *same* posterior contraction rate with respect to the $L^2(p_0)$-norm, which depends on $d$, $\nu$, and $\beta$, and is optimal if the regularities of $f_0$ and the prior match. These results imply the following immediate corollary, which follows by convexity of $\|\cdot\|^2_{L^2(p_0)}$ using Jensen's inequality.

**Corollary 9.** *Under the assumptions of Theorems 5, 6 and 8, it follows that, for some $C > 0$*

$$\mathbb{E}_{\boldsymbol{x},\boldsymbol{y}} \left\| m_{\Pi(\cdot|\boldsymbol{x},\boldsymbol{y})} - f_0 \right\|_{L^2(p_0)}^2 \leq C n^{-\frac{2\min(\beta,\nu)}{2\nu+d}} \tag{13}$$

*where $m_{\Pi(\cdot|\boldsymbol{x},\boldsymbol{y})}$ is the posterior mean given a particular value of $(x_i, y_i)_{i=1}^n$.*

When $\nu = \beta$, the optimality of the rates we present in the manifold setting can be easily inferred by lower bounding the $L^2$-risk of the posterior mean by the $L^2$-risk over a small subset of $\mathcal{M}$ and using charts, which translates the problem into the Euclidean framework for which the rate is known to be optimal—see for instance Tsybakov [52].

To contextualize this, observe that even in cases where the geometry of the manifold is non-flat, the asymptotic rates are unaffected by the choice of the prior's length scale $\kappa$—in either the intrinsic, or the extrinsic case—but only by the smoothness parameter $\nu$. Indeed, the RKHS of the process is only determined—up to norm equivalence—by $\nu$, which plays an important role in the proofs. This, and the fact that extrinsic processes attain the same rates, implies that the study of asymptotic posterior contraction rates *cannot detect geometry* in our setting, as was already hinted by Yang and Dunson [63]. Hence, in the geometric setting, optimal posterior contraction rates should be thought of more as a basic property that any reasonable model should satisfy. Differences in performance will be down to constant factors alone—but as we will see, these can be significant. To understand these differences, we turn to empirical analysis.

## 4 Experiments

From Theorems 5, 6 and 8, we know that intrinsic and extrinsic Gaussian processes exhibit the same posterior contraction rates in the asymptotic regime. Here, we study how these rates manifest themselves in practice, by examining how worst-case errors akin to those of Corollary 9 behave numerically. Specifically, we consider the pointwise worst-case error

$$v^{(\tau)}(t) = \sup_{\|f_0\|_{H^{\nu+d/2}} \leq 1} \mathbb{E}_{\varepsilon_i \sim \mathrm{N}(0,\sigma_\varepsilon^2)} \left| m_{\Pi(\cdot|\boldsymbol{x},\boldsymbol{y})}^{(\tau)}(t) - f_0(t) \right|^2 \tag{14}$$

where $m_{\Pi(\cdot|\boldsymbol{x},\boldsymbol{y})}^{(\tau)}$ is the posterior mean corresponding to the zero-mean Matérn Gaussian process prior with smoothness $\nu$, length scale $\kappa$, amplitude $\sigma_f^2$, which is intrinsic if $\tau = \mathrm{i}$ or extrinsic if $\tau = \mathrm{e}$. We use a Gaussian likelihood with noise variance $\sigma_\varepsilon^2$ and observations $y_i = f_0(x_i) + \varepsilon_i$, and examine this quantity as a function of the evaluation location $t \in \mathcal{M}$. By allowing us to assess how error varies in different regions of the manifold, this provides us with a fine-grained picture of how posterior contraction behaves.

One can show that $v^{(\tau)}$ may be computed without numerically solving an infinite-dimensional optimization problem. Specifically, (14) can be calculated, in the respective intrinsic and extrinsic cases, using

$$v^{(\mathrm{i})}(t) = k^{(\mathrm{i})}(t,t) - \mathbf{K}_{t\mathbf{X}}^{(\mathrm{i})} \left( \mathbf{K}_{\mathbf{X}\mathbf{X}}^{(\mathrm{i})} + \sigma_\varepsilon^2 \mathbf{I} \right)^{-1} \mathbf{K}_{\mathbf{X}t}^{(\mathrm{i})} \tag{15}$$

$$v^{(\mathrm{e})}(t) \approx (\mathbf{K}_{t\mathbf{X}'}^{(\mathrm{i})} - \boldsymbol{\alpha}_t \mathbf{K}_{\mathbf{X}\mathbf{X}'}^{(\mathrm{i})})(\mathbf{K}_{\mathbf{X}'\mathbf{X}'}^{(\mathrm{i})})^{-1}(\mathbf{K}_{\mathbf{X}'t}^{(\mathrm{i})} - \mathbf{K}_{\mathbf{X}'\mathbf{X}}^{(\mathrm{i})}\boldsymbol{\alpha}_t^\top) + \sigma_\varepsilon^2 \boldsymbol{\alpha}_t \boldsymbol{\alpha}_t^\top \tag{16}$$

where, for the extrinsic case, $\boldsymbol{\alpha}_t = \mathbf{K}_{t\mathbf{X}}^{(\mathrm{i})}(\mathbf{K}_{\mathbf{X}\mathbf{X}}^{(\mathrm{e})} + \sigma_\varepsilon^2 \mathbf{I})^{-1}$, and $\mathbf{X}'$ is a set of points sampled uniformly from the manifold $\mathcal{M}$, the size of which determines approximation quality. The intrinsic expression is simply the posterior variance $k_{\Pi(\cdot|\boldsymbol{x},\boldsymbol{y})}^{(\mathrm{i})}(t,t)$, and its connection with worst-case error is a well-known folklore result mentioned somewhat implicitly in, for instance, Mutny and Krause [35]. The extrinsic expression is very-closely-related, and arises by numerically approximating a certain RKHS norm. A derivation of both is given in Appendix F. To assess the approximation error of this formula, we also consider an analog of (16) but instead defined for the intrinsic model, and compare it to (15): in all cases, the difference between the exact and approximate expression was found to be smaller than differences between models. By computing these expressions, we therefore obtain, up to numerics, the pointwise worst-case expected error in our regression model.

For $\mathcal{M}$ we consider three settings: a dumbbell-shaped manifold, a sphere, and the dragon manifold from the Stanford 3D scanning repository. In all cases, we perform computations by approximating

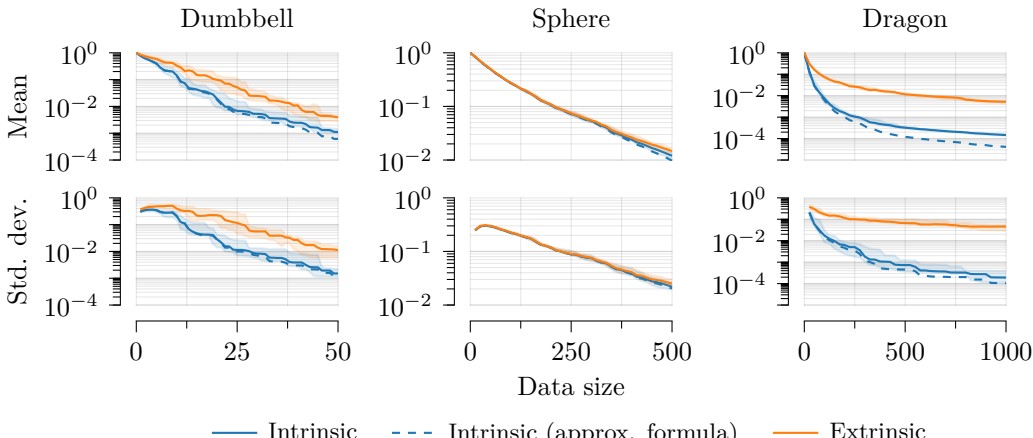

Figure 3: Worst-case error estimates for the intrinsic and extrinsic processes, on the *dumbbell*, *sphere*, and *dragon* manifolds (lower is better, $y$ axis is in the logarithmic scale). We see that, on the dumbbell and dragon manifold, intrinsic models achieve lower expected errors than extrinsic models for the ranges considered (top), and that their expected error consistently varies less as a function of space (bottom). In contrast, on the sphere, both models achieve similar performance, with differences between models falling within the range of variability caused by different random number seeds. We also see that the difference between computing the pointwise worst-case error exactly and approximately, in the intrinsic case where computing this difference is possible, is small in all cases.

the manifold using a mesh, and implementing the truncated Karhunen–Loève expansion with $J = 500$ eigenpairs obtained from the mesh. We fix smoothness $\nu = \frac{5}{2}$, amplitude $\sigma_f^2 = 1$, and noise variance $\sigma_\varepsilon^2 = 0.0005$, for both the intrinsic and extrinsic Matérn Gaussian processes. Since the interpretation of the length scale parameter is manifold-specific, for the intrinsic Gaussian processes we set $\kappa = 200$ for the dumbbell, $\kappa = 0.25$ for the sphere, and $\kappa = 0.05$ for the dragon manifold. In all cases, this yielded functions that were neither close to being globally-constant, nor resembled noise. Each experiment was repeated 10 times to assess variability. Complete experimental details are given in Appendix G.[2]

The length scales $\kappa$ are defined differently for intrinsic and extrinsic Matérn kernels: in particular, using the same length scale in both models can result in kernels behaving very differently. To alleviate this, for the extrinsic process, we set the length scale by maximizing the extrinsic process' marginal likelihood using the full dataset generated by the intrinsic process, except in the dumbbell's case where the full dataset is relatively small, and therefore a larger set of 500 points was used instead. This allows us to numerically match intrinsic and extrinsic length scales to ensure a reasonably-fair comparison.

Figure 3 shows the mean, and spatial standard deviation of $v_\tau(t)$, where by *spatial standard deviation* we mean the sample standard deviation computed with respect to locations in space, rather than with respect to different randomly sampled datasets. From this, we see that on the dumbbell and dragon manifold—whose geometry differs significantly from the respective ambient Euclidean spaces— intrinsic models obtain better mean performance. The standard deviation plot reveals that intrinsic models have errors that are less-variable across space. This means that extrinsic models exhibit higher errors in some regions rather than others—such as, for instance, regions where embedded Euclidean and Riemannian distances differ—whereas in intrinsic models the error decays in a more spatially-uniform manner.

In contrast, on the sphere, both models perform similarly. Moreover, both the mean and spatial standard deviation decrease at approximately the same rates, indicating that the extrinsic model's predictions are correct about-as-often as the intrinsic model's, as a function of space. This confirms the view that, since the sphere does not possess any bottleneck-like areas where embedded Euclidean distances are extremely different from their Riemannian analogs, it is significantly less affected by differences coming from embeddings.

---

[2]Code available at: HTTPS://GITHUB.COM/ATERENIN/GEOMETRIC_ASYMPTOTICS.

In total, our experiments confirm that there are manifolds on which geometric models can perform significantly better than non-geometric models. This phenomenon was also noticed in Dunson et al. [17], where a prior based on the eigendecomposition of a random geometric graph, which can be thought as an approximation of our intrinsic Matérn processes, is compared to a standard extrinsic Gaussian process. In our experiments, we see this through expected errors, mirroring prior results on Bayesian optimization performance. From our theoretical results, such differences cannot be captured through posterior contraction rates, and therefore would require sharper technical tools, such as non-asymptotic analysis, to quantify theoretically.

## 5 Conclusion

In this work, we studied the asymptotic behavior of Gaussian process regression with different classes of Matérn processes on Riemannian manifolds. By using various results on Sobolev spaces on manifolds we derived posterior contraction rates for intrinsic Matérn process defined via their Karhunen-Loeve decomposition in the Laplace–Beltrami eigenbasis, including processes arising from truncation of the respective sum which can be implemented in practice. Next, using trace and extension theorems which relate manifold and Euclidean Sobolev spaces, we derived similar contraction rates for the restriction of an ambient Matérn process in the case where the manifold is embedded in Euclidean space. These theoretical asymptotic results were supplemented by experiments on several examples, showing significant differences in performance between intrinsic and extrincic methods in the small sample size regime when the manifold's geometric structure differs from the ambient Euclidean space. Our work therefore shows that capturing such differences cannot be done through asymptotic contraction rates, motivating and paving the way for further work on non-asymptotic error analysis to capture empirically-observed differences between extrinsic and intrinsic models.

## Acknowledgments

The authors are grateful to Mojmír Mutný and Prof. Andreas Krause for fruitful discussions concerning this work. PR and JR were supported by the European Research Council (ERC) under the European Union's Horizon 2020 research and innovation programme (grant agreement No. 834175). VB was supported by an ETH Zürich Postdoctoral Fellowship. AT was supported by Cornell University, jointly via the Center for Data Science for Enterprise and Society, the College of Engineering, and the Ann S. Bowers College of Computing and Information Science.

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

# A  Preliminaries

Here we describe preliminaries necessary for Appendices B to E. This includes some basic properties of the Laplace–Beltrami operator on compact manifolds, partitions of unity subordinate to atlases, function spaces such as Hölder, Sobolev and Besov spaces, general Gaussian random elements on Banach spaces, and a certain technical lemma. Hereinafter, the expression $a \lesssim b$ means $a \leq Cb$ for some constant $C > 0$ whose value is irrelevant to our claims.

## A.1  Laplace–Beltrami Operator and Subordinate Partitions of Unity

Let $\mathcal{M}$ denote a compact connected Riemannian manifold without boundary of dimension $d \in \mathbb{Z}_{>0}$. The Laplace–Beltrami operator $\Delta$ on $\mathcal{M}$ is self-adjoint and positive semi-definite [48, Theorem 2.4]. Let $(L^2(\mathcal{M}), \langle \cdot, \cdot \rangle_{L^2(\mathcal{M})})$ denote the Hilbert space of square integrable (equivalence classes of) functions on $\mathcal{M}$ with respect to the standard Riemannian volume measure.

By standard theory [13, 23], there exists an orthonormal basis $\{f_j\}_{j=0}^{\infty}$ of $L^2(\mathcal{M})$ consisting of the eigenfunctions of $\Delta$, namely $\Delta f_j = -\lambda_j f_j$, with $\lambda_j \geq 0$. We assume that the pairs $(\lambda_j, f_j)$ are sorted such that $0 = \lambda_0 \leq \lambda_j \leq \lambda_{j+1}$. The growth of $\lambda_j$ can be characterized as follows.

**Result 10** (Weyl's Law). *There exists a constant $C > 0$ such that for all $j$ large enough we have*

$$C^{-1} j^{2/d} \leq \lambda_j \leq C j^{2/d}. \tag{17}$$

*Proof.* See Chavel [13], Chapter 1. $\qquad \square$

Following De Vito et al. [16] and Große and Schneider [24] we fix a family $\mathcal{T} = (\mathcal{U}_l, \phi_l, \chi_l)_{l=1}^{L}$, where $L \in \mathbb{Z}_{>0}$, the local coordinates $\phi_l : \mathcal{U}_l \subset \mathcal{M} \to \mathcal{V}_l = \phi_l(\mathcal{U}_l) \subset \mathbb{R}^d$ are smooth diffeomorphisms, and the functions $\chi_l$ form a partition of unity subordinate to $\{\mathcal{U}_l\}_{l=1}^{L}$, that is $\chi_l \in \mathcal{C}^{\infty}(\mathcal{M})$, $\mathrm{supp}(\chi_l) \subset \mathcal{U}_l$, $0 \leq \chi_l \leq 1$ and $\sum_l \chi_l = 1$—here, we can choose $L$ finite by compactness of $\mathcal{M}$. For convenience and without loss of generality, we assume that $\mathcal{V}_l = (0,1)^d$. With this, we can start defining function spaces on $\mathcal{M}$.

## A.2  Hölder Spaces $\mathcal{CH}^{\gamma}$ and the spaces of continuous functions $\mathcal{C}$ and smooth functions $\mathcal{C}^{\infty}$

Consider an arbitrary domain $\mathcal{X} \subseteq \mathbb{R}^d$ or $\mathcal{X} \subseteq \mathcal{M}$. We denote the class of infinitely differentiable functions on $\mathcal{X}$ by $\mathcal{C}^{\infty}(\mathcal{X})$. Let $\mathcal{C}^k(\mathcal{X})$ denote the Banach space of $k \in \mathbb{Z}_{>0}$ times continuously differentiable functions on $\mathcal{X}$ with finite norm

$$\|f\|_{\mathcal{C}^k(\mathcal{X})} = \sup_{x \in \mathcal{X}} |\nabla^k f(x)| \tag{18}$$

where $\nabla^k$ is the $k$th covariant derivative, as in Hebey [25], Section 2.1 and Aubin [3], Definition 2.2. We also write $\mathcal{C}(\mathcal{X}) = \mathcal{C}^0(\mathcal{X})$ for the space of continuous functions on $\mathcal{X}$.

The Euclidean Hölder spaces $\mathcal{CH}^{\gamma}(\mathbb{R}^d)$, where $\gamma = k + \alpha$ with $k \in \mathbb{Z}_{\geq 0}$ and $0 < \alpha \leq 1$, are defined by[3]

$$\mathcal{CH}^{\gamma}(\mathbb{R}^d) = \left\{ f \in \mathcal{C}^k(\mathbb{R}^d) : \|f\|_{\mathcal{CH}^{\gamma}(\mathbb{R}^d)} < \infty \right\} \tag{19}$$

where

$$\|f\|_{\mathcal{CH}^{\gamma}(\mathbb{R}^d)} = \|f\|_{\mathcal{C}^k(\mathbb{R}^d)} + \sup_{\boldsymbol{x}, \boldsymbol{y} \in \mathbb{R}^d,\, \boldsymbol{x} \neq \boldsymbol{y}} \frac{|f(\boldsymbol{x}) - f(\boldsymbol{y})|}{\|\boldsymbol{x} - \boldsymbol{y}\|_{\mathbb{R}^d}^{\alpha}}. \tag{20}$$

More information on these definitions may be found, for instance, in Giné and Nickl [21] and Triebel [51]. We now turn to the manifold versions of the Hölder spaces.

---

[3] Hölder spaces are often also denoted by $\mathcal{C}^{\gamma}$ as well, with $\gamma \in \mathbb{R}_{>0}$. Since, using this formulation, they do *not* coincide with $\mathcal{C}^k(\mathcal{X})$ when $k = \gamma \in \mathbb{Z}_{\geq 0}$, we use the notation $\mathcal{CH}^{\gamma}$ to avoid confusion.

**Definition 11** (Hölder spaces). *For all $\gamma > 0$ we define the Hölder space $\mathcal{CH}^\gamma(\mathcal{M})$ on the manifold $\mathcal{M}$ to be the space of all $f : \mathcal{M} \to \mathbb{R}$ satisfying*

$$\|f\|_{\mathcal{CH}^\gamma(\mathcal{M})} = \sum_{l=1}^{L} \left\|(\chi_l f) \circ \phi_l^{-1}\right\|_{\mathcal{CH}^\gamma(\mathbb{R}^d)} < \infty. \tag{21}$$

Since the charts $\phi_l$ are smooth, Definition 11 can be easily seen to be independent of the chosen atlas, with equivalence of norms.

### A.3   Sobolev and Besov Spaces

We now introduce the manifold versions of the Sobolev and Besov spaces, whose definitions in the standard Euclidean case may be found, for instance, in Triebel [51]. For Sobolev spaces we use the Bessel-potential-based definition, following De Vito et al. [16].

**Definition 12** (Sobolev spaces). *For any $s > 0$ we define the Sobolev space $H^s(\mathcal{M})$ on the manifold $\mathcal{M}$ as the Hilbert space of functions $f \in L^2(\mathcal{M})$ such that $\|f\|_{H^s(\mathcal{M})}^2 = \langle f, f \rangle_{H^s(\mathcal{M})} < \infty$ where*

$$\langle f, g \rangle_{H^s(\mathcal{M})} = \sum_{j=0}^{\infty} (1 + \lambda_j)^s \langle f, f_j \rangle_{L^2(\mathcal{M})} \langle g, f_j \rangle_{L^2(\mathcal{M})}. \tag{22}$$

**Remark 13.** *It is easy to see that substituting $(1 + \lambda_j)^s$ in (22) with $\beta(\alpha + \lambda_j)^s$ or with $(\alpha + \beta\lambda_j^s)$ for any $\alpha, \beta > 0$ results in the same set of functions and an equivalent norm. The former follows from Borovitskiy et al. [11], eq. (109). The latter follows from the Binomial Theorem.*

For Besov spaces we follow Coulhon et al. [14] and Castillo et al. [12] and define them in terms of approximations by low-frequency functions. We fix a function $\Phi \in \mathcal{C}^\infty(\mathbb{R}_{\geq 0}, \mathbb{R}_{\geq 0})$ such that $K = \text{supp}(\Phi) \subset [0, 2]$ and $\Phi(x) = 1$ for $x \in [0, 1]$. We also define the functions $\Phi_j(x) = \Phi(2^{-j}x)$.

Coulhon et al. [14], Corollary 3.6 shows that the operators $\Phi_j(\sqrt{\Delta})$ defined by

$$\Phi_j\left(\sqrt{\Delta}\right)f = \sum_{j \geq 0} \Phi_j\left(\sqrt{\lambda_j}\right)\langle f_j, f \rangle f_j \tag{23}$$

are bounded in the space $L^p(\mathcal{M})$ for all $1 \leq p \leq \infty$.[4] Moreover, the same result also shows that we can express any $f \in L^p(\mathcal{M})$ as $f = \lim_{j \to \infty} \Phi_j(\sqrt{\Delta})f$ in $L^p(\mathcal{M})$. $\Phi_j(\sqrt{\Delta})f$ can intuitively be considered as a version of $f$ filtered by a low-pass filter. The next definition introduces the Besov spaces $B_{p,q}^s(\mathcal{M})$, which are formulated in terms of quality-of-approximation by low-frequency functions.

**Definition 14** (Besov spaces). *For any $s > 0$ and $1 \leq p, q \leq \infty$ we define the Besov space $B_{p,q}^s(\mathcal{M})$ on the manifold $\mathcal{M}$ as the space of functions $f \in L^p(\mathcal{M})$ such that $\|f\|_{B_{p,q}^s(\mathcal{M})} < \infty$ where*

$$\|f\|_{B_{p,q}^s(\mathcal{M})} = \begin{cases} \|f\|_{L^p(\mathcal{M})} + \left(\sum_{j \geq 0}\left(2^{js}\|\Phi_j(\sqrt{\Delta})f - f\|_{L^p(\mathcal{M})}\right)^q\right)^{1/q} & \text{if } q < +\infty \\ \|f\|_{L^p(\mathcal{M})} + \sup_{j \geq 0} 2^{js}\|\Phi_j(\sqrt{\Delta})f - f\|_{L^p(\mathcal{M})} & \text{if } q = +\infty. \end{cases} \tag{24}$$

The classical Besov spaces $B_{2,2}^s$ coincide with the Sobolev spaces $H^s$ on $\mathbb{R}^d$, in the sense that they define the same set of functions and equivalent norms—see for instance Giné and Nickl [21] section 4.3.6—and even on manifolds if one follows the construction of Triebel [51], pages 7.3–7.4 for Besov spaces. Since our definition of the Besov spaces is somewhat non-standard, we present a proof.

**Proposition 15.** *For all $s > 0$, $H^s(\mathcal{M}) = B_{2,2}^s(\mathcal{M})$ as sets and there exist two constants $C_1, C_2 > 0$ such that for all $f \in H^s(\mathcal{M}) = B_{2,2}^s(\mathcal{M})$ we have*

$$C_1\|f\|_{H^s(\mathcal{M})} \leq \|f\|_{B_{2,2}^s(\mathcal{M})} \leq C_2\|f\|_{H^s(\mathcal{M})}. \tag{25}$$

---

[4]The space $L^p(\mathcal{M})$ is the Banach space of functions (or rather their equivalence classes) that are integrable when raised to the power $p < \infty$ or essentially bounded for $p = \infty$. See for instance Triebel [50] for details.

*Proof.* It is enough to prove (25), the rest will follow automatically. The main technical tools used in the proof are Result 10 and summation by parts. First, we prove the upper bound. Denote the support set of $\Phi$ by $K = \text{supp}(\Phi)$. Notice that

$$\left(\|f\|_{B_{2,2}^s(\mathcal{M})} - \|f\|_{L^2(\mathcal{M})}\right)^2 = \sum_{j \geq 0} 2^{2js} \left\|\Phi_j\left(\sqrt{\Delta}\right) f - f\right\|_{L^2(\mathcal{M})}^2 \tag{26}$$

$$= \sum_{j \geq 0} 2^{2js} \sum_{l:\sqrt{\lambda_l} \notin 2^j K} \left|\langle f_l, f \rangle_{L^2(\mathcal{M})}\right|^2 \tag{27}$$

$$\leq \sum_{j \geq 0} 2^{2js} \sum_{l:\sqrt{\lambda_l} > 2^j} \left|\langle f_l, f \rangle_{L^2(\mathcal{M})}\right|^2. \tag{28}$$

The last inequality results from the fact that $[0, 1] \subset K$. By Weyl's Law (Result 10), there exists a constant $C > 0$ such that $\lambda_l \leq C l^{2/d}$. Without loss of generality, we can assume that $C = 2^{2r}$, $r \in \mathbb{Z}_{>0}$. Hence $\sqrt{\lambda_l} > 2^j$ implies $l > 2^{d(j-r)}$, thus we have

$$\left(\|f\|_{B_{2,2}^s(\mathcal{M})} - \|f\|_{L^2(\mathcal{M})}\right)^2 \leq \sum_{j \geq 0} 2^{2js} \sum_{l > 2^{d(j-r)}} \left|\langle f_l, f \rangle_{L^2(\mathcal{M})}\right|^2 \tag{29}$$

$$= \sum_{0 \leq j \leq r} 2^{2js} \sum_{l > 2^{d(j-r)}} \left|\langle f_l, f \rangle_{L^2(\mathcal{M})}\right|^2 + \sum_{j > r} 2^{2js} \sum_{l > 2^{d(j-r)}} \left|\langle f_l, f \rangle_{L^2(\mathcal{M})}\right|^2 \tag{30}$$

$$\leq r 2^{2rs} \|f\|_{L^2(\mathcal{M})}^2 + 2^{2rs} \sum_{j > 0} 2^{2js} \sum_{l > 2^{dj}} \left|\langle f_l, f \rangle_{L^2(\mathcal{M})}\right|^2. \tag{31}$$

Now let $R_j = \sum_{l > 2^{dj}} \left|\langle f_l, f \rangle_{L^2(\mathcal{M})}\right|^2$ and $S_J = \sum_{j=1}^J 2^{2js} \leq \frac{2^{2s}}{2^{2s}-1} 2^{2Js}$, $S_0 = 0$. Write

$$\sum_{j > 0} 2^{2js} \sum_{l > 2^{dj}} \left|\langle f_l, f \rangle_{L^2(\mathcal{M})}\right|^2 = \sum_{j > 0} (S_j - S_{j-1}) R_j \tag{32}$$

$$= \sum_{j > 0} S_j (R_j - R_{j+1}) \tag{33}$$

$$= \sum_{j > 0} S_j \sum_{2^{dj} < l \leq 2^{d(j+1)}} \left|\langle f_l, f \rangle_{L^2(\mathcal{M})}\right|^2 \tag{34}$$

$$\leq \frac{2^{2s}}{2^{2s}-1} \sum_{j > 0} 2^{2js} \sum_{2^{dj} < l \leq 2^{d(j+1)}} \left|\langle f_l, f \rangle_{L^2(\mathcal{M})}\right|^2 \tag{35}$$

$$\leq \frac{2^{2s}}{2^{2s}-1} \sum_{j > 0} \sum_{2^{dj} < l \leq 2^{d(j+1)}} l^{2s/d} \left|\langle f_l, f \rangle_{L^2(\mathcal{M})}\right|^2 \tag{36}$$

$$\leq \frac{c^s 2^{2s}}{2^{2s}-1} \sum_{j > 0} \sum_{2^{dj} < l \leq 2^{d(j+1)}} \lambda_l^s \left|\langle f_l, f \rangle_{L^2(\mathcal{M})}\right|^2 \tag{37}$$

$$= \frac{c^s 2^{2s}}{2^{2s}-1} \sum_{l > 2^d} \lambda_l^s \left|\langle f_l, f \rangle_{L^2(\mathcal{M})}\right|^2 \tag{38}$$

$$\leq \frac{c^s 2^{2s}}{2^{2s}-1} \sum_{l \geq 0} \lambda_l^s \left|\langle f_l, f \rangle_{L^2(\mathcal{M})}\right|^2 \tag{39}$$

where we have used Result 10 to get existence of a $c$ such that $l^{2/d} \leq c\lambda_l$. This proves the upper bound. The proof for the lower bound is similar. $\qquad\square$

Proposition 15 provides a characterization of the Sobolev spaces $H^s(\mathcal{M})$. There is yet another characterization of these spaces that will be useful later, in terms of charts. We present this characterization as part of the following result.

**Theorem 16.** *On the Sobolev space $H^s(\mathcal{M})$, the following norms are equivalent:*

$$\|f\|_{H^s(\mathcal{M})} = \left( \sum_{j=0}^{\infty} (1+\lambda_j)^s \langle f, f_j \rangle_{L^2(\mathcal{M})}^2 \right)^{1/2} \tag{40}$$

$$\|f\|_{B_{2,2}^s(\mathcal{M})} = \|f\|_{L^2(\mathcal{M})} + \left( \sum_{j\geq 0} \left( 2^{js} \|\Phi_j(\sqrt{\Delta})f - f\|_{L^2(\mathcal{M})} \right)^2 \right)^{1/2} \tag{41}$$

$$\|f\|_{H_{\mathcal{T}}^s(\mathcal{M})} = \left( \sum_{l=1}^{L} \left\| (\chi_l f) \circ \phi_l^{-1} \right\|_{H^s(\mathbb{R}^d)}^2 \right)^{1/2} \tag{42}$$

*Proof.* The equivalence between $\|\cdot\|_{H^s(\mathcal{M})}$ and $\|f\|_{B_{2,2}^s(\mathcal{M})}$ is given by Proposition 15. The proof of equivalence between $\|\cdot\|_{H^s(\mathcal{M})}$ and $\|f\|_{H_{\mathcal{T}}^s(\mathcal{M})}$ can be found in De Vito et al. [16]. $\qquad\square$

### A.4 Gaussian Random Elements

Here we recall the definition of a Gaussian process as a Banach-space-valued random variable, following for instance van Zanten and van der Vaart [59].

**Definition 17** (Gaussian random element). *Let $(\mathbb{B}, \|\cdot\|_{\mathbb{B}})$ be a Banach space, and $f$ be a Borel random variable with values in $\mathbb{B}$ almost surely. We say that $f$ is a Gaussian random element if $b^*(f)$ is a univariate Gaussian random variable for every bounded linear functional $b^* : \mathbb{B} \to \mathbb{R}$.*

Random variables of this kind are also sometimes called *Gaussian in the sense of duality*. One should think of a Gaussian random element as a generalization of a Gaussian process, but which is better-behaved from a function-analytic point of view and in particular does not require the process to be an actual function—as opposed to, for instance, a measure or a distribution. Many connections between the usual Gaussian processes and Gaussian random elements exist, see Lifshits [32], Ghosal and van der Vaart [20], Appendix I, van der Vaart and van Zanten [57] for details. The following observation about Gaussian random elements will be useful later.

**Lemma 18.** *A Gaussian process $f$ on the manifold $\mathcal{M}$ with almost surely continuous sample paths is a Gaussian random element in the Banach space $\left( \mathcal{C}(\mathcal{M}), \|\cdot\|_{\mathcal{C}(\mathcal{M})} \right)$ of continuous functions on $\mathcal{M}$.*

*Proof.* Since $\mathcal{C}(\mathcal{M})$ is separable, this follows from Lemma I.6 in Ghosal and van der Vaart [20]. $\quad\square$

### A.5 A Technical Lemma

In order to apply Bernstein's inequality when going from the error at input locations to the $L^2(p_0)$ error, we will use the following technical extrapolation lemma.

**Lemma 19.** *For any function $g : \mathcal{M} \to \mathbb{R}$, a number $\gamma \in \mathbb{R}_{>0} \setminus \mathbb{Z}_{>0}$ and a density $p_0 : \mathcal{M} \to \mathbb{R}_{>0}$ with $1 \lesssim p_0$, we have*

$$\|g\|_{L^\infty(\mathcal{M})} \lesssim \|g\|_{\mathcal{CH}^\gamma(\mathcal{M})}^{\frac{d}{2\gamma+d}} \|g\|_{L^2(p_0)}^{\frac{2\gamma}{2\gamma+d}}. \tag{43}$$

*Proof.* We use Lemma 15 from van der Vaart and van Zanten [56] and push it through charts. More precisely we have, using $B_{\infty,\infty}^\gamma([0,1]^D) = \mathcal{CH}^\gamma([0,1]^D)$ for $\gamma \notin \mathbb{Z}_{>0}$, that

$$\|g\|_{L^\infty(\mathcal{M})} \leq \sum_l \left\| (\chi_l g) \circ \phi_l^{-1} \right\|_{L^\infty(\mathcal{V}_l)} \tag{44}$$

$$\lesssim \max_l \left\| (\chi_l g) \circ \phi_l^{-1} \right\|_{\mathcal{CH}^\gamma(\mathcal{V}_l)}^{\frac{d}{2\gamma+d}} \left\| (\chi_l g) \circ \phi_l^{-1} \right\|_{L^2(\mathcal{V}_l)}^{\frac{2\gamma}{2\gamma+d}}. \tag{45}$$

By definition of the the manifold Hölder spaces, this gives

$$\|g\|_{L^\infty(\mathcal{M})} \lesssim \|g\|_{\mathcal{CH}^\gamma(\mathcal{M})}^{\frac{d}{2\gamma+d}} \max_l \left\| (\chi_l g) \circ \phi_l^{-1} \right\|_{L^2(\mathcal{V}_l)}^{\frac{2\gamma}{2\gamma+d}}. \tag{46}$$

Finally, since $\chi_l$ are bounded, the charts are smooth, and $p_0$ is lower bounded, we have

$$\left\|(\chi_l g) \circ \phi_l^{-1}\right\|_{L^2(\mathcal{V}_l)}^2 = \int_{\mathcal{V}_l} \left|(\chi_l g) \circ \phi_l^{-1}(y)\right|^2 dy \lesssim \int_{\mathcal{U}_l} g^2(x) p_0(x) \mu(dx) \lesssim \|g\|_{L^2(\mathcal{M})}^2 \qquad (47)$$

which gives the result. $\qquad \square$

This concludes the necessary preliminaries. We now turn to the proofs of our main results.

## B  Proofs

Throughout this section we use the notation from Appendix A and results from Appendices C to E. We start by defining our notation for Gaussian likelihood and probability distribution of the sample.

**Definition 20.** *For every $\boldsymbol{x} \in \mathcal{M}^n$ and $f : \mathcal{M} \to \mathbb{R}$ we define $p_{\boldsymbol{x},\boldsymbol{y}}^{(f)}$ to be the joint distribution corresponding to the marginal $p_{\boldsymbol{x}} = p_0$ and conditional $p_{\boldsymbol{y}|\boldsymbol{x}}^{(f)} = \mathrm{N}(f(\boldsymbol{x}), \sigma_\varepsilon^2 \mathbf{I})$, where $f(\boldsymbol{x})$ is the vector with entries $f(x_i)$. Expectations with respect to $p_{\boldsymbol{x},\boldsymbol{y}}^{(f)}$, to $p_{\boldsymbol{x}}$ and to $p_{\boldsymbol{y}|\boldsymbol{x}}^{(f)}$ we denote by $\mathbb{E}_{\boldsymbol{x},\boldsymbol{y}}^{(f)}$, by $\mathbb{E}_{\boldsymbol{x}}$ and by $\mathbb{E}_{\boldsymbol{y}|\boldsymbol{x}}^{(f)}$, respectively. Sometimes, when $f$ is clear from the context, we omit the subscript $^{(f)}$.*

Using van der Vaart and van Zanten [56], Theorem 1, which is valid for any compact metric space and thus also for $\mathcal{M}$, we can deduce a posterior contraction rate at data input locactions, with respect to the *empirical $L^2$-norm*[5]

$$\|f\|_n = \left( \frac{1}{n} \sum_{i=1}^n f(x_i)^2 \right)^{1/2} \qquad (48)$$

by studying the *concentration functions* with respect to the uniform norm. This is the purpose of the following lemma, whose proof mainly follows [56], but crucially relies on two new components: (i) the *small ball asymptotics* for manifolds we study in Appendix E and (ii) the characterization of the RKHS corresponding to both intrinsic and extrinsic priors as manifold Sobolev spaces we obtain in Appendix C. We recall that the prior $\Pi_n$ may depend on $n$ if we consider a truncated intrinsic Matérn process.

**Theorem 21.** *Let $\Pi_n$ denote the prior in either Theorem 5, Theorem 6 or Theorem 8 with smoothness parameter $\nu > d/2$. Let $\mathbb{H}_n$ denote the corresponding RKHS. Define the CONCENTRATION FUNCTION for $f_0 \in C(\mathcal{M})$ and $\varepsilon > 0$ by*

$$\varphi_{f_0}(\varepsilon) = -\ln \mathbb{P}_{f \sim \Pi_n}\left[ \|f\|_{L^\infty(\mathcal{M})} < \varepsilon \right] + \inf_{f \in \mathbb{H}_n : \|f - f_0\|_{L^\infty(\mathcal{M})} < \varepsilon} \|f\|_{\mathbb{H}_n}^2. \qquad (49)$$

*Then if $f_0 \in H^\beta(\mathcal{M}) \cap B_{\infty,\infty}^\beta(\mathcal{M}), \beta > 0$ we have $\varphi_{f_0}(\varepsilon_n) \leq n\varepsilon_n^2$ for $\varepsilon_n$ a multiple of $n^{-\frac{\min(\nu,\beta)}{2\nu+d}}$.*

*Proof.* The first term on the right-hand side of Equation (49) is bounded by $C\varepsilon^{-d/\nu}$ by Lemma 33. To bound the second term, we assume, without loss of generality,[6] that $\nu \geq \beta$. Consider an approximation $f = \Phi_j(\sqrt{\Delta})f_0$ of $f_0$, where $c\varepsilon \leq 2^{-\beta j} \leq \varepsilon$ and $c > 0$ does not depend on $j$. Since we assume $f_0 \in B_{\infty,\infty}^\beta(\mathcal{M})$, by definition of $B_{\infty,\infty}^\beta(\mathcal{M})$ we have

$$\|f_0 - f\|_{L^\infty(\mathcal{M})} \leq \|f_0\|_{B_{\infty,\infty}^\beta(\mathcal{M})} 2^{-\beta j} \lesssim \varepsilon \qquad (50)$$

where in the last inequality the $B_{\infty,\infty}^\beta(\mathcal{M})$-norm is the constant implied by notation $\lesssim$. We now show that

$$\|f\|_{\mathbb{H}_n}^2 \lesssim \varepsilon^{-\frac{2}{\beta}(\nu-\beta+d/2)}. \qquad (51)$$

First notice that by Lemma 24 and Proposition 27, for any prior considered here we have $\mathbb{H}_n \subseteq H^{\nu+d/2}(\mathcal{M})$ and $\|\cdot\|_{\mathbb{H}_n} \lesssim \|\cdot\|_{H^{\nu+d/2}(\mathcal{M})}$ with a constant that does not depend on $n$. Hence using

---

[5]This is actually a seminorm, but we follow the rest of the literature in referring to it as a norm.

[6]If $\beta > \nu$ then $f_0 \in H^\beta(\mathcal{M}) \cap B_{\infty,\infty}^\beta \subseteq H^\nu(\mathcal{M}) \cap B_{\infty,\infty}^\nu(\mathcal{M})$ gives $\varepsilon_n \propto n^{-\frac{\nu}{2\nu+d}} = n^{-\frac{\min(\beta,\nu)}{2\nu+d}}$.

Result 10 (Weyl's Law) and properties of $\Phi$, we have

$$\|f\|_{\mathbb{H}_n}^2 \lesssim \|f\|_{H^{\nu+d/2}(\mathcal{M})}^2 \tag{52}$$

$$= \sum_{l \geq 0} (1 + \lambda_l)^{\nu+d/2} \Phi^2 \left(2^{-j}\sqrt{\lambda_l}\right) \left|\langle f_l, f_0 \rangle_{L^2(\mathcal{M})}\right|^2 \tag{53}$$

$$\leq \sum_{l:\sqrt{\lambda_l} \leq 2^{j+1}} (1 + \lambda_l)^{\nu+d/2-\beta} (1 + \lambda_l)^{\beta} \left|\langle f_l, f_0 \rangle_{L^2(\mathcal{M})}\right|^2 \tag{54}$$

$$\leq 2^{(j+2)(\nu+d/2-\beta)} \sum_{l:\sqrt{\lambda_l} \leq 2^{j+1}} (1 + \lambda_l)^{\beta} \left|\langle f_l, f_0 \rangle_{L^2(\mathcal{M})}\right|^2 \tag{55}$$

$$\leq 2^{(j+2)(\nu+d/2-\beta)} \sum_{l \geq 0} (1 + \lambda_l)^{\beta} \left|\langle f_l, f_0 \rangle_{L^2(\mathcal{M})}\right|^2 \tag{56}$$

$$= 2^{(j+2)(\nu+d/2-\beta)} \|f_0\|_{H^{\beta}(\mathcal{M})}^2 \tag{57}$$

$$\lesssim \varepsilon^{-\frac{1}{\beta}(\nu+d/2-\beta)} \|f_0\|_{H^{\beta}(\mathcal{M})}^2 \lesssim \varepsilon^{-\frac{2}{\beta}(\nu+d/2-\beta)} \|f_0\|_{H^{\beta}(\mathcal{M})}^2. \tag{58}$$

Our assumption $\nu \geq \beta$ implies that

$$\frac{2}{\beta}(\nu - \beta + d/2) \geq \frac{d}{\beta} \geq \frac{d}{\nu}. \tag{59}$$

Hence, we have $\varepsilon^{-d/\nu} \leq \varepsilon^{-\frac{2}{\beta}(\nu-\beta+d/2)}$ which gives us $\varphi_{f_0}(\varepsilon) \lesssim \varepsilon^{-\frac{2}{\beta}(\nu-\beta+d/2)}$. It is then easy to check that $\varepsilon_n = Mn^{-\frac{\beta}{2\nu+d}}$ satisfies $\varphi_{f_0}(\varepsilon_n) \leq n\varepsilon_n^2$ for $M > 0$ large enough. $\qquad \square$

From this, we deduce an upper bound on the error in *the empirical $L^2$ norm* $\|\cdot\|_n$, that is, on the Euclidean distance between the posterior Gaussian process $f$ and the ground truth function $f_0$ evaluated at data locations $x_i$.

**Lemma 22.** *Let $\Pi_n$ denote the prior in either Theorem 5, Theorem 6 or Theorem 8 with smoothness parameter $\nu > d/2$. Fix $f_0 \in H^{\beta}(\mathcal{M}) \cap B_{\infty,\infty}^{\beta}(\mathcal{M})$ with $\beta > 0$. Then*

$$\mathbb{E}_{f \sim \Pi_n(\cdot|\boldsymbol{x},\boldsymbol{y})} \|f - f_0\|_n^2 \leq \varepsilon_n^2 \tag{60}$$

*for $\varepsilon_n \propto n^{-\frac{\min(\nu,\beta)}{2\nu+d}}$ with constant depending on $f_0, \nu$ but not on $\boldsymbol{x}$.*

*Proof.* By Theorem 21 for $\varepsilon_n$ a multiple of $n^{-\frac{\min(\beta,\nu)}{2\nu+d}}$, we have $\varphi_{f_0}(\varepsilon_n) \leq n\varepsilon_n^2$. By virtue of this, the proof of Theorem 1 and Proposition 11 of van der Vaart and van Zanten [56] imply the result. Indeed, the proof of Theorem 1 relies solely on the fact that $\varphi_{f_0}(\varepsilon_n/2) \leq n\varepsilon_n^2$ and an application of van der Vaart and van Zanten [56], Proposition 11. We have $\varphi_{f_0}(\varepsilon_n) \leq n\varepsilon_n^2 \leq n(2\varepsilon_n)^2$ and hence the condition is satisfied with $\varepsilon_n$ replaced by $2\varepsilon_n$. $\qquad \square$

We now turn to the proofs of our main results, Theorems 5, 6 and 8, which for convenience we restate below. The idea of these proofs is to extend the result of Lemma 22 from input locations to the whole manifold $\mathcal{M}$ using an appropriate concentration inequality. To this end, the proof closely follows the one of Theorem 2 in van der Vaart and van Zanten [56], but relies on Lemma 22 proved above and on the concentration inequality we prove in Appendix D along with some important sample differentiablity properties of the prior processes.

**Theorem 5.** *Let $f$ be a Riemannian Matérn Gaussian process prior of Definition 4 with smoothness parameter $\nu > d/2$ and let $f_0$ satisfy Assumption 3. Then there is a $C > 0$ such that*

$$\mathbb{E}_{\boldsymbol{x},\boldsymbol{y}} \mathbb{E}_{f \sim \Pi(\cdot|\boldsymbol{x},\boldsymbol{y})} \|f - f_0\|_{L^2(p_0)}^2 \leq Cn^{-\frac{2\min(\beta,\nu)}{2\nu+d}}. \tag{9}$$

**Theorem 6.** *Let $f$ be a Riemannian Matérn Gaussian process prior on $\mathcal{M}$ with smoothness parameter $\nu > d/2$, modified to truncate the infinite sum to at least $J_n \geq cn^{\frac{d(\min(1,\nu/\beta))}{2\nu+d}}$ terms, and let $f_0$ satisfy Assumption 3. Then there is a $C > 0$ such that*

$$\mathbb{E}_{\boldsymbol{x},\boldsymbol{y}} \mathbb{E}_{f \sim \Pi(\cdot|\boldsymbol{x},\boldsymbol{y})} \|f - f_0\|_{L^2(p_0)}^2 \leq Cn^{-\frac{2\min(\beta,\nu)}{2\nu+d}}. \tag{10}$$

**Theorem 8.** *Let $f$ be a mean-zero extrinsic Matérn Gaussian process prior with smoothness parameter $\nu > d/2$ on $\mathcal{M}$, and let $f_0$ satisfy Assumption 3. Then for some $C > 0$ we have*

$$\mathbb{E}_{\boldsymbol{x},\boldsymbol{y}} \, \mathbb{E}_{f \sim \Pi(\cdot|\boldsymbol{x},\boldsymbol{y})} \|f - f_0\|_{L^2(p_0)}^2 \leq C n^{-\frac{2\min(\beta,\nu)}{2\nu+d}}. \tag{12}$$

*Proof of Theorems 5, 6 and 8.* To ease notation, we denote the expectations under the true data generating process by $\mathbb{E}_{\boldsymbol{x},\boldsymbol{y}} = \mathbb{E}_{\boldsymbol{x},\boldsymbol{y}}^{(f_0)}$ and $\mathbb{E}_{\boldsymbol{y}|\boldsymbol{x}} = \mathbb{E}_{\boldsymbol{y}|\boldsymbol{x}}^{(f_0)}$, omitting the superscript $(\cdot)^{f_0}$. Take $\varepsilon_n \propto n^{-\frac{\min(\beta,\nu)}{2\nu+d}}$ satisfying $\varphi_{f_0}(\varepsilon_n/2) \leq n\varepsilon_n^2$, noting that such a rate exists by Theorem 21. Then, for each $n$, there exists an element $f_n \in \mathbb{H}_n$, where $\mathbb{H}_n$ is the RKHS corresponding to $\Pi_n$, satisfying

$$\|f_n\|_{\mathbb{H}_n}^2 \leq n\varepsilon_n^2 \qquad\qquad \|f_n - f_0\|_{L^\infty(\mathcal{M})} \leq \varepsilon_n/2. \tag{61}$$

Write

$$\varepsilon_n^{-2} \, \mathbb{E}_{\boldsymbol{x},\boldsymbol{y}} \, \mathbb{E}_{f \sim \Pi_n(\cdot|\boldsymbol{x},\boldsymbol{y})} \|f - f_0\|_{L^2(p_0)}^2 \lesssim \varepsilon_n^{-2} \, \mathbb{E}_{\boldsymbol{x},\boldsymbol{y}} \, \mathbb{E}_{f \sim \Pi_n(\cdot|\boldsymbol{x},\boldsymbol{y})} \|f_n - f_0\|_{L^2(p_0)}^2 \tag{62}$$

$$+ \varepsilon_n^{-2} \, \mathbb{E}_{\boldsymbol{x},\boldsymbol{y}} \, \mathbb{E}_{f \sim \Pi_n(\cdot|\boldsymbol{x},\boldsymbol{y})} \|f - f_n\|_{L^2(p_0)}^2 \tag{63}$$

$$\lesssim 1 + \varepsilon_n^{-2} \, \mathbb{E}_{\boldsymbol{x},\boldsymbol{y}} \, \mathbb{E}_{f \sim \Pi_n(\cdot|\boldsymbol{x},\boldsymbol{y})} \|f - f_n\|_{L^2(p_0)}^2. \tag{64}$$

Thus, we can work with $f - f_n$ instead of $f - f_0$. Define $\mathcal{B}(r) = \{\|f - f_n\|_{L^2(p_0)} > \varepsilon_n r\}$. Then

$$\varepsilon_n^{-2} \, \mathbb{E}_{f \sim \Pi_n(\cdot|\boldsymbol{x},\boldsymbol{y})} \|f - f_n\|_{L^2(p_0)}^2. = \int_0^\infty 2r \, \mathbb{P}_{f \sim \Pi_n(\cdot|\boldsymbol{x},\boldsymbol{y})}(\mathcal{B}(r)) \, \mathrm{d}r. \tag{65}$$

Fix a $\gamma$ such that $d/2 < \gamma < \nu, \gamma \notin \mathbb{Z}_{>0}$ and $s > 0, \tau > 0$ and define

$$\mathcal{B}^{(\mathrm{I})}(r) = \{2\|f - f_n\|_n > \varepsilon_n r\} \tag{66}$$

$$\mathcal{B}^{(\mathrm{II})}(r) = \left\{\|f\|_{\mathcal{CH}^\gamma(\mathcal{M})} > \tau\sqrt{n}\varepsilon_n r^s\right\} \tag{67}$$

$$\mathcal{B}^{(\mathrm{III})}(r) = \mathcal{B}(r) \setminus \left(\mathcal{B}^{(\mathrm{I})}(r) \cup \mathcal{B}^{(\mathrm{II})}(r)\right). \tag{68}$$

Then $\mathcal{B}(r) \subseteq \mathcal{B}^{(\mathrm{I})}(r) \cup \mathcal{B}^{(\mathrm{II})}(r) \cup \mathcal{B}^{(\mathrm{III})}(r)$, and thus for an indexed family of events $\mathcal{A}_r$ to be chosen later, we have

$$\varepsilon_n^{-2} \, \mathbb{E}_{f \sim \Pi_n(\cdot|\boldsymbol{x},\boldsymbol{y})} \|f - f_n\|_{L^2(p_0)}^2 \lesssim \int_0^\infty r \, \mathbb{P}_{f \sim \Pi_n(\cdot|\boldsymbol{x},\boldsymbol{y})}\left(\mathcal{B}^{(\mathrm{I})}(r)\right) \mathrm{d}r + \int_0^\infty r \mathbb{1}_{\mathcal{A}_r^c} \, \mathrm{d}r \tag{69}$$

$$+ \int_0^\infty r \mathbb{1}_{\mathcal{A}_r} \, \mathbb{P}_{f \sim \Pi_n(\cdot|\boldsymbol{x},\boldsymbol{y})}\left(\mathcal{B}^{(\mathrm{II})}(r)\right) \mathrm{d}r + \int_0^\infty r \mathbb{1}_{\mathcal{A}_r} \, \mathbb{P}_{f \sim \Pi_n(\cdot|\boldsymbol{x},\boldsymbol{y})}\left(\mathcal{B}^{(\mathrm{III})}(r)\right) \mathrm{d}r. \tag{70}$$

For the first term, by Lemma 22 applied conditionally on $\boldsymbol{x}$, for which we got a bound on the integrated empirical $L^2$-norm uniformly on the design points, we have

$$\mathbb{E}_{\boldsymbol{x},\boldsymbol{y}} \int_0^\infty r \, \mathbb{P}_{f \sim \Pi_n(\cdot|\boldsymbol{x},\boldsymbol{y})}\left(\mathcal{B}^{(\mathrm{I})}(r)\right) \mathrm{d}r \lesssim \varepsilon_n^{-2} \, \mathbb{E}_{\boldsymbol{x},\boldsymbol{y}} \, \mathbb{E}_{f \sim \Pi_n(\cdot|\boldsymbol{x},\boldsymbol{y})} \|f - f_0\|_n^2 \lesssim \varepsilon_n^{-2} \varepsilon_n^2 = 1. \tag{71}$$

Moreover, by Lemma 14 of van der Vaart and van Zanten [56] applied with $r$ in the notation of the reference being equal to $\sqrt{n}\varepsilon_n r^s$, for each $r > 0$, the event

$$\mathcal{A}_r(\boldsymbol{x}) = \left\{\boldsymbol{u} \in \mathbb{R}^n : \int \frac{p_{\boldsymbol{y}|\boldsymbol{x}}^{(f)}(\boldsymbol{u})}{p_{\boldsymbol{y}|\boldsymbol{x}}^{(f_0)}(\boldsymbol{u})} \, \mathrm{d}\Pi_n(f) \geq e^{-n\varepsilon_n^2 r^{2s}} \, \mathbb{P}_{f \sim \Pi_n}\left(\|f - f_0\|_{L^\infty(\mathcal{M})} < \varepsilon_n r^s\right)\right\} \tag{72}$$

$$\supseteq \left\{\boldsymbol{u} \in \mathbb{R}^n : \int \frac{p_{\boldsymbol{y}|\boldsymbol{x}}^{(f)}(\boldsymbol{u})}{p_{\boldsymbol{y}|\boldsymbol{x}}^{(f_0)}(\boldsymbol{u})} \, \mathrm{d}\Pi_n(f) \geq e^{-n\varepsilon_n^2 r^{2s}} \, \mathbb{P}_{f \sim \Pi_n}(\|f - f_0\|_n < \varepsilon_n r^s)\right\} \tag{73}$$

is such that

$$p_{\boldsymbol{y}|\boldsymbol{x}}^{(f_0)}[\mathcal{A}_r^c(\boldsymbol{x})] \leq e^{-n\varepsilon_n^2 r^{2s}/8}. \tag{74}$$

It should be noted that the $\sqrt{n}$ factor disappears because of the discrepancy between the empirical $L^2$ norm $\|\cdot\|_n$ and the Euclidean norm used in van der Vaart and van Zanten [56]. By Fubini's Theorem and since $n\varepsilon_n^2 \geq n^{\frac{d}{2\nu+d}} \geq 1$, the second term is bounded by

$$\mathbb{E}_{\boldsymbol{x},\boldsymbol{y}} \int_0^\infty r \mathbb{1}_{\mathcal{A}_r^c(\boldsymbol{x})} \, \mathrm{d}r = \int_0^\infty r \, \mathbb{E}_{\boldsymbol{x}} \big[ \mathbb{E}_{\boldsymbol{y}|\boldsymbol{x}} \big[ \mathbb{1}_{\mathcal{A}_r^c(\boldsymbol{x})} \big] \big] \, \mathrm{d}r \tag{75}$$

$$\leq \int_0^\infty r e^{-n\varepsilon_n^2 r^{2s}/8} \, \mathrm{d}r \tag{76}$$

$$\leq \int_0^\infty r e^{-r^{2s}/8} \, \mathrm{d}r = C_s < \infty. \tag{77}$$

It remains to bound the last two terms. By Bayes' Rule, we have the equality

$$\mathbb{P}_{f\sim\Pi_n(\cdot|\boldsymbol{x},\boldsymbol{y})} \Big( \|f\|_{\mathcal{CH}^\gamma(\mathcal{M})} > \tau\sqrt{n}\varepsilon_n r^s \Big) = \frac{\int_{\|f\|_{\mathcal{CH}^\gamma(\mathcal{M})} > \tau\sqrt{n}\varepsilon_n r^s} p_{\boldsymbol{y}|\boldsymbol{x}}^{(f)}(\boldsymbol{y}) \, \mathrm{d}\Pi_n(f)}{\int p_{\boldsymbol{y}|\boldsymbol{x}}^{(f)}(\boldsymbol{y}) \, \mathrm{d}\Pi_n(f)} \tag{78}$$

$$= \frac{\int_{\|f\|_{\mathcal{CH}^\gamma(\mathcal{M})} > \tau\sqrt{n}\varepsilon_n r^s} \frac{p_{\boldsymbol{y}|\boldsymbol{x}}^{(f)}(\boldsymbol{y})}{p_{\boldsymbol{y}|\boldsymbol{x}}^{(f_0)}(\boldsymbol{y})} \, \mathrm{d}\Pi_n(f)}{\int \frac{p_{\boldsymbol{y}|\boldsymbol{x}}^{(f)}(\boldsymbol{y})}{p_{\boldsymbol{y}|\boldsymbol{x}}^{(f_0)}(\boldsymbol{y})} \, \mathrm{d}\Pi_n(f)} \tag{79}$$

therefore for $\boldsymbol{y} \in \mathcal{A}_r(\boldsymbol{x})$, we have

$$\mathbb{P}_{f\sim\Pi_n(\cdot|\boldsymbol{x},\boldsymbol{y})} \Big( \|f\|_{\mathcal{CH}^\gamma(\mathcal{M})} > \tau\sqrt{n}\varepsilon_n r^s \Big) \tag{80}$$

$$\leq \frac{e^{n\varepsilon_n^2 r^{2s}}}{\mathbb{P}_{f\sim\Pi_n}\Big( \|f - f_0\|_{L^\infty(\mathcal{M})} < \varepsilon_n r^s \Big)} \int_{\|f\|_{\mathcal{CH}^\gamma(\mathcal{M})} > \tau\sqrt{n}\varepsilon_n r^s} \frac{p_{\boldsymbol{y}|\boldsymbol{x}}^{(f)}(\boldsymbol{y})}{p_{\boldsymbol{y}|\boldsymbol{x}}^{(f_0)}(\boldsymbol{y})} \, \mathrm{d}\Pi_n(f). \tag{81}$$

Hence taking expectations, using Tonelli's Theorem, and $\mathbb{E}_{\boldsymbol{x},\boldsymbol{y}} \frac{p_{\boldsymbol{y}|\boldsymbol{x}}^{(f)}(\boldsymbol{y})}{p_{\boldsymbol{y}|\boldsymbol{x}}^{(f_0)}(\boldsymbol{y})} = 1$ gives

$$\mathbb{E}_{\boldsymbol{x},\boldsymbol{y}} \Big[ \mathbb{1}_{\mathcal{A}_r(\boldsymbol{x})} \mathbb{P}_{f\sim\Pi_n(\cdot|\boldsymbol{x},\boldsymbol{y})} \Big( \|f\|_{\mathcal{CH}^\gamma(\mathcal{M})} > \tau\sqrt{n}\varepsilon_n r^s \Big) \Big] \tag{82}$$

$$\leq \frac{e^{n\varepsilon_n^2 r^{2s}}}{\mathbb{P}_{f\sim\Pi_n}\Big( \|f - f_0\|_{L^\infty(\mathcal{M})} < \varepsilon_n r^s \Big)} \mathbb{E}_{\boldsymbol{x},\boldsymbol{y}} \int_{\|f\|_{\mathcal{CH}^\gamma(\mathcal{M})} > \tau\sqrt{n}\varepsilon_n r^s} \frac{p_{\boldsymbol{y}|\boldsymbol{x}}^{(f)}(\boldsymbol{y})}{p_{\boldsymbol{y}|\boldsymbol{x}}^{(f_0)}(\boldsymbol{y})} \, \mathrm{d}\Pi_n(f) \tag{83}$$

$$= \frac{e^{n\varepsilon_n^2 r^{2s}}}{\mathbb{P}_{f\sim\Pi_n}\Big( \|f - f_0\|_{L^\infty(\mathcal{M})} < \varepsilon_n r^s \Big)} \int_{\|f\|_{\mathcal{CH}^\gamma(\mathcal{M})} > \tau\sqrt{n}\varepsilon_n r^s} \mathbb{E}_{\boldsymbol{x},\boldsymbol{y}} \frac{p_{\boldsymbol{y}|\boldsymbol{x}}^{(f)}(\boldsymbol{y})}{p_{\boldsymbol{y}|\boldsymbol{x}}^{(f_0)}(\boldsymbol{y})} \, \mathrm{d}\Pi_n(f) \tag{84}$$

$$= \frac{e^{n\varepsilon_n^2 r^{2s}}}{\mathbb{P}_{f\sim\Pi_n}\Big( \|f - f_0\|_{L^\infty(\mathcal{M})} < \varepsilon_n r^s \Big)} \mathbb{P}_{f\sim\Pi_n}\Big( \|f\|_{\mathcal{CH}^\gamma(\mathcal{M})} > \tau\sqrt{n}\varepsilon_n r^s \Big). \tag{85}$$

Therefore, the third term can be bounded as

$$\mathbb{E}_{\boldsymbol{x},\boldsymbol{y}} \int_0^\infty r \mathbb{1}_{\mathcal{A}_r} \mathbb{P}_{f\sim\Pi_n(\cdot|\boldsymbol{x},\boldsymbol{y})} \Big( \mathcal{B}^{(\mathrm{II})}(r) \Big) \, \mathrm{d}r \tag{86}$$

$$\leq \int_0^\infty r \frac{e^{n\varepsilon_n^2 r^{2s}}}{\mathbb{P}_{f\sim\Pi_n}\Big( \|f - f_0\|_{L^\infty(\mathcal{M})} < \varepsilon_n r^s \Big)} \mathbb{P}_{f\sim\Pi_n}\Big( \|f\|_{\mathcal{CH}^\gamma(\mathcal{M})} > \tau\sqrt{n}\varepsilon_n r^s \Big) \, \mathrm{d}r. \tag{87}$$

Now, using Lemma 31, for a possibly small constant $c > 0$ independent of $n$, we have

$$\mathbb{P}_{f\sim\Pi_n(\cdot|\boldsymbol{x},\boldsymbol{y})} \Big( \|f\|_{\mathcal{CH}^\gamma(\mathcal{M})} > \tau\sqrt{n}\varepsilon_n r^s \Big) \leq e^{-c\tau^2 n\varepsilon_n^2 r^{2s}}. \tag{88}$$

Moreover, by using the bound on the concentration function in Theorem 21 and Ghosal and van der Vaart [20], Proposition 11.19, we can assume that

$$\mathbb{P}_{f\sim\Pi_n} \Big[ \|f - f_0\|_{L^\infty(\mathcal{M})} < \varepsilon_n r^s \Big] \geq e^{-c^{-1} n\varepsilon_n^2 r^{2s}}. \tag{89}$$

Therefore, the third term is bounded by

$$\mathbb{E}_{\boldsymbol{x},\boldsymbol{y}}\int_0^\infty r\mathbb{1}_{\mathcal{A}_r(\boldsymbol{x})}\,\mathbb{P}_{f\sim\Pi_n(\cdot|\boldsymbol{x},\boldsymbol{y})}\Big(\mathcal{B}^{(\mathrm{II})}(r)\Big)\,\mathrm{d}r \le \int_0^\infty re^{-n(c\tau^2-1)\varepsilon_n^2 r^{2s}}e^{c^{-1}n\varepsilon_n^2 r^{2s}}\,\mathrm{d}r \tag{90}$$

$$\le \int_0^\infty re^{-r^{2s}}\,\mathrm{d}r < \infty \tag{91}$$

if $\tau^2 c > 2 + c^{-1}$. It remains to bound the last term. We have, by the same arguments as above, that

$$\mathbb{E}_{\boldsymbol{x},\boldsymbol{y}}\int_0^\infty r\mathbb{1}_{\mathcal{A}_r(\boldsymbol{x})}\,\mathbb{P}_{f\sim\Pi_n(\cdot|\boldsymbol{x},\boldsymbol{y})}\Big(\mathcal{B}^{(\mathrm{III})}(r)\Big)\,\mathrm{d}r \tag{92}$$

$$= \mathbb{E}_{\boldsymbol{x},\boldsymbol{y}}\int_0^\infty r\mathbb{1}_{\mathcal{A}_r(\boldsymbol{x})} \tag{93}$$

$$\times\,\mathbb{P}_{f\sim\Pi_n(\cdot|\boldsymbol{x},\boldsymbol{y})}\Big(\|f\|_{\mathcal{CH}^\gamma(\mathcal{M})}\le\tau\sqrt{n}\varepsilon_n r^s, 2\|f-f_n\|_n\le\varepsilon_n r\le\|f-f_n\|_{L^2(p_0)}\Big)\,\mathrm{d}r \tag{94}$$

$$\le \int_0^\infty r\frac{e^{n\varepsilon_n^2 r^{2s}}}{\mathbb{P}_{f\sim\Pi_n}\Big(\|f-f_0\|_{L^\infty(\mathcal{M})}<\varepsilon_n r^s\Big)} \tag{95}$$

$$\times\,\mathbb{E}_{\boldsymbol{x}}\,\mathbb{P}_{f\sim\Pi_n}\Big(\|f\|_{\mathcal{CH}^\gamma(\mathcal{M})}\le\tau\sqrt{n}\varepsilon_n r^s, 2\|f-f_n\|_n\le\varepsilon_n r\le\|f-f_n\|_{L^2(p_0)}\Big)\,\mathrm{d}r \tag{96}$$

$$\le \int_0^\infty re^{(c+1)n\varepsilon_n^2 r^{2s}} \tag{97}$$

$$\times\int_{\|f\|_{\mathcal{CH}^\gamma(\mathcal{M})}\le\tau\sqrt{n}\varepsilon_n r^s,\varepsilon_n r\le\|f-f_n\|_{L^2(p_0)}}\mathbb{E}_{\boldsymbol{x}}\,\mathbb{1}_{\|f-f_n\|_{L^2(p_0)}\ge 2\|f-f_n\|_n}\,\mathrm{d}\Pi_n(f)\,\mathrm{d}r. \tag{98}$$

As the squared empirical $L^2$-norm is a Monte Carlo approximation of the true $L^2$-norm, the probability in the integrand can be controlled via a concentration inequality. As in van der Vaart and van Zanten [56], we use Bernstein's inequality [58, Lemma 2.2.9]. For a collection $Y_1,\ldots Y_n$ of random variables such that $\mathbb{E}\,Y_i = 0$ and $Y_i \in [-M, M]$ almost surely for some constant $M > 0$, this inequality asserts

$$\mathbb{P}(|Y_1+\ldots+Y_n| > x) \le 2\exp\Big(-\frac{1}{2}\frac{x^2}{v + Mx/3}\Big) \tag{99}$$

where $v \ge \mathrm{Var}(Y_1+\ldots+Y_n)$. We put $Y_i = \frac{1}{n}(f(x_i)-f_n(x_i))^2 - \frac{1}{n}\|f-f_n\|_{L^2(p_0)}^2$ where $x_i$ are IID with $x_i \sim p_0$. It is easy to check that $\mathbb{E}\,Y_i = 0$ and $Y_i \in [-M, M]$ for $M = \frac{1}{n}\|f-f_n\|_{L^\infty(\mathcal{M})}$. Furthermore, $v = \frac{1}{n}\|f-f_n\|_{L^\infty(\mathcal{M})}^2\|f-f_n\|_{L^2(\mathcal{M})}^2$ upper-bounds the variance of the respective sum, since

$$\mathrm{Var}(Y_1+\ldots+Y_n) = \frac{1}{n}\mathrm{Var}_{x\sim p_0}(f(x)-f_n(x))^2 \le \frac{1}{n}\mathbb{E}_{x\sim p_0}(f(x)-f_n(x))^4 \tag{100}$$

$$\le \frac{\|f-f_n\|_{L^\infty(\mathcal{M})}^2}{n}\mathbb{E}_{x\sim p_0}(f(x)-f_n(x))^2 = v. \tag{101}$$

Using Bernstein's inequality with $Y_i$, $M$, $v$ as above and with $x = \frac{3}{4}\|f-f_n\|_{L^2(p_0)}^2$, we have

$$\mathbb{E}_{\boldsymbol{x}}\,\mathbb{1}_{\|f-f_n\|_{L^2(p_0)}\ge 2\|f-f_n\|_n} = \mathbb{E}_{\boldsymbol{x}}\,\mathbb{1}_{\|f-f_n\|_{L^2(p_0)}^2\ge 4\|f-f_n\|_n^2} \tag{102}$$

$$= \mathbb{E}_{\boldsymbol{x}}\,\mathbb{1}_{\|f-f_n\|_n^2-\|f-f_n\|_{L^2(p_0)}^2\le-\frac{3}{4}\|f-f_n\|_{L^2(p_0)}^2} \tag{103}$$

$$\lesssim \exp\Big(-\frac{1}{2}\frac{x^2}{v+Mx/3}\Big) \tag{104}$$

$$= \exp\Big(-\frac{\frac{9}{32}\|f-f_n\|_{L^2(p_0)}^4}{v+\frac{1}{n}\|f-f_n\|_{L^\infty(\mathcal{M})}^2\frac{3}{4}\|f-f_n\|_{L^2(p_0)}^2/3}\Big) \tag{105}$$

$$= \exp\Big(-\frac{9n}{32}\frac{4}{5}\frac{\|f-f_n\|_{L^2(p_0)}^2}{\|f-f_n\|_{L^\infty(\mathcal{M})}^2}\Big) \tag{106}$$

$$= \exp\Big(-\frac{9n}{40}\frac{\|f-f_n\|_{L^2(p_0)}^2}{\|f-f_n\|_{L^\infty(\mathcal{M})}^2}\Big). \tag{107}$$

Moreover, by Lemma 19, since $\gamma \notin \mathbb{Z}_{>0}$, we have

$$\|f - f_n\|_{L^\infty(\mathcal{M})} \lesssim \|f - f_n\|_{\mathcal{CH}^\gamma(\mathcal{M})}^{\frac{d}{2\gamma+d}} \|f - f_n\|_{L^2(p_0)}^{\frac{2\gamma}{2\gamma+d}}. \tag{108}$$

Using the Sobolev Embedding Theorem, given in De Vito et al. [16], Theorem 4, $\|f - f_n\|_{\mathcal{CH}^\gamma(\mathcal{M})} \lesssim \|f_n\|_{\mathbb{H}} + \|f\|_{\mathcal{CH}^\gamma(\mathcal{M})} \lesssim \tau\sqrt{n}\varepsilon_n r^s$ whenever $\|f\|_{\mathcal{CH}^\gamma(\mathcal{M})} \leq \tau\sqrt{n}\varepsilon_n r^s$. Therefore, for a constant $c > 0$, when $\|f\|_{\mathcal{CH}^\gamma(\mathcal{M})} \leq \tau\sqrt{n}\varepsilon_n r^s$ and $\varepsilon_n r \leq \|f - f_n\|_{L^2(p_0)}$, we have that

$$\mathbb{E}_{\boldsymbol{x}} \mathbb{1}_{\|f-f_n\|_{L^2(\mathcal{M})} \geq 2\|f-f_n\|_n} \lesssim \exp\left(-cn\frac{\|f - f_n\|_{L^2(p_0)}^2}{\|f - f_n\|_{\mathcal{CH}^\gamma(\mathcal{M})}^{\frac{2d}{2\gamma+d}} \|f - f_n\|_{L^2(p_0)}^{\frac{4\gamma}{2\gamma+d}}}\right) \tag{109}$$

$$\leq e^{-c\tau^{-\frac{2d}{2\gamma+d}} n^{\frac{2\gamma}{2\gamma+d}} r^{\frac{2d}{2\gamma+d}(1-s)}}. \tag{110}$$

Hence, we can bound the last term as

$$\mathbb{E}_{\boldsymbol{x},\boldsymbol{y}} \int_0^\infty r\mathbb{1}_{\mathcal{A}_r} \mathbb{P}_{f\sim\Pi_n(\cdot|\boldsymbol{x},\boldsymbol{y})}\left(\mathcal{B}^{(\mathrm{III})}(r)\right) \mathrm{d}r \tag{111}$$

$$\lesssim \int_0^\infty r e^{(c+1)n\varepsilon_n^2 r^{2s}} e^{-c\tau^{-\frac{2d}{2\gamma+d}} n^{\frac{2\gamma}{2\gamma+d}} r^{\frac{2d}{2\gamma+d}(1-s)}} \, \mathrm{d}r. \tag{112}$$

We have $n^{\frac{2\gamma}{2\gamma+d}} = n\left(n^{-\frac{d/2}{2\gamma+d}}\right)^2$. Since $\varepsilon_n \lesssim n^{-\frac{\min(\nu,\beta)}{2\nu+d}}$ and $\min(\nu,\beta) > d/2$, we have $n\varepsilon_n^2 \lesssim n^{\frac{2\gamma}{2\gamma+d}}$ for some $\gamma \in (d/2, \nu)$. Moreover, for this choice of $\gamma$ and $s$ small enough we have $\frac{2d}{2\gamma+d}(1 - s) \geq 2s$, which proves that for some constants $C, C', C'' > 0$ the fourth term is bounded by

$$C \int_0^\infty r e^{-C'r^{C''}} \, \mathrm{d}r < \infty. \tag{113}$$

This concludes the proof. $\qquad\square$

**Remark 23.** *Following the proof, it is easy to see that $\|f - f_0\|_{L^2(p_0)}^2$ on the left-hand side of Equations (9), (10) and (12) can be replaced with $\|f - f_0\|_{L^2(p_0)}^q$ for any $q > 1$, changing the exponent $-\frac{2\min(\beta,\nu)}{2\nu+d}$ on the right-hand side to $-\frac{q\min(\beta,\nu)}{2\nu+d}$.*

## C Characterizing Reproducing Kernel Hilbert Spaces of Matérn Kernels

We start by describing the reproducing kernel Hilbert spaces (RKHSs) of the (truncated) intrinsic Matérn processes, proving that they coincide with (certain subspaces of) the manifold Sobolev spaces. We follow the ideas of Borovitskiy et al. [11], where the same was shown somewhat implicitly. We consider the more-involved case of the extrinsic Matérn processes immediately after.

The next lemma describes the RKHS of the intrinsic Matérn processes, including truncated variants. This result is easy to obtain since we have defined them in terms of the Karhunen–Loève expansions.

**Lemma 24.** *Let $\mathbb{H}_J$ be the RKHS of the intrinsic Matérn Gaussian process with smoothness parameter $\nu$ truncated at the level $J \in \mathbb{Z}_{>0} \cup \{\infty\}$, and let $\{f_j\}_{j=0}^\infty$ be the orthonormal basis of Laplace–Beltrami eigenfunctions. The space $\mathbb{H}_J$ is norm-equivalent—with constants depending only on $\nu, \kappa$ and $\sigma_f^2$—to the set of functions $f = \sum_{j=1}^J b_j f_j$ with $b_j \in \mathbb{R}$, equipped with the inner product*

$$\left\langle \sum_{j=1}^J b_j f_j, \sum_{j=1}^J b_j' f_j \right\rangle_{\mathbb{H}_J} = \sum_{j=1}^J (1 + \lambda_j)^{\nu+d/2} b_j b_j'. \tag{114}$$

*In particular, $\mathbb{H}_J \subset H^{\nu+d/2}(\mathcal{M})$ for all $J$, and for every $h \in \mathbb{H}_J$ we have $\|h\|_{\mathbb{H}_J} = \|h\|_{H^{\nu+d/2}(\mathcal{M})}$.*

*Proof.* By direct computation, the covariance $k$ of the (truncated) intrinsic Gaussian process is

$$k(x, x') = \frac{\sigma_f^2}{C_{\nu,\kappa}} \sum_{j=1}^J \left(\frac{2\nu}{\kappa^2} + \lambda_j\right)^{-(\nu+d/2)} f_j(x) f_j(x'). \tag{115}$$

Hence, the covariance operator $K : L^2(\mathcal{M}) \to L^2(\mathcal{M})$ defined by

$$(Kf)(x) = \int_{\mathcal{M}} k(x, x') f(x') \, \mathrm{d}x' \tag{116}$$

is diagonal in the basis $\{f_j\}_{j=1}^J$, with $Kf_j = \frac{\sigma_f^2}{C_{\nu,\kappa}} \left(\frac{2\nu}{\kappa^2} + \lambda_j\right)^{-(\nu+d/2)} f_j$. Then, Kanagawa et al. [28], Theorem 4.2 implies that $\mathbb{H}_J$ consists of functions of form $f = \sum_{j=1}^J a_j f_j$ satisfying

$$\|f\|_{\mathbb{H}_J}^2 = \frac{\sigma_f^2}{C_{\nu,\kappa}} \sum_{j=1}^J \left(\frac{2\nu}{\kappa^2} + \lambda_j\right)^{\nu+d/2} |a_j|^2 < \infty. \tag{117}$$

Using the elementary inequality $\min\left(\frac{2\nu}{\kappa^2}, 1\right) \leq \frac{\frac{2\nu}{\kappa^2} + \lambda}{1+\lambda} \leq \max\left(\frac{2\nu}{\kappa^2}, 1\right)$, we find that this space is norm-equivalent to the space $H_J^{\nu+d/2}(\mathcal{M})$ of functions $f = \sum_{j=1}^J a_j f_j$ satisfying

$$\|f\|_{H_J^{\nu+d/2}(\mathcal{M})}^2 = \sum_{j=1}^J (1 + \lambda_j)^{\nu+d/2} |a_j|^2 < \infty \tag{118}$$

where the comparison constants $\sqrt{\frac{\sigma_f^2}{C_{\nu,\kappa}} \min\left(1, \frac{2\nu}{\kappa^2}\right)}$ and $\sqrt{\frac{\sigma_f^2}{C_{\nu,\kappa}} \max\left(1, \frac{2\nu}{\kappa^2}\right)}$ only depend on the parameters $\nu, \kappa, \sigma_f^2$. $\qquad\square$

The next two theorems will be useful to characterize the RKHS of the extrinsic Matérn process on $\mathcal{M}$. We start by a lemma relating the RKHS of the restriction of a Gaussian process to the original one.

**Lemma 25.** *Assume that $k$ is a kernel on $\mathbb{R}^d$, $f \sim \mathrm{GP}(0, k)$ with almost surely continuous sample paths and $\widetilde{\mathbb{H}}$ is the RKHS of $k$. If $\mathcal{M} \subseteq \mathbb{R}^d$ is a submanifold, then the RKHS $\mathbb{H}$ corresponding to the restricted process $f_{|\mathcal{M}}$ is the set of all restrictions $g_{|\mathcal{M}}$ of functions $g \in \widetilde{\mathbb{H}}$ equipped with the norm*

$$\|h\|_{\mathbb{H}} = \inf_{g \in \widetilde{\mathbb{H}}, \, g_{|\mathcal{M}} = h} \|g\|_{\widetilde{\mathbb{H}}}. \tag{119}$$

*Moreover there always exists an element $g \in \widetilde{\mathbb{H}}$ such that $g_{|\mathcal{M}} = f$ and $\|g\|_{\widetilde{\mathbb{H}}} = \|f\|_{\mathbb{H}}$.*

*Proof.* Lemma 5.1 in Yang and Dunson [63]. $\qquad\square$

The last result will be used to characterize the RKHS of the extrinsic Matérn Gaussian processes using trace and extension operators. The second ingredient for this is the following.

**Theorem 26.** *If $s > \frac{D-d}{2}$ then the restriction operator extends to a bounded linear map $\mathrm{Tr} : H^s(\mathbb{R}^D) \to H^{s-\frac{D-d}{2}}(\mathcal{M})$. Moreover, for every $u > 0$ there exists a bounded right inverse $\mathrm{Ex} : H^u(\mathcal{M}) \to H^{u+\frac{D-d}{2}}(\mathbb{R}^D)$ such that $\mathrm{Tr} \circ \mathrm{Ex} = \mathrm{id}_{H^u(\mathcal{M})}$ where $\mathrm{Tr}$ corresponds to $s = u + \frac{D-d}{2}$.*

*Proof.* Theorem 4.10 in Große and Schneider [24]. $\qquad\square$

The last two results allow us to characterize the RKHS of the extrinsic Matérn process on $\mathcal{M}$.

**Proposition 27.** *The RKHS $\mathbb{H}$ of a restricted extrinsic Matérn process $f$ with smoothness parameter $\nu$ on $\mathcal{M}$ is norm-equivalent to the Sobolev space $H^{\nu+d/2}(\mathcal{M})$.*

*Proof.* Using Lemma 25, the RKHS $\mathbb{H}$ can be characterized as the set of functions $f : \mathcal{M} \to \mathbb{R}$ that are the restrictions of some $g \in \widetilde{\mathbb{H}}$, where $\widetilde{\mathbb{H}}$ is the RKHS of the ambient Matérn process $\tilde{f}$, with

$$\|f\|_{\mathbb{H}} = \inf_{g \in \widetilde{\mathbb{H}}, \, g_{|\mathcal{M}} = f} \|g\|_{\widetilde{\mathbb{H}}}. \tag{120}$$

Since $\widetilde{\mathbb{H}}$ is norm-equivalent to the Sobolev space[7] $H^{\nu+D/2}(\mathbb{R}^D)$—see for instance Kanagawa et al. [28]—by Theorem 26, for every $f \in \mathbb{H}$ we have

$$\|f\|_{\mathbb{H}} \lesssim \|\mathrm{Ex}(f)\|_{H^{\nu+D/2}(\mathbb{R}^D)} \lesssim \|f\|_{H^{\nu+D/2-\frac{D-d}{2}}(\mathcal{M})} = \|f\|_{H^{\nu+d/2}(\mathcal{M})}. \tag{121}$$

Similarly, for every $g \in \widetilde{\mathbb{H}}$ with $g_{|\mathcal{M}} = f$, we have

$$\|f\|_{H^{\nu+d/2}(\mathcal{M})} = \|g_{|\mathcal{M}}\|_{H^{\nu+d/2}(\mathcal{M})} \lesssim \|g\|_{H^{\nu+D/2}(\mathbb{R}^D)} \lesssim \|g\|_{\widetilde{\mathbb{H}}}. \tag{122}$$

Hence, taking the infimum, we obtain

$$\|f\|_{H^{\nu+d/2}(\mathcal{M})} \lesssim \inf_{g \in \widetilde{\mathbb{H}},\ g_{|\mathcal{M}}=f} \|g\|_{\widetilde{\mathbb{H}}} = \|f\|_{\mathbb{H}}. \tag{123}$$

The claim follows. $\qquad\square$

## D  Concentration and Sample Path Regularity

In this section, we prove that intrinsic, truncated intrinsic and extrinsic Matérn processes are sub-Gaussian, uniformly with respect to the truncation parameter in the case of the truncated intrinsic Matérn process, and live in Hölder spaces with appropriate exponents. On our way to proving this, we characterize sample path regularity of (truncated) intrinsic Matérn processes, which is of independent interest. A simple way to do this would be to build on the results of Appendix C, using Driscoll's Theorem—given in Kanagawa et al. [28], Theorem 4.9—and the Sobolev Embedding Theorem—De Vito et al. [16], Theorem 4—but that would only give us that the sample paths are almost surely in $\mathcal{CH}^\gamma(\mathcal{M})$ for every $0 < \gamma < \nu - d/2, \gamma \notin \mathbb{N}$, whereas here we improve the range of index to $\gamma < \nu$.

Kolmogorov's continuity criterion is a standard tool to show that a given stochastic process has a Hölder continuous version: we re-prove it here because we will need a form of the result which gives explicit control of the Hölder norms, which is not usually included in the respective statement.

**Lemma 28** (Kolmogorov's continuity criterion). *If $g \sim \Pi$ is a zero-mean Gaussian process on $[0,1]^d$ with*

$$\mathbb{E}\,|g(x) - g(y)|^2 \leq C\|x - y\|_{\mathbb{R}^d}^{2\rho} \tag{124}$$

*for some $0 < \rho \leq 1$ and $C > 0$, then there exists a version of $g$ with samples paths in $\mathcal{CH}^\alpha([0,1]^d)$ for every $0 < \alpha < \rho$. Moreover for every $\alpha < \rho$ this version satisfies $\mathbb{E}\,\|g\|_{\mathcal{CH}^\alpha([0,1]^d)}^2 \leq C'$ where $C' < +\infty$ depends only on $C, \rho, \alpha$ and $d$.*

*Proof.* Take $x, y \in [0,1]^d, M > 0$ and $q \in \mathbb{N}$. Since the random variable $g(x) - g(y)$ is Gaussian we have

$$\mathbb{E}\,|g(x) - g(y)|^{2q} = \frac{(2q)!}{2^q q!}\left(\mathbb{E}\,|g(x) - g(y)|^2\right)^q \leq C_q\|x - y\|_{\mathbb{R}^d}^{2\rho q} \tag{125}$$

where we have defined $C_q = C^q \frac{(2q)!}{2^q q!}$. The reason for considering the the $2q$th power will become clear later in the proof. By Markov's inequality, for every $x, y \in [0,1]^d$ we have

$$\mathbb{P}(|g(x) - g(y)| > u) \leq C_q u^{-2q}\|x - y\|_{\mathbb{R}^d}^{2q\rho}. \tag{126}$$

Now, take $X = \cup_{k\geq 0} X_k, X_k = 2^{-k}\mathbb{Z}^d \cap [0,1]^d$. Then, the previous inequality applied to any adjacent $x, y \in X_k$, where we see $X_k$ as a graph where two vertices are connected if they differ by $2^{-k}$, and $u = M2^{-k\alpha}$, implies

$$\mathbb{P}(|g(x) - g(y)| > M2^{-k\alpha}) \leq C_q M^{-2q} 2^{-2kq(\rho-\alpha)}. \tag{127}$$

---

[7]Note that this norm-equivalence is the only property of the Gaussian process we use in the proofs. Any other Gaussian process satisfying this, including ones different from the Matérn processes of Borovitskiy et al. [11], would also work. This is of potential interest since other Euclidean kernels, such as Wendland kernels [61], are known to possess RKHSs which are norm-equivalent to those of the Matérn kernel.

Denote $K = \sup_{x,y \in X \text{ adjacent}} \frac{|g(x) - g(y)|}{\|x-y\|^\alpha}$. Summing over $k \geq 1$ and adjacent points in $X$—and there are at most $C2^{k(d+1)}$ of them where $C > 0$ is an absolute constant—gives us for $q > \frac{d+1}{2(\rho-\alpha)}$ that

$$\mathbb{P}(K > M) \leq \sum_{k \geq 0} \sum_{x,y \in X \text{ adjacent}} \mathbb{P}\big(|g(x) - g(y)| > M 2^{-k\alpha}\big) \tag{128}$$

$$\leq C \sum_{k \geq 0} 2^{k(d+1)} C_q M^{-2q} 2^{-2kq(\rho-\alpha)} = \frac{CC_q}{1 - 2^{2q(\rho-\alpha)-d-1}} M^{-2q}. \tag{129}$$

In particular, for all $q > \max\left(1, \frac{d+1}{2(\rho-\alpha)}\right)$, we have

$$\mathbb{E}(K^2) = 2 \int_0^\infty M \, \mathbb{P}(K > M) \, \mathrm{d}M = 2 \int_0^1 M \, \mathbb{P}(K > M) \, \mathrm{d}M + 2 \int_1^\infty M \, \mathbb{P}(k > M) \, \mathrm{d}M \tag{130}$$

$$\leq 2 + 2 \int_1^\infty M \, \mathbb{P}(K > M) \, \mathrm{d}M \leq C_{C,\alpha,\rho,d} \tag{131}$$

for some constant $C_{C,\alpha,\rho,d} < +\infty$. This means that $K$ is finite almost surely. Since $X$ is dense in $[0,1]^d$ and $g$ is almost surely uniformly continuous on $X$ by the classical version of the Kolmogorov's continuity criterion—see for instance Lototsky and Rozovsky [34]—$g$ admits a unique continuous extension to $[0,1]^d$ on a probability one event $\mathcal{A}$. Let us define

$$\overline{g}(x) = \begin{cases} \lim_{y \to x, y \in X} g(y) & \text{on } \mathcal{A}, \\ 0 & \text{otherwise}, \end{cases} \tag{132}$$

for $x \in [0,1]^d$. For any $x, y \in [0,1]^d$ and $x_n \to x, y_n \to y, x_n, y_n \in X$ we have on $\mathcal{A}$

$$|\overline{g}(x) - \overline{g}(y)| \leq \liminf_{n \to \infty} (|\overline{g}(x) - \overline{g}(x_n)| + |\overline{g}(x_n) - \overline{g}(y_n)| + |\overline{g}(y_n) - \overline{g}(y)|) \tag{133}$$

$$\leq \liminf_{n \to \infty} (|\overline{g}(x) - \overline{g}(x_n)| + K\|x_n - y_n\|_{\mathbb{R}^d}^\alpha + |\overline{g}(y_n) - \overline{g}(y)|) \tag{134}$$

$$= K\|x - y\|_{\mathbb{R}^d}^\alpha. \tag{135}$$

On the complement of $\mathcal{A}$, we have $\overline{g} = 0$. Hence, $\overline{g}$ is $\alpha$-Hölder continuous on $[0,1]^d$ with the same (random) constant $K$, since

$$\mathbb{E}\|\overline{g}\|_{\mathcal{CH}^\alpha([0,1]^d)}^2 = \mathbb{E}\left(\sup_{x,y \in X} \frac{|\overline{g}(x) - \overline{g}(y)|}{\|x-y\|^\alpha}\right)^2 \leq \mathbb{E}\, K^2 \leq C_{C,\alpha,\rho,d} < \infty. \tag{136}$$

Since $\overline{g}$ is a version of $g$, the claim follows. $\qquad\square$

The next lemma applies our version of Kolmogorov's criterion, Lemma 28, to the intrinsic Matérn processes on $\mathcal{M}$ by considering charts. This allows us to show that the sample paths are almost surely in $\mathcal{CH}^\gamma(\mathcal{M})$ for every $\gamma < \nu$, which is both used in our arguments and also of independent interest. For the claims in Appendix B, we need to ensure that this property holds somewhat uniformly with respect to the truncation parameter, which is why we tracked the constants in our proof of Kolmogorov's criterion. As we will see, the main difficulty in the proof of the next result will be to tackle the case of regularity strictly larger than 1.

**Lemma 29.** *Let $f \sim \Pi_n$ be an intrinsic Matérn process with smoothness parameter $\nu > 0$ truncated at $J_n \in \mathbb{N} \cup \{\infty\}$. Then for every $\gamma < \nu$ we have*

$$\sup_n \mathbb{E}_{f \sim \Pi_n}\left[\|f\|_{\mathcal{CH}^\gamma(\mathcal{M})}^2\right] < \infty. \tag{137}$$

*Proof.* We start with the case $\nu \leq 1$. Take $1 \leq l \leq L$ and define $h_l = (\chi_l f) \circ \phi_l^{-1}$. Then $h_l$ is a Gaussian process with covariance kernel $\widetilde{K}_l$ given by

$$\widetilde{K}_l(x,y) = (\chi_l \circ \phi_l^{-1})(x) K(\phi_l^{-1}(x), \phi_l^{-1}(y))(\chi_l \circ \phi_l^{-1})(y) \tag{138}$$

where $x, y \in \mathcal{V}_l$ and $K(x, y) = \mathrm{Cov}(f(x), f(y))$ is the covariance kernel of $f$. Let $\widetilde{\mathbb{H}}_l$ be the RKHS induced by $\widetilde{K}_l$. We seek to apply Lemma 28 to $h_l$. For all $x, y \in \mathcal{V}_l$, where we recall that we can assume that $\mathcal{V}_l = (0, 1)^d$,, we have

$$\mathbb{E}_{f \sim \Pi_n} |h_l(x) - h_l(y)|^2 = \widetilde{K}_l(x, x) + \widetilde{K}_l(y, y) - 2\widetilde{K}_l(x, y) \tag{139}$$

$$= \left\| \widetilde{K}_l(x, \cdot) - \widetilde{K}_l(y, \cdot) \right\|_{\widetilde{\mathbb{H}}_l}^2 \tag{140}$$

$$= \sup_{\|\varphi\|_{\widetilde{\mathbb{H}}_l} = 1} \left| \left\langle \widetilde{K}_l(x, \cdot) - \widetilde{K}_l(y, \cdot), \varphi \right\rangle_{\widetilde{\mathbb{H}}_l} \right|^2 \tag{141}$$

$$= \sup_{\|\varphi\|_{\widetilde{\mathbb{H}}_l} = 1} |\varphi(x) - \varphi(y)|^2 \tag{142}$$

$$\leq \sup_{\|\varphi\|_{\widetilde{\mathbb{H}}_l} = 1} \|\varphi\|_{\mathcal{CH}^\nu(\mathcal{V}_l)}^2 \|x - y\|_{\mathbb{R}^d}^{2\nu} \tag{143}$$

where $\|\varphi\|_{\mathcal{CH}^\nu(\mathcal{V}_l)}^2$ can potentially be infinite or unbounded: we will show this is not the case. To do this it suffices to show that we have a continuous embedding $\widetilde{\mathbb{H}}_l \hookrightarrow \mathcal{CH}^\nu(\mathcal{V}_l)$, that is $\|\cdot\|_{\mathcal{CH}^\nu(\mathcal{V}_l)} \lesssim \|\cdot\|_{\widetilde{\mathbb{H}}_l}$. The RKHS $\widetilde{\mathbb{H}}_l$ is by definition the completion of

$$\left\{ \sum_{i=1}^p \alpha_i \widetilde{K}_l(x_i, \cdot) : p \geq 1, \alpha_i \in \mathbb{R}, x_i \in \mathcal{V}_l \right\} \tag{144}$$

$$= \left\{ \sum_{i=1}^p \alpha_i \big(\chi_l \circ \phi_l^{-1}\big)(x_i)\big(\chi_l \circ \phi_l^{-1}\big)(\cdot) K\big(\phi_l^{-1}(x_i), \phi_l^{-1}(\cdot)\big) : p \geq 1, \alpha_i \in \mathbb{R}, x_i \in \mathcal{V}_l \right\} \tag{145}$$

with respect to the topology induced by the RKHS norm

$$\left\| \sum_{i=1}^p \alpha_i \widetilde{K}_l(x_i, \cdot) \right\|_{\widetilde{\mathbb{H}}_l}^2 = \sum_{i,j=1}^p \alpha_i \alpha_j \big(\chi_l \circ \phi_l^{-1}\big)(x_i)\big(\chi_l \circ \phi_l^{-1}\big)(x_j) K\big(\phi_l^{-1}(x_i), \phi_l^{-1}(x_j)\big). \tag{146}$$

Denote the RKHS of $K$ by $\mathbb{H}$. By Theorem 16, and by the equality $\|\cdot\|_{\mathbb{H}} = \|\cdot\|_{H^{\nu+d/2}(\mathcal{M})}$ on $\mathbb{H}$ which follows by Lemma 24, we have

$$\left\| \sum_{i=1}^p \alpha_i \widetilde{K}_l(x_i, \cdot) \right\|_{H^{\nu+d/2}(\mathbb{R}^d)}^2 \tag{147}$$

$$= \left\| \sum_{i=1}^p \alpha_i \big(\chi_l \circ \phi_l^{-1}\big)(x_i)\big(\chi_l \circ \phi_l^{-1}\big)(\cdot) K\big(\phi_l^{-1}(x_i), \phi_l^{-1}(\cdot)\big) \right\|_{H^{\nu+d/2}(\mathbb{R}^d)}^2 \tag{148}$$

$$\lesssim \left\| \sum_{i=1}^p \alpha_i \big(\chi_l \circ \phi_l^{-1}\big)(x_i) K\big(\phi_l^{-1}(x_i), \cdot\big) \right\|_{H^{\nu+d/2}(\mathcal{M})}^2 \tag{149}$$

$$= \left\| \sum_{i=1}^p \alpha_i \big(\chi_l \circ \phi_l^{-1}\big)(x_i) K\big(\phi_l^{-1}(x_i), \cdot\big) \right\|_{\mathbb{H}}^2 \tag{150}$$

$$= \sum_{i,j=1}^p \alpha_i \alpha_j \big(\chi_l \circ \phi_l^{-1}\big)(x_i)\big(\chi_l \circ \phi_l^{-1}\big)(x_j) K\big(\phi_l^{-1}(x_i), \phi_l^{-1}(x_j)\big) \tag{151}$$

$$= \left\| \sum_{i=1}^p \alpha_i \widetilde{K}_l(x_i, \cdot) \right\|_{\widetilde{\mathbb{H}}_l}^2. \tag{152}$$

Therefore, we have a continuous embedding $\widetilde{\mathbb{H}}_l \hookrightarrow H^{\nu+d/2}(\mathbb{R}^d)$ with $\|\cdot\|_{H^{\nu+d/2}(\mathbb{R}^d)} \lesssim \|\cdot\|_{\widetilde{\mathbb{H}}_l}$ on $\widetilde{\mathbb{H}}_l$. By the Sobolev Embedding Theorem in $\mathbb{R}^d$—see for instance Triebel [50], Section 2.7.1,

Remark 2—we have $B_{2,2}^{\nu+d/2}(\mathbb{R}^d) = H^{\nu+d/2}(\mathbb{R}^d) \hookrightarrow \mathcal{CH}^\nu(\mathbb{R}^d)$, which implies $\widetilde{\mathbb{H}}_l \hookrightarrow \mathcal{CH}^\nu(\mathbb{R}^d)$ by composition. Thus, there exists a constant $C = C_\nu$ such that

$$\mathbb{E}_{f\sim\Pi_n}|h_l(x) - h_l(y)|^2 \leq C\|x-y\|_{\mathbb{R}^d}^{2\nu} \tag{153}$$

for $x,y \in \mathcal{V}_l$. Hence, by applying Lemma 28, there exists a version $\tilde{h}_l$ of $h_l$ with almost surely $\alpha$-Hölder continuous sample paths for every $\alpha < \nu$. Now consider $\tilde{f} = \sum_{l=1}^L \tilde{h}_l \circ \phi_l$. Then $\tilde{f}$ is a version of $f$. We proceed to bound $\mathbb{E}_{f\sim\Pi_n}\left[\|\tilde{f}\|_{\mathcal{CH}^\alpha(\mathcal{M})}^2\right]$. For any $1 \leq l, r \leq L$ write

$$\left|\tilde{h}_r(\phi_r(\phi_l^{-1}(x))) - \tilde{h}_r(\phi_r(\phi_l^{-1}(y)))\right| \leq K\left|\phi_r(\phi_l^{-1}(x)) - \phi_r(\phi_l^{-1}(y))\right|^\alpha \leq CK|x-y|^\alpha \tag{154}$$

where $C$ is the Lipshitz constant of $\phi_r \circ \phi_l^{-1}$ which is well defined and finite because this composition is a diffeomorphism and $K$ is a random constant with $\mathbb{E}_{f\sim\Pi_n} K^2 \leq C_{\alpha,\nu,d}$. Hence

$$\mathbb{E}_{f\sim\Pi_n}\|\tilde{f}\|_{\mathcal{CH}^\alpha(\mathcal{M})}^2 = \mathbb{E}_{f\sim\Pi_n} \sum_{l=1}^L \left\|\left(\chi_l \tilde{f}\right) \circ \phi_l^{-1}\right\|_{\mathcal{CH}^\alpha(\mathbb{R}^d)}^2 \tag{155}$$

$$= \mathbb{E}_{f\sim\Pi_n} \sum_{l=1}^L \left\|\left(\chi_l \sum_{r=1}^L \tilde{h}_r \circ \phi_r\right) \circ \phi_l^{-1}\right\|_{\mathcal{CH}^\alpha(\mathbb{R}^d)}^2 \tag{156}$$

$$\lesssim \sum_{l=1}^L \sum_{r=1}^L \mathbb{E}_{f\sim\Pi_n}\left\|\tilde{h}_r \circ \phi_r \circ \phi_l^{-1}\right\|_{\mathcal{CH}^\alpha(\mathbb{R}^d)}^2 \lesssim \mathbb{E}_{f\sim\Pi_n} K^2 \leq C_{\alpha,\nu,d}. \tag{157}$$

which gives the $\nu \leq 1$ case.

We now turn to the general case. The proof will be similar to the one of Ghosal and van der Vaart [20], Proposition I.3 although we need to control the Hölder norms and work through charts since our Gaussian processes are supported on manifolds. Assume for simplicity that $d = 1, 1 < \nu \leq 2$, otherwise it suffices to introduce coordinates and to proceed by induction on $\lfloor\nu\rfloor$. Let $l \in \{1,\ldots,L\}$, and as before define $\widetilde{K}_l(x,y) = \left(\chi_l \circ \phi_l^{-1}\right)(x)\left(\chi_l \circ \phi_l^{-1}\right)(y)K\left(\phi_l^{-1}(x), \phi_l^{-1}(y)\right)$ the covariance kernel of $h_l = (\chi_l f) \circ \phi_l^{-1}$ as well as $\widetilde{\mathbb{H}}_l$ its RKHS.

First, let us construct an $L^2(\Omega)$-derivative $\dot{h}_l$ of $h_l$, where $L^2(\Omega)$ is the space of random variables with finite variance with $\langle a,b\rangle_{L^2(\Omega)} = \mathbb{E}(ab)$. This derivative is a square integrable process on $\mathcal{V}_l$ such that

$$\mathbb{E}_{f\sim\Pi_n}\left|\frac{h_l(x+\delta) - h_l(x)}{h} - \dot{h}_l(x)\right|^2 \to 0 \tag{158}$$

as $h \to 0$, for all $x \in \mathcal{V}_l$. For this, we will first show that $\frac{\partial\widetilde{K}_l}{\partial x}(x,\cdot) \in \widetilde{\mathbb{H}}_l$ for every $x \in \mathcal{V}_l$—here $\frac{\partial\widetilde{K}_l}{\partial x}$ denotes the derivative of the function $\widetilde{K}_l(\cdot,\cdot')$ with respect to the first argument—and that

$$\left\|\frac{\partial\widetilde{K}_l}{\partial x}(x,\cdot) - \frac{\partial\widetilde{K}_l}{\partial x}(x',\cdot)\right\|_{\widetilde{\mathbb{H}}_l} \leq C_\nu|x-x'|^{\nu-1}. \tag{159}$$

We first show that $\frac{\widetilde{K}_l(x+\delta,\cdot)-\widetilde{K}_l(x,\cdot)}{h}$ is a Cauchy net[8] in $\widetilde{\mathbb{H}}_l$. We have

$$\left\| \frac{\widetilde{K}_l(x+\delta,\cdot)-\widetilde{K}_l(x,\cdot)}{h} - \frac{\widetilde{K}_l(x+\delta',\cdot)-\widetilde{K}_l(x,\cdot)}{h'} \right\|_{\widetilde{\mathbb{H}}_l} \tag{160}$$

$$= \sup_{\|\varphi\|_{\widetilde{\mathbb{H}}_l}=1} \left\langle \frac{\widetilde{K}_l(x+\delta,\cdot)-\widetilde{K}_l(x,\cdot)}{h} - \frac{\widetilde{K}_l(x+\delta',\cdot)-\widetilde{K}_l(x,\cdot)}{h'}, \varphi \right\rangle_{\widetilde{\mathbb{H}}_l} \tag{161}$$

$$= \sup_{\|\varphi\|_{\widetilde{\mathbb{H}}_l}=1} \left( \frac{\varphi(x+\delta)-\varphi(x)}{h} - \frac{\varphi(x+\delta')-\varphi(x)}{h'} \right) \tag{162}$$

$$= \sup_{\|\varphi\|_{\widetilde{\mathbb{H}}_l}=1} \int_0^1 [\varphi'(x+th) - \varphi'(x+th')]\,\mathrm{d}t \tag{163}$$

$$\leq \sup_{\|\varphi\|_{\widetilde{\mathbb{H}}_l}=1} \|\varphi'\|_{\mathcal{CH}^{\nu-1}(\mathcal{V}_l)} |h-h'|^{\nu-1} \tag{164}$$

$$\leq \sup_{\|\varphi\|_{\widetilde{\mathbb{H}}_l}=1} \|\varphi\|_{\mathcal{CH}^{\nu}(\mathcal{V}_l)} |h-h'|^{\nu-1} \tag{165}$$

where in (163) the derivative $\varphi'$ exists because, exactly as in the case $\nu \leq 1$, we can show show that $\widetilde{\mathbb{H}}_l \hookrightarrow \mathcal{CH}^\nu(\mathbb{R}^d)$. This also implies that for a constant $C = C_\nu$ we have

$$\left\| \frac{\widetilde{K}_l(x+\delta,\cdot)-\widetilde{K}_l(x,\cdot)}{h} - \frac{\widetilde{K}_l(x+\delta',\cdot)-\widetilde{K}_l(x,\cdot)}{h'} \right\|_{\widetilde{\mathbb{H}}_l} \leq C|h-h'|^{\nu-1}. \tag{166}$$

As $|h-h'|^{\nu-1} \to 0$ when $h,h' \to 0$, because $\nu > 1$, this proves that $\frac{\widetilde{K}_l(x+\delta,\cdot)-\widetilde{K}_l(x,\cdot)}{h}$ is a Cauchy net in $\widetilde{\mathbb{H}}_l$: by completeness of $\widetilde{\mathbb{H}}_l$ it converges in $\widetilde{\mathbb{H}}_l$ to a limit. Since by general properties of RKHSs, convergence in $\widetilde{\mathbb{H}}_l$ implies pointwise convergence, the limit satisfies

$$\lim_{h \to 0} \frac{\widetilde{K}_l(x+\delta,y)-\widetilde{K}_l(x,y)}{h} = \frac{\partial \widetilde{K}_l}{\partial x}(x,y). \tag{167}$$

Hence the partial derivative $\frac{\partial \widetilde{K}_l}{\partial x}(x,y)$ exists for all $y$ and $\frac{\partial \widetilde{K}_l}{\partial x}(x,\cdot) \in \widetilde{\mathbb{H}}_l$. Moreover, by the isometry $L^2(\Omega) \ni h_l(x) \mapsto \mathbb{E}_{f \sim \Pi_n} h_l(x)h_l(\cdot) = \widetilde{K}_l(x,\cdot) \in \widetilde{\mathbb{H}}_l$, we deduce that $h_l$ is actually $L^2(\Omega)$-differentiable, with an $L^2(\Omega)$-derivative denoted as $\dot{h}_l$, and that the derivative process $\dot{h}_l$ is Gaussian, as it is an $L^2(\Omega)$-limit of Gaussian random variables, satisfying $\mathbb{E}_{f \sim \Pi_n} \dot{h}_l(x)\dot{h}_l(y) = \left\langle \frac{\partial \widetilde{K}_l}{\partial x}(x,\cdot), \frac{\partial \widetilde{K}_l}{\partial x}(y,\cdot) \right\rangle_{\widetilde{\mathbb{H}}_l}$.

Having established the existence of an $L^2(\Omega)$-derivative $\dot{h}_l$ of the process $h_l$, we would like now to show that $\dot{h}_l$ possesses a $(\gamma-1)$-regular version for every $\gamma < \nu$. For this, we would like to apply Lemma 28 to $\dot{h}_l$. Notice that, by isometry, for all $h > 0$ we have

$$\mathbb{E}_{f \sim \Pi_n} \left| \dot{h}_l(x) - \dot{h}_l(y) \right|^2 = \left\| \frac{\partial \widetilde{K}_l}{\partial x}(y,\cdot) - \frac{\partial \widetilde{K}_l}{\partial x}(x,\cdot) \right\|_{\widetilde{\mathbb{H}}_l}^2 \tag{168}$$

$$\leq 3 \left\| \frac{\widetilde{K}_l(y+\delta,\cdot)-\widetilde{K}_l(y,\cdot)}{h} - \frac{\partial \widetilde{K}_l}{\partial x}(y,\cdot) \right\|_{\widetilde{\mathbb{H}}_l}^2 \tag{169}$$

$$+ 3 \left\| \frac{\widetilde{K}_l(x+\delta,\cdot)-\widetilde{K}_l(x,\cdot)}{h} - \frac{\partial \widetilde{K}_l}{\partial x}(x,\cdot) \right\|_{\widetilde{\mathbb{H}}_l}^2 \tag{170}$$

$$+ 3 \left\| \frac{\widetilde{K}_l(x+\delta,\cdot)-\widetilde{K}_l(x,\cdot)}{h} - \frac{\widetilde{K}_l(y+\delta,\cdot)-\widetilde{K}_l(y,\cdot)}{h} \right\|_{\widetilde{\mathbb{H}}_l}^2. \tag{171}$$

---

[8]See for instance Aliprantis and Border [1] for a review of Cauchy nets.

Therefore, by the same arguments as above, we have

$$\left(\mathbb{E}_{f\sim\Pi_n}\left|\dot{h}_l(x)-\dot{h}_l(y)\right|^2\right)^{1/2} \lesssim \liminf_{h\to 0}\left\|\frac{\widetilde{K}_l(x+\delta,\cdot)-\widetilde{K}_l(x,\cdot)}{h}-\frac{\widetilde{K}_l(y+\delta,\cdot)-\widetilde{K}_l(y,\cdot)}{h}\right\|_{\widetilde{\mathbb{H}}_l} \quad (172)$$

$$\leq \liminf_{h\to 0}\sup_{\|\varphi\|_{\widetilde{\mathbb{H}}_l}=1}\int_0^1|\varphi'(x+th)-\varphi'(y+th)|\,\mathrm{d}t \quad (173)$$

$$\leq \liminf_{h\to 0}C_\nu|x-y|^{\nu-1}=C_\nu|x-y|^{\nu-1} \quad (174)$$

where the transition from the second-to-last line to the last line is similar to (163)–(165).

Therefore, we can apply Lemma 28 to $\dot{h}_l$, and find a version $\tilde{h}'_l$ of $\dot{h}_l$ with sample paths in $\mathcal{CH}^{\alpha-1}(\mathcal{V}_l)$ almost surely for all $\alpha < \nu$, such that

$$\mathbb{E}_{f\sim\Pi_n}\big\|\tilde{h}'_l\big\|^2_{\mathcal{CH}^{\alpha-1}(\mathcal{V}_l)} \leq C_{\nu,\alpha} < +\infty \qquad \alpha < \nu. \quad (175)$$

This gives Hölder regularity of the respective derivatives: we now integrate these to obtain a Hölder-regular version of the process itself. Take any $c_l \in (0,1)$ and consider $\tilde{h}_l = h_l(c_l) + \int_{c_l}^{\cdot} \tilde{h}'_l(t)\,\mathrm{d}t$. Then since $\tilde{h}'_l$ is almost surely in $\mathcal{CH}^{\alpha-1}(\mathcal{V}_l)$, $\tilde{h}_l$ is has sample paths almost surely in $\mathcal{CH}^{\alpha}(\mathcal{V}_l)$. Moreover, it is easy to check using our previous results that $\tilde{h}_l$ has an $L^2(\Omega)$-derivative given by $\tilde{h}'_l$. This implies that $\tilde{h}_l$ is a version of $h_l$.

To conclude the argument, we construct $\tilde{f}$ from the obtained parts, by pulling $\tilde{h}_l$ back from the charts to the manifold. Consider $\tilde{f} = \sum_{l=1}^L \tilde{h}_l \circ \phi_l$. Arguing as in the case $\nu \leq 1$, we see that $\tilde{f}$ is a version of $f$ with $\mathcal{CH}^{\alpha}(\mathcal{M})$ sample paths for every $\alpha < \nu$, and for every $\alpha < \nu$ we have $\mathbb{E}_{f\sim\Pi_n}\|\tilde{f}\|^2_{\mathcal{CH}^{\alpha}(\mathcal{M})} \leq C_{\alpha,\nu} < +\infty$. This gives the claim. $\qquad\square$

With this, it is easy to prove that all Matérn Gaussian processes considered in this paper can be seen as Gaussian random elements in the Banach space $\big(\mathcal{C}(\mathcal{M}),\|\cdot\|_{\mathcal{C}(\mathcal{M})}\big)$ of continuous functions on $\mathcal{M}$. This allows us to use the same proof scheme as in van der Vaart and van Zanten [56] through the control of the *concentration functions* defined in Appendix B. It is also important that we work with Gaussian random elements in $\mathcal{C}(\mathcal{M})$—and not only with the classical notion of Gaussian process, as the concentration functions are defined using the *Gaussian random element RKHS* defined in van Zanten and van der Vaart [59], which can potentially be different from the classical RKHS. Fortunately, when the process is a Gaussian random element in $\mathcal{C}(\mathcal{M})$, van Zanten and van der Vaart [59], Theorem 2.1 implies that the two notions of RKHS coincide.

**Corollary 30.** *The intrinsic Matérn Gaussian processes of Definition 4, their truncated versions as in Theorem 6 as well as the extrinsic Matérn Gaussian processes of Definition 7 are Gaussian random elements in $\big(\mathcal{C}(\mathcal{M}),\|\cdot\|_{\mathcal{C}(\mathcal{M})}\big)$.*

*Proof.* By Lemma 18, it suffices to show that the processes have almost surely continuous sample paths. Since Euclidean Matérn Gaussian processes have continuous sample paths, this implies the same for their restrictions, the extrinsic Matérn Gaussian processes on $\mathcal{M}$. For the intrinsic Matérn process, we apply Lemma 29. $\qquad\square$

Using Lemma 29 and known properties of Euclidean Matérn processes, we now show, in a sense, that all of the Matérn processes presented in this paper are sub-Gaussian, in a manner which holds uniformly with respect to the truncation parameter in the case of the truncated intrinsic Matérn process, and live in Hölder spaces with appropriate exponents. This result is used to control Hölder norms when going from the error at input locations to the $L^2(p_0)$-error. We use the notation $\Pi_n$ to emphasize that the prior depends on $n$ when we consider a truncated intrinsic Matérn process.

**Lemma 31.** *For $f \sim \Pi_n$ with $\Pi_n$ the prior in either Definition 4, Theorem 6 or Definition 7, for every $\nu > 0$ and $\gamma < \nu, \gamma \notin \mathbb{Z}_{>0}$, there exists a constant $\sigma(f) = \sigma_\gamma(f)$ independent of $n$ we have for $x > 0$ that*

$$\mathbb{P}\Big(\|f\|_{\mathcal{CH}^\gamma(\mathcal{M})} > (x+1)\sigma(f)\Big) \leq 2e^{-x^2/2}. \quad (176)$$

*Proof.* We start with the restriction $f$ of an extrinsic Matérn process $\tilde{f}$ to $\mathcal{M}$, as in Definition 7. By van der Vaart and van Zanten [56], Section 3.1, for every $\gamma < \nu$ we have $\tilde{f} \in \mathcal{CH}^\gamma\big([0,1]^D\big)$ almost surely. By Ghosal and van der Vaart [20], Lemma I.7, for every $\gamma < \nu$, $\tilde{f}$ is a Gaussian random element in the Banach space $\mathcal{CH}^\gamma\big([0,1]^D\big)$. In particular, by the Borell–TIS inequality [20, Proposition I.8] we have for $x > 0$ that

$$\mathbb{P}\Big(\|\tilde{f}\|_{\mathcal{CH}^\gamma([0,1]^D)} > (x+1)\sigma\big(\tilde{f}\big)\Big) \leq 2e^{-x^2/2} \tag{177}$$

where $\sigma\big(\tilde{f}\big) = \Big(\mathbb{E}\,\|\tilde{f}\|^2_{\mathcal{CH}^\gamma([0,1]^D)}\Big)^{1/2} < \infty$. Since $\mathcal{M}$ is smooth, the restriction $f$ also satisfies for $x > 0$ the expression

$$\mathbb{P}\Big(\|f\|_{\mathcal{CH}^\gamma(\mathcal{M})} > (x+1)\sigma(f)\Big) \leq 2e^{-x^2/2} \tag{178}$$

perhaps for a possibly larger constant $\sigma(f)$. Finally, the case of the intrinsic Matérn process $f \sim \Pi_n$ truncated at $J_n \in \mathbb{Z}_{>0} \cup \{\infty\}$ follows in the same manner, as we have shown in Lemma 29 that $\sup_{n \geq 1} \mathbb{E}_{f \sim \Pi_n} \|f\|^2_{\mathcal{CH}^\alpha(\mathcal{M})} \leq C_{\alpha,\nu}$. $\qquad\square$

# E  Small Ball Asymptotics

Here, we bound the probability that a Matérn Gaussian process lies in the $\varepsilon$-ball with respect to the $L^\infty$-norm for small $\varepsilon$. For a Banach space $\mathbb{B}$, an element $x \in \mathbb{B}$ and a number $r \in \mathbb{R}_{>0}$, let us denote the closed $r$-ball around $x$ by $B_r^{\mathbb{B}}(x)$. We start by an upper bound on the metric entropy of Sobolev balls on $\mathcal{M}$ with respect to the uniform norm.

**Lemma 32** (Entropy of Sobolev balls)**.** *For any $s > 0$ define the $\varepsilon$-covering number of $A \subseteq H^s(\mathcal{M})$ with respect to the norm $\|\cdot\|_{L^\infty(\mathcal{M})}$ by*

$$N\Big(\varepsilon, A, \|\cdot\|_{L^\infty(\mathcal{M})}\Big) = \operatorname*{arg\,min}_{J \in \mathbb{Z}_{>0}} \bigg\{ \exists h_1, .., h_J \in A : A \subset \bigcup_{j=1}^J B_\varepsilon^{L^\infty(\mathcal{M})}(h_j) \bigg\}. \tag{179}$$

*Then for any $s > d/2$, there exist $C, \varepsilon_0 > 0$ such that for every $\varepsilon \leq \varepsilon_0$*

$$\ln N\Big(\varepsilon, B_1^{H^s(\mathcal{M})}(0), \|\cdot\|_{L^\infty(\mathcal{M})}\Big) \leq C\varepsilon^{-\frac{d}{s}}, \tag{180}$$

*where the left-hand side of the inequality above, as a function of $\varepsilon$, is called the* METRIC ENTROPY *of the Sobolev ball $B_1^{H^s(\mathcal{M})}(0)$ with respect to the uniform norm $\|\cdot\|_{L^\infty(\mathcal{M})}$.*

*Proof.* Using charts we will reduce the problem to the entropy of the unit ball of the Sobolev space $H^s\big([0,1]^d\big)$ for which the upper bound is known. Take $f \in B_1^{H^s(\mathcal{M})}(0)$ and consider approximations of $f$ by $g$ of the form

$$g = \sum_{l=1}^L \chi_l(g_l \circ \phi_l) \tag{181}$$

for some functions $g_l : \mathcal{V}_l \to \mathbb{R}$ where $\mathcal{V}_l \subseteq \mathbb{R}^d$. We have

$$\|f - g\|_{L^\infty(\mathcal{M})} = \Big\|\sum_{l=1}^L \chi_l(g_l \circ \phi_l - f)\Big\|_{L^\infty(\mathcal{M})} \leq \sum_{l=1}^L \big\|\chi_l(g_l \circ \phi_l - f)\big\|_{L^\infty(\mathcal{U}_l)} \tag{182}$$

$$\leq \sum_{l=1}^L \big\|g_l \circ \phi_l - f\big\|_{L^\infty(\mathcal{U}_l)} \leq \sum_{l=1}^L \big\|g_l - f \circ \phi_l^{-1}\big\|_{L^\infty(\mathcal{V}_l)} \tag{183}$$

$$\leq L \max_{1 \leq l \leq L} \big\|g_l - f \circ \phi_l^{-1}\big\|_{L^\infty([0,1]^d)}. \tag{184}$$

This means that to approximate $f$ by $g$ uniformly on $\mathcal{M}$, we need to choose the functions $g_l$ that approximate $f \circ \phi_l^{-1}$ well with respect to the uniform norm on $[0,1]^d$.

Next, we show that the functions $f \circ \phi_l^{-1}$ are contained in an Euclidean Sobolev ball of radius $R$, with $R$ depending only on $\nu$ and the atlas. To do this we use Große and Schneider [24], Lemma 2.1 to write

$$\left\| f \circ \phi_l^{-1} \right\|_{H^s([0,1]^d)} = \left\| \sum_{l'=1}^{L} (\chi_{l'} f) \circ \phi_l^{-1} \right\|_{H^s([0,1]^d)} \leq \sum_{l'=1}^{L} \left\| (\chi_{l'} f) \circ \phi_l^{-1} \right\|_{H^s([0,1]^d)} \tag{185}$$

$$= \sum_{l'=1}^{L} \left\| (\chi_{l'} f) \circ \phi_{l'}^{-1} \circ \phi_{l'} \circ \phi_l^{-1} \right\|_{H^s([0,1]^d)} \tag{186}$$

$$\lesssim \sum_{l'=1}^{L} \left\| (\chi_{l'} f) \circ \phi_{l'}^{-1} \right\|_{H^s([0,1]^d)} \lesssim \|f\|_{H^s(\mathcal{M})}. \tag{187}$$

Note, importantly, that the remark just above Große and Schneider [24], Lemma 2.1 allows us to consider Besov spaces $B_{2,2}^s$ coinciding with the Sobolev spaces $H^s$ instead of the Besov spaces $B_{2,\infty}^s$—to get from the second line to the third. Note also that the constant hidden behind the notation $\lesssim$ in the last line where we use Theorem 16 is the radius $R$. Without loss of generality we assume $R = 1$. By the Euclidean counterpart of the result we are proving [21, Theorem 4.3.36], we have

$$\ln N\left(\varepsilon, B_1^{H^s[0,1]^d}(0), \|\cdot\|_{L^\infty([0,1]^d)}\right) \lesssim \varepsilon^{-\frac{d}{s}}. \tag{188}$$

Let $h_1, .., h_J \in B_1^{H^s([0,1]^d)}(0)$ be such that $B_1^{H^s([0,1]^d)}(0) \subset \bigcup_{j=1}^{J} B_{\varepsilon/L}^{L^\infty([0,1]^d)}(h_k)$. Then for any $f \in B_1^{H^s(\mathcal{M})}(0)$ there exists a sequence $\{j_l\}_{l=1}^{L} \subseteq \{1, .., J\}$ such that

$$\left\| f - \sum_{l=1}^{L} \chi_l (h_{j_l} \circ \phi_l) \right\|_{L^\infty(\mathcal{M})} < L\frac{\varepsilon}{L} = \varepsilon. \tag{189}$$

This shows that $N\left(\varepsilon, B_1^{H^s(\mathcal{M})}(0), \|\cdot\|_{L^\infty(\mathcal{M})}\right) \leq LJ$, where $L$ is just the number of charts, thereby proving the claim. $\qquad \square$

For the related *diffusion spaces* [16], the RKHS corresponding to the heat (diffusion) kernels, Castillo et al. [12] uses the results of Coulhon et al. [14] to bound the entropy in terms of a wavelet frame instead of relying on charts. We believe this alternative proof scheme should work in our case as well. However, we could not, to the best of our effort, get a tight-enough bound for the Sobolev spaces by directly using the results of Coulhon et al. [14] and therefore we chose to rely on charts instead.

Having established regularity properties for our prior processes, we now turn to the *small ball problem*: we want to find sharp lower bounds on $\mathbb{P}(\|f\|_{L^\infty(\mathcal{M})} < \varepsilon)$ where $f \sim \Pi$ is our prior process. This will be crucial in order to control the *concentration functions* used in Appendix B. In fact, it is well-known that this problem is closely related to the estimation of the metric entropy of the unit ball of the RKHS of $f$ with respect to the uniform norm: see Li and Linde [31] for details. Since we have already characterized the RKHS of our processes in Proposition 27 and Lemma 24, we are able to lower bound the small-ball probabilities. The technicality here involves getting a bound uniform in the truncation parameter for the truncated intrinsic Matérn process, as the truncated Matérn process is a sequence of priors rather than a fixed prior.

**Lemma 33.** *If $f \sim \Pi_n$ where $\Pi_n$ is the prior in either Definition 4 and Theorem 6 or Definition 7 with smoothness parameter $\nu > d/2$, then there exist two constants $C, \varepsilon_0 > 0$ that do not depend on $n$ such that for all $\varepsilon \leq \varepsilon_0$ we have $-\ln \mathbb{P}\left(\|f\|_{L^\infty(\mathcal{M})} < \varepsilon\right) \leq C\varepsilon^{-\frac{d}{\nu}}$.*

*Proof.* Because the processes are Gaussian random elements in $\mathcal{C}(\mathcal{M})$ by Lemma 18, their stochastic process RKHS given by Proposition 27 coincide with their *Gaussian random element RKHS* defined in van Zanten and van der Vaart [59]. Hence, for the non-truncated intrinsic and the extrinsic Matérn processes, the result follows by a direct application of Lemma 32 and Li and Linde [31], Theorem 1.2. For the intrinsic Matérn process truncated at $J_n$ it is not immediately clear that the constants $C, \varepsilon_0$ can be taken independent of $n$, so we go through the proof of Li and Linde [31], Proposition 3.1 to see that this is, in fact, the case. We first need a crude upper bound of the form

$$-\ln \mathbb{P}\left(\|f\|_{L^\infty(\mathcal{M})} < \varepsilon\right) \leq c\varepsilon^{-c} \tag{190}$$

for some possibly large constant $c > 0$. To get such a bound, we use Castillo et al. [12], Proposition 3 which shows the existence of a universal constant $C > 0$ such that for all $\varepsilon \leq \min(1, 4\sigma(f))$

$$-\ln\mathbb{P}\Big(\|f\|_{L^\infty(\mathcal{M})} < \varepsilon\Big) \leq Cn(\varepsilon)\ln\left(\frac{6n(\varepsilon)\max(1,\sigma(f))}{\varepsilon}\right) \tag{191}$$

where $\sigma(f) = \Big(\mathbb{E}_{f\sim\Pi_n}\|f\|^2_{L^\infty(\mathcal{M})}\Big)^{1/2}$ and $n(\varepsilon)$ is defined in the following way in Castillo et al. [12], page 684 using auxiliary quantities $l_J$ that are defined in Li and Linde [31], page 1562, namely

$$n(\varepsilon) = \max\{J \geq 0 : 4l_J(f) \geq \varepsilon\}, \tag{192}$$

$$l_J(f) = \inf\Big\{\Big(\mathbb{E}_{f\sim\Pi_n}\Big\|\sum_{j\geq J}\varepsilon_j h_j\Big\|^2_{L^\infty(\mathcal{M})}\Big)^{1/2} : f \overset{(d)}{=} \sum_{j\geq 1}\varepsilon_j h_j\Big\} \tag{193}$$

with $\overset{(d)}{=}$ standing for the equality in distributions and the infimum being taken over all possible decompositions $\sum_{j\geq 1}\varepsilon_j h_j$ with $h_j \in \mathcal{C}(\mathcal{M})$, $\varepsilon_j$ being a sequence of IID $N(0,1)$ random variables, and the series being required to converge uniformly almost surely.[9]

The function $f = \sum_{j=1}^{J_n+1}\big(\frac{2\nu}{\kappa^2} + \lambda_{j-1}\big)^{-\frac{\nu+d/2}{2}}\varepsilon_j f_{j-1}$ is a valid decomposition. Therefore

$$l_J(f) \leq \left(\mathbb{E}_{f\sim\Pi_n}\Big\|\sum_{j=J}^{J_n+1}\Big(\frac{2\nu}{\kappa^2} + \lambda_{j-1}\Big)^{-\frac{\nu+d/2}{2}}\varepsilon_j f_{j-1}\Big\|^2_{L^\infty(\mathcal{M})}\right)^{1/2}. \tag{194}$$

By the Sobolev Embedding Theorem and by Weyl's Law, given in Result 10, for every $d/2 < \gamma < \nu$ there exists a constant $C = C_{\gamma,\mathcal{M}}$ such that for all $J \in \mathbb{Z}_{>0}$, allowing $C$ to change from line to line, we have

$$\mathbb{E}_{f\sim\Pi_n}\Big\|\sum_{j=J}^{J_n+1}(1+\lambda_{j-1})^{-\frac{\nu+d/2}{2}}\varepsilon_j f_{j-1}\Big\|^2_{L^\infty(\mathcal{M})} \leq C^2\,\mathbb{E}_{f\sim\Pi_n}\Big\|\sum_{j=J}^{J_n+1}(1+\lambda_{j-1})^{-\frac{\nu+d/2}{2}}\varepsilon_j f_{j-1}\Big\|^2_{H^\gamma(\mathcal{M})} \tag{195}$$

$$= C^2\sum_{j=J}^{J_n+1}(1+\lambda_{j-1})^{-(\nu+d/2-\gamma)} \tag{196}$$

$$\leq C^2\sum_{j=J}^{J_n+1}j^{-(1+2(\nu-\gamma)/d)} \tag{197}$$

$$\leq C^2\sum_{j\geq J}j^{-(1+2(\nu-\gamma)/d)} \tag{198}$$

$$\leq C^2 J^{-2(\nu-\gamma)/d}. \tag{199}$$

By choosing $J = 1$ this gives us $\sigma(f) \leq C$ independent of $n$. Moreover, by choosing $J \geq C\varepsilon^{-\frac{d}{2(\nu-\gamma)}}$, again for a comparison constant $C$ independent of $n$, this gives us $n(\varepsilon) \leq C\varepsilon^{-\frac{d}{2(\nu-\gamma)}}$ for $C$ independent of $n$. This implies using Castillo et al. [12], Proposition 3 that

$$-\ln\mathbb{P}\Big(\|f\|_{L^\infty(\mathcal{M})} < \varepsilon\Big) \leq c\varepsilon^{-c} \tag{200}$$

for $c > 0$ independent of $n$.

With this crude bound, we can now continue the proof of Li and Linde [31], Proposition 3.1. For this, we need a metric entropy estimate. For this notice that for all $J \in \mathbb{Z}_{>0} \cup \{\infty\}$ we have $B_1^{\mathbb{H}^J}(0) \subset B_1^{\mathbb{H}^\infty}(0) = B_1^{H^{\nu+d/2}(\mathcal{M})}(0)$, and therefore using Lemma 32, we have the metric entropy estimate

$$\ln N\Big(\varepsilon, B_1^{\mathbb{H}^J}(0), \|\cdot\|_{L^\infty(\mathcal{M})}\Big) \leq C\varepsilon^{-\frac{d}{\nu+d/2}} \tag{201}$$

for a constant $C > 0$ independent of $J$. Therefore following the proof of proposition 3.1 in Li and Linde [31] (with $J \equiv 1$) we find $-\ln\mathbb{P}\Big(\|f\|_{L^\infty(\mathcal{M})} < \varepsilon\Big) \leq C\varepsilon^{-\frac{d}{\nu}}$ for every $\varepsilon \leq \varepsilon_0$, where $C, \varepsilon_0 > 0$ are constants independent of $n$. $\qquad\square$

---

[9]We consider $\sum_{j\geq 1}$, unlike $\sum_{j\geq 0}$ frequently used above, in order to follow the respective references.

## F   Expressions for Pointwise Worst-case Errors

Let $k$ be a kernel on some abstract input domain $\mathcal{X}$, and let $\mathbb{H}_k$ be the respective RKHS. Consider $n$ input values $\mathbf{X} \subseteq \mathcal{X}$ and let $\sigma_\varepsilon^2 > 0$ be the noise variance. Define

$$m_{k,\mathbf{X},f,\varepsilon}(t) = \mathbf{K}_{t\mathbf{X}}(\mathbf{K}_{\mathbf{XX}} + \sigma_\varepsilon^2\mathbf{I})^{-1}(f(\mathbf{X}) + \boldsymbol{\varepsilon}), \tag{202}$$

$$v^{(\mathrm{i})}(t) = v_{k,\mathbf{X}}(t) = k(t,t) - \mathbf{K}_{t\mathbf{X}}\left(\mathbf{K}_{\mathbf{XX}} + \sigma_\varepsilon^2\mathbf{I}\right)^{-1}\mathbf{K}_{\mathbf{X}t}. \tag{203}$$

**Proposition 34.** *With notation above*

$$v^{(\mathrm{i})}(t) = \sup_{f\in\mathcal{H}_k,\|f\|_{\mathcal{H}_k}\leq 1} \mathbb{E}_{\boldsymbol{\varepsilon}\sim\mathrm{N}(\mathbf{0},\sigma_\varepsilon^2\mathbf{I})}|f(t) - m_{k,\mathbf{X},f,\varepsilon}(t)|^2. \tag{204}$$

*Proof.* To simplify notation, we shorten $\mathbb{E}_{\boldsymbol{\varepsilon}\sim\mathrm{N}(\mathbf{0},\sigma_\varepsilon^2\mathbf{I})}$ to $\mathbb{E}$ and denote $\boldsymbol{\alpha} = \mathbf{K}_{t\mathbf{X}}(\mathbf{K}_{\mathbf{XX}} + \sigma_\varepsilon^2\mathbf{I})^{-1}$. First of all, by direct computation,

$$\mathbb{E}\, m_{k,\mathbf{X},f,\varepsilon}(t) = \boldsymbol{\alpha}f(\mathbf{X}), \tag{205}$$

$$\mathbb{E}\, m_{k,\mathbf{X},f,\varepsilon}(t)^2 = \boldsymbol{\alpha}f(\mathbf{X})f(\mathbf{X})^\top\boldsymbol{\alpha}^\top + \sigma_\varepsilon^2\boldsymbol{\alpha}\boldsymbol{\alpha}^\top. \tag{206}$$

Write

$$\mathbb{E}|f(t) - m_{k,\mathbf{X},f,\varepsilon}(t)|^2 = f(t)^2 - 2f(t)\,\mathbb{E}\, m_{k,\mathbf{X},f,\varepsilon}(t) + \mathbb{E}\, m_{k,\mathbf{X},f,\varepsilon}(t)^2 \tag{207}$$

$$= f(t)^2 - 2f(t)\boldsymbol{\alpha}f(\mathbf{X}) + \boldsymbol{\alpha}f(\mathbf{X})f(\mathbf{X})^\top\boldsymbol{\alpha}^\top + \sigma_\varepsilon^2\boldsymbol{\alpha}\boldsymbol{\alpha}^\top \tag{208}$$

$$= (f(t) - \boldsymbol{\alpha}f(\mathbf{X}))^2 + \sigma_\varepsilon^2\boldsymbol{\alpha}\boldsymbol{\alpha}^\top \tag{209}$$

$$= \left\langle k(t,\cdot) - \sum_{j=1}^n \alpha_j k(x_j,\cdot), f\right\rangle_{\mathbb{H}_k}^2 + \sigma_\varepsilon^2\boldsymbol{\alpha}\boldsymbol{\alpha}^\top. \tag{210}$$

As $\|g\|_{\mathbb{H}_k} = \sup_{f\in\mathbb{H}_k,\|f\|_{\mathbb{H}_k}\leq 1}\langle g,f\rangle_{\mathbb{H}_k}$, implying $\sup_{f\in\mathbb{H}_k,\|f\|_{\mathbb{H}_k}\leq 1}\langle g,f\rangle_{\mathbb{H}_k}^2 = \|g\|_{\mathbb{H}_k}^2$, we have

$$\sup_{\substack{f\in\mathcal{H}_k\\\|f\|_{\mathcal{H}_k}\leq 1}} \mathbb{E}|f(t) - m_{k,\mathbf{X},f,\varepsilon}(t)|^2 = \left\|k(t,\cdot) - \sum_{j=1}^n \alpha_j k(x_j,\cdot)\right\|_{\mathbb{H}_k}^2 + \sigma_\varepsilon^2\boldsymbol{\alpha}\boldsymbol{\alpha}^\top \tag{211}$$

$$= k(t,t) - 2\boldsymbol{\alpha}\mathbf{K}_{\mathbf{X}t} + \underbrace{\boldsymbol{\alpha}\mathbf{K}_{\mathbf{XX}}\boldsymbol{\alpha}^\top + \sigma_\varepsilon^2\boldsymbol{\alpha}\boldsymbol{\alpha}^\top}_{\boldsymbol{\alpha}\mathbf{K}_{\mathbf{X}t}} \tag{212}$$

$$= k(t,t) - \boldsymbol{\alpha}\mathbf{K}_{\mathbf{X}t} = \underbrace{k(t,t) - \mathbf{K}_{t\mathbf{X}}(\mathbf{K}_{\mathbf{XX}} + \sigma_\varepsilon^2\mathbf{I})^{-1}\mathbf{K}_{\mathbf{X}t}}_{v_{k,\mathbf{X}}(t)}. $$

$$\tag{213}$$

$\square$

We now move to the misspecified case. Consider the RKHS $\mathcal{H}_c$ for some other kernel $c : \mathcal{X} \times \mathcal{X} \to \mathbb{R}$ instead of $\mathbb{H}_k$. Then, continuing from (210), write

$$\sup_{\substack{f\in\mathcal{H}_c\\\|f\|_{\mathcal{H}_c}\leq 1}} \mathbb{E}|f(t) - m_{k,\mathbf{X},f,\varepsilon}(t)|^2 = \left\|c(t,\cdot) - \sum_{j=1}^n \alpha_j c(x_j,\cdot)\right\|_{\mathbb{H}_c}^2 + \sigma_\varepsilon^2\boldsymbol{\alpha}\boldsymbol{\alpha}^\top. \tag{214}$$

The next question is how to compute the norm on the right-hand side. There is not much hope of doing this exactly in the misspecified case: thus, we consider approximations. To this end, we take some large set of locations $\mathbf{X}' \subseteq \mathcal{X}$. Then we use $\|g\|_{\mathbb{H}_c}^2 \approx g(\mathbf{X}')^\top\mathbf{C}_{\mathbf{X}'\mathbf{X}'}^{-1}g(\mathbf{X}')$ for $g(\cdot) = c(t,\cdot) - \sum_{j=1}^n \alpha_j c(x_j,\cdot)$. As a result, we obtain the approximation

$$\sup_{\substack{f\in\mathcal{H}_c\\\|f\|_{\mathcal{H}_c}\leq 1}} \mathbb{E}|f(t) - m_{k,\mathbf{X},f,\varepsilon}(t)|^2 \approx g(\mathbf{X}')^\top\mathbf{C}_{\mathbf{X}'\mathbf{X}'}^{-1}g(\mathbf{X}') + \sigma_\varepsilon^2\boldsymbol{\alpha}\boldsymbol{\alpha}^\top = \tilde{v}_{k,c,\mathbf{X}}(t) = v^{(\mathrm{e})}(t) \tag{215}$$

where $v^{(\mathrm{e})}(t)$ was first introduced in Section 4

To compute spatial averages of this quantity, let $g_t(\cdot) = c(t, \cdot) - \sum_{j=1}^{n} \alpha_j c(x_j, \cdot)$, the same as $g$ before, but now with explicit dependence on $t$. Similarly, put $\boldsymbol{\alpha}_t = \mathbf{K}_{t\mathbf{X}}(\mathbf{K}_{\mathbf{XX}} + \sigma_\varepsilon^2 \mathbf{I})^{-1}$. Then

$$g_t(\mathbf{X}') = \mathbf{C}_{\mathbf{X}' t} - \mathbf{C}_{\mathbf{X}' \mathbf{X}}\boldsymbol{\alpha}_t^\top = \mathbf{C}_{\mathbf{X}' t} - \mathbf{C}_{\mathbf{X}' \mathbf{X}}(\mathbf{K}_{\mathbf{XX}} + \sigma_\varepsilon^2 \mathbf{I})^{-1}\mathbf{K}_{\mathbf{X}t} \qquad (216)$$

$$g_t(\mathbf{X}')^\top \mathbf{C}_{\mathbf{X}'\mathbf{X}'}^{-1} g_t(\mathbf{X}') = (\mathbf{C}_{t\mathbf{X}'} - \boldsymbol{\alpha}_t \mathbf{C}_{\mathbf{X}\mathbf{X}'})\mathbf{C}_{\mathbf{X}'\mathbf{X}'}^{-1}(\mathbf{C}_{\mathbf{X}' t} - \mathbf{C}_{\mathbf{X}' \mathbf{X}}\boldsymbol{\alpha}_t^\top). \qquad (217)$$

From here we can also deduce that

$$\frac{1}{|\mathbf{X}'|} \sum_{t \in \mathbf{X}'} \tilde{v}_{k,c,\mathbf{X}}(t) = \frac{1}{|\mathbf{X}'|} \sum_{t \in \mathbf{X}'} g_t(\mathbf{X}')^\top \mathbf{C}_{\mathbf{X}'\mathbf{X}'}^{-1} g_t(\mathbf{X}') \qquad (218)$$

$$= \frac{1}{|\mathbf{X}'|} \operatorname{tr}\big(g_{\mathbf{X}'}(\mathbf{X}')^\top \mathbf{C}_{\mathbf{X}'\mathbf{X}'}^{-1} g_{\mathbf{X}'}(\mathbf{X}')\big) \qquad (219)$$

where $g_{\mathbf{X}'}(\mathbf{X}') = \mathbf{C}_{\mathbf{X}' \mathbf{X}'} - \mathbf{C}_{\mathbf{X}' \mathbf{X}}(\mathbf{K}_{\mathbf{XX}} + \sigma_\varepsilon^2 \mathbf{I})^{-1}\mathbf{K}_{\mathbf{XX}'}$.

## G  Full Experimental Details

All of our kernels were computed using GPJAX [38] and the GEOMETRIC KERNELS library.[10] We use three manifolds, each represented by a mesh: (i) a dumbbell-shaped manifold represented as a mesh with 1556 nodes, (ii) a sphere represented by an icosahedral mesh with 2562 nodes, and (iii) the Stanford dragon mesh, preprocessed to keep only its largest connected component, which has 100179 nodes. For the sphere, we also considered a finer icosahedral mesh with 10242 nodes, but this was found to have virtually no effect on the computed pointwise expected errors.

We use extrinsic Matérn and Riemannian Matérn kernels with the following hyperparameters: $\sigma_f^2 = 1$ and $\sigma_\varepsilon^2 = 0.0005$. For the truncated Karhunen–Loève expansion, we used $J = 500$ eigenpairs obtained from the mesh. We selected smoothness values to ensure norm-equivalence of the intrinsic and extrinsic kernels' reproducing kernel Hilbert spaces, which was $\nu = 5/2$ for the intrinsic model, and $\nu = 5/2 + d/2$ for the extrinsic model, where $d$ is the manifold's dimension. We used different length scales for each manifold: $\kappa = 200$ for the dumbbell, $\kappa = 0.25$ for the sphere, and $\kappa = 0.05$ for the dragon, selected to ensure that the Gaussian processes were neither approximately constant, nor white-noise-like. We considered data sizes of $N = 50$, $N = 500$, and $N = 1000$, respectively, for the dumbbell, sphere, and dragon, sampled uniformly from the mesh's nodes, which in each case resulted in a reasonably-uniform distribution of points across the manifold. Finally, for the extrinsic pointwise error approximation, we used a subset $\mathbf{X}'$ uniformly sampled from each mesh's nodes, of size equal to the data size. For each respective test set, we used the full mesh. Each experiment was repeated for 10 different seeds.

To set the length scales for the extrinsic process, we used maximum marginal likelihood optimization on the full data, except for the dumbbell whose full data size is small and for which we instead generated a larger set consisting of 500 points. We optimzied only the length scale, leaving all other hyperparameters fixed. We used ADAM with a learning rate of 0.005, and an initialization equal to the length scale $\kappa$ of the intrinsic model, except for the dumbbell where this lead to divergence and we instead used an initial value of $\kappa/4$. We ran the optimizer for a total of 1000 steps. With these settings, we found empirically that maximum marginal likelihood optimization always converged.

---

[10]See HTTPS://GPJAX.READTHEDOCS.IO and HTTPS://GEOMETRIC-KERNELS.GITHUB.IO.

