## A   Preliminaries

Here we describe preliminaries necessary for Appendices B and C. This includes some basic properties of the Laplace–Beltrami operator on compact manifolds, partitions of unity subordinate to atlases, function spaces such as Hölder, Sobolev and Besov, and general Gaussian random elements on Banach spaces.

### A.1   Laplace–Beltrami Operator and Subordinate Partitions of Unity

Recall that $\mathcal{M}$ denotes a compact Riemannian manifold. The Laplace-Beltrami operator $\Delta$ on $\mathcal{M}$ is self-adjoint and positive semi-definite [40, Theorem 2.4]. Let $\left(L^2(\mathcal{M}), \langle \cdot, \cdot \rangle\right)$ denote the Hilbert space of square integrable functions on $\mathcal{M}$ with respect to the standard Riemannian volume measure.[2]

By standard theory [10, 20], there exists an orthonormal basis $\{f_j\}_{j=0}^{\infty}$ of $L^2(\mathcal{M})$ consisting of the eigenfunctions of $\Delta$, such that $\Delta f_j = \lambda_j f_j$ with $\lambda_j \geq 0$. We assume that the pairs $(\lambda_j, f_j)$ are sorted such that $0 = \lambda_0 \leq \lambda_j \leq \lambda_{j+1}$. The growth of $\lambda_j$ can be characterized as follows.

**Result 10** (Weyl's Law). *There exists a constant $C > 0$ such that for all $j$ large enough*

$$C^{-1} j^{2/d} \leq \lambda_j \leq C j^{2/d} \tag{16}$$

*Proof.* See Chavel [10], Chapter 1. $\qquad\square$

Following De Vito et al. [13] and Große and Schneider [21] we fix a family $\mathcal{T} = (\mathcal{U}_l, \phi_l, \chi_l)_{l=1}^{L}$ of $\mathcal{M}$, where $L \in \mathbb{N}$, the local coordinates $\phi_l : \mathcal{U}_l \subset \mathcal{M} \to \mathcal{V}_l = \phi_l(\mathcal{U}_l) \subset \mathbb{R}^d$ are smooth diffeomorphisms, and the functions $\chi_l$ form a partition of the unity subordinate to $\{\mathcal{U}_l\}_{l=1}^{L}$, i.e. $\chi_l \in \mathcal{C}^\infty(\mathcal{M}), \operatorname{supp}(\chi_l) \subset \mathcal{U}_l, 0 \leq \chi_l \leq 1$ and $\sum_l \chi_l = 1$.[3] For convenience and without loss of generality we assume that $\mathcal{V}_l \subset [0,1]^d$ and that it is of the form $\mathcal{V}_l = (a_l, b_l)^d, 0 < a_l < b_l 1$.[4] With this, we can start defining function spaces on $\mathcal{M}$.

### A.2   Hölder Spaces

We start with the manifold versions of the Euclidean Hölder spaces $\mathcal{C}^\gamma(\mathbb{R}^d)$, whose definitions may be found, for instance, in Giné and Nickl [18] and Triebel [42].

**Definition 11** (Hölder spaces). *For all $\gamma > 0$ we define the Hölder space $\mathcal{C}^\gamma(\mathcal{M})$ on the manifold $\mathcal{M}$ to be the space of all $f : \mathcal{M} \to \mathbb{R}$ satisfying*

$$\|f\|_{\mathcal{C}^\gamma(\mathcal{M})} = \sum_{l=1}^{L} \left\|(\chi_l f) \circ \phi_l^{-1}\right\|_{\mathcal{C}^\gamma(\mathbb{R}^d)} < \infty. \tag{17}$$

Since the charts $\phi_l$ are smooth, Definition 11 can be easily seen to be independent of the chosen atlas, with equivalence of norms.

### A.3   Sobolev and Besov Spaces

We now introduce the manifold versions of the Sobolev and Besov spaces, whose definitions in the standard Euclidean case may be found, for instance, in Triebel [42]. For Sobolev spaces we use the Bessel-potential-based definition, following De Vito et al. [13].

**Definition 12** (Sobolev spaces). *For any $s > 0$ we define the Sobolev space $H^s(\mathcal{M})$ on the manifold $\mathcal{M}$ as the Hilbert space of functions $f \in L^2(\mathcal{M})$ such that $\|f\|_{H^s(\mathcal{M})}^2 = \langle f, f \rangle_{H^s(\mathcal{M})} < \infty$ where*

$$\langle f, g \rangle_{H^s(\mathcal{M})} = \sum_{j=0}^{\infty} (1 + \lambda_j)^s \langle f, f_j \rangle_{L^2(\mathcal{M})} \langle g, f_j \rangle_{L^2(\mathcal{M})}. \tag{18}$$

---

[2] Strictly speaking, $L^2(\mathcal{M})$ consists of equivalence classes with respect to the almost everywhere equality.

[3] We can choose $L$ finite by compactness of $\mathcal{M}$.

[4] To see this, take $\tilde{\phi}_l = \exp_{x_l}^{-1}$ and define $\phi_l = T_l \circ \tilde{\phi}_l$ where $T_l$ is an appropriate affine transformation. We can assume that $\mathcal{V}_l = (a_l, b_l)^d$ by positivity of the injectivity radius at $x_l$.

535 **Remark 13.** *It is easy to see that substituting $(1 + \lambda_j)^s$ in Equation (18) with $\beta(\alpha + \lambda_j)^s$ or with*
536 $\alpha + \beta\lambda_j^s$ *for any $\alpha, \beta > 0$ results in the same set of functions and an equivalent norm. The former*
537 *follows from Borovitskiy et al. [8], eq. (109). The latter follows from the Binomial Theorem.*

538 For Besov spaces we follow Coulhon et al. [11] and Castillo et al. [9] and define them in terms of
539 approximations by low-frequency functions. We fix a function $\Phi \in \mathcal{C}^\infty(\mathbb{R}_+, \mathbb{R}_+)$ such that $K =$
540 $\mathrm{supp}(\Phi) \subset [0, 2]$ and $\Phi(x) = 1$ for $x \in [0, 1]$. We also define functions $\Phi_j$ by $\Phi_j(x) = \Phi(2^{-j}x)$.

541 Coulhon et al. [11], Corollary 3.6 shows that the operators $\Phi_j(\sqrt{\Delta})$ defined by functional calculus—
542 discussed, for instance, in Borovitskiy et al. [8]—are bounded in the space $L^p(\mathcal{M})$ for all $1 \leq p \leq$
543 $\infty$.[5] Moreover, it shows that $f = \lim_{j \to \infty} \Phi_j(\sqrt{\Delta})f$ in $L^p(\mathcal{M})$ for every $f \in L^p(\mathcal{M})$. $\Phi_j(\sqrt{\Delta})f$
544 can intuitively be considered as a version of $f$ filtered by a low-pass filter. More explicitly we can
545 write

$$\Phi_j\left(\sqrt{\Delta}\right)f = \sum_{j \geq 0} \Phi\left(\sqrt{\lambda_j}\right)\langle f_j, f\rangle f_j \tag{19}$$

546 which is indeed a filtered version of $f$ as $\Phi$ has compact support. The next definition introduces the
547 Besov spaces $B_{p,q}^s(\mathcal{M})$, which are formulated in terms of quality-of-approximation by low-frequency
548 functions.

549 **Definition 14** (Besov spaces). *For any $s > 0$ and $1 \leq p, q \leq \infty$ we define the Besov space $B_{p,q}^s(\mathcal{M})$*
550 *on the manifold $\mathcal{M}$ as the space of functions $f \in L^p(\mathcal{M})$ such that $\|f\|_{B_{p,q}^s(\mathcal{M})} < \infty$ where*

$$\|f\|_{B_{p,q}^s(\mathcal{M})} = \begin{cases} \|f\|_{L^p} + \left(\sum_{j \geq 0}\left(2^{js}\|\Phi_j(\sqrt{\Delta})f - f\|_{L^p}\right)^q\right)^{1/q} & \text{if } q < +\infty \\ \|f\|_{L^p} + \sup_{j \geq 0} 2^{js}\|\Phi_j(\sqrt{\Delta})f - f\|_{L^p} & \text{if } q = +\infty. \end{cases} \tag{20}$$

551 It turns out that $B_{2,2}^s(\mathcal{M})$ coincide with the Sobolev spaces $H^s(\mathcal{M})$, in the sense that they define
552 the same set of functions and equivalent norms. The same is known for Besov and Sobolev spaces
553 on $\mathbb{R}^d$—see for instance Giné and Nickl [18] section 4.3.6—and even on manifolds if one follows
554 the construction of Triebel [42], pages 7.3–7.4 for Besov spaces. Since our definition is somewhat
555 non-standard, we present the proof.

556 **Proposition 15.** *For all $s > 0$, $H^s(\mathcal{M}) = B_{2,2}^s(\mathcal{M})$ as sets and there exist two constants $C_1, C_2 > 0$*
557 *such that for all $f \in H^s(\mathcal{M}) = B_{2,2}^s(\mathcal{M})$ we have*

$$C_1\|f\|_{H^s(\mathcal{M})} \leq \|f\|_{B_{2,2}^s(\mathcal{M})} \leq C_2\|f\|_{H^s(\mathcal{M})}. \tag{21}$$

558 *Proof.* It is enough to prove (21), the rest will follow automatically. The main technical tools used in
559 the proof are Result 10 and summation by parts. Let $K = \mathrm{supp}(\Phi)$. For the upper bound, notice that

$$\|f\|_{B_{2,2}^s(\mathcal{M})}^2 = \sum_{j \geq 0} 2^{2js}\left\|\Phi_j\left(\sqrt{\Delta}\right)f - f\right\|_{L^2(\mathcal{M})}^2 \tag{22}$$

$$= \sum_{j \geq 0} 2^{2js} \sum_{l: \sqrt{\lambda_l} \notin 2^j K} |\langle f_l, f\rangle_2|^2 \tag{23}$$

$$\leq \sum_{j \geq 0} 2^{2js} \sum_{l: \sqrt{\lambda_l} > 2^j} |\langle f_l, f\rangle_2|^2. \tag{24}$$

560 The last inequality results from the fact that $[0, 1] \subset K$. By Weyl's law Result 10 there exists a
561 constant $c > 0$ such that $\lambda_l \leq cl^{2/d}$. Without loss of generality we can assume that $c = 2^{2r}, r \in \mathbb{N}$.
562 Since $\sqrt{\lambda_l} > 2^j$ implies $l > 2^{d(j-r)}$ we have

$$\|f\|_{B_{2,2}^s(\mathcal{M})}^2 \leq \sum_{j \geq 0} 2^{2js} \sum_{l > 2^{d(j-r)}} |\langle f_l, f\rangle_2|^2 \tag{25}$$

$$= \sum_{j \leq r} 2^{2js} \sum_{l > 2^{d(j-r)}} |\langle f_l, f\rangle_2|^2 + \sum_{j > r} 2^{2js} \sum_{l > 2^{d(j-r)}} |\langle f_l, f\rangle_2|^2 \tag{26}$$

$$\leq r2^{2rs}\|f\|_{L^2(\mathcal{M})}^2 + 2^{2rs} \sum_{j \geq 0} 2^{2js} \sum_{l > 2^{dj}} |\langle f_l, f\rangle_2|^2. \tag{27}$$

---

[5]The space $L^p(\mathcal{M})$ is the Banach space of functions (or rather their equivalence classes) that are integrable
when raised to the power $p$, see for instance Triebel [41] for details on these spaces.

563 Now let $R_j = \sum_{l>2^{dj}}|\langle f_l, f\rangle_2|^2$ and $S_J = \sum_{j=0}^{J} 2^{2js} \leq \frac{2^{2s}}{2^{2s}-1}2^{2Js}, S_{-1} = 0$. Write

$$\sum_{j\geq 0} 2^{2js}\sum_{l>2^{dj}}|\langle f_l, f\rangle_2|^2 = \sum_{j\geq 0}(S_j - S_{j-1})R_j \tag{28}$$

$$= \sum_{j\geq 0} S_j(R_j - R_{j+1}) - S_0 R_1 \tag{29}$$

$$\leq \sum_{j\geq 1} S_j(R_j - R_{j+1}) \tag{30}$$

$$= \sum_{j\geq 0} S_j \sum_{2^{dj}<l\leq 2^{(j+1)d}}|\langle f_l, f\rangle_2|^2 \tag{31}$$

$$\leq \frac{2^{2s}}{2^{2s}-1}\sum_{j\geq 0} 2^{2js}\sum_{2^{dj}<l\leq 2^{(j+1)d}}|\langle f_l, f\rangle_2|^2 \tag{32}$$

$$\leq \frac{2^{2s}}{2^{2s}-1}\sum_{j\geq 0}\sum_{2^{dj}<l\leq 2^{(j+1)d}}l^{2s/d}|\langle f_l, f\rangle_2|^2 \tag{33}$$

$$\leq \frac{c'^s 2^{2s}}{2^{2s}-1}\sum_{j\geq 0}\sum_{2^{dj}<l\leq 2^{(j+1)d}}\lambda_l^s|\langle f_l, f\rangle_2|^2 \tag{34}$$

$$= \frac{c'^s 2^{2s}}{2^{2s}-1}\sum_{l>2^d}\lambda_l^s|\langle f_l, f\rangle_2|^2 \tag{35}$$

$$\leq \frac{c'^s 2^{2s}}{2^{2s}-1}\sum_{l\geq 0}\lambda_l^s|\langle f_l, f\rangle_2|^2. \tag{36}$$

564 Where we have used Result 10 to get existence of $c'$ such that $l^{2/d} \leq c'\lambda_l$. This proves the upper
565 bound with $C_2 = r2^{2rs}\left(1 + \frac{c'^s 2^{2s}}{2^{2s}-1}\right)$. The proof for the lower bound is similar. $\qquad\square$

566 Proposition 15 provides a characterization of the Sobolev spaces $H^s(\mathcal{M})$. There is yet another charac-
567 terization of these spaces that will be useful later, in terms of charts. We present this characterization
568 as part of the following result.

569 **Theorem 16.** *On the Sobolev space $H^s(\mathcal{M})$, the following norms are equivalent:*

$$\|f\|_{H^s(\mathcal{M})} = \left(\sum_{j=0}^{\infty}(1+\lambda_j)^s\langle f, f_j\rangle_{L^2(\mathcal{M})}^2\right)^{1/2} \tag{37}$$

$$\|f\|_{B_{2,2}^s(\mathcal{M})} = \|f\|_{L^2} + \left(\sum_j\left(2^{js}\|\Phi_j(\sqrt{\Delta})f - f\|_{L^2(\mathcal{M})}\right)^2\right)^{1/2} \tag{38}$$

$$\|f\|_{H_{\mathcal{T}}^s(\mathcal{M})} = \left(\sum_{l=1}^{L}\|(\chi_l f)\circ \phi_l^{-1}\|_{H^s(\mathbb{R}^d)}^2\right)^{1/2} \tag{39}$$

570 *Proof.* The equivalence between $\|\cdot\|_{H^s(\mathcal{M})}$ and $\|f\|_{B_{2,2}^s(\mathcal{M})}$ is given by Proposition 15. The equiva-
571 lence between $\|\cdot\|_{H^s(\mathcal{M})}$ and $\|f\|_{H_{\mathcal{T}}^s(\mathcal{M})}$ is proved in De Vito et al. [13]. $\qquad\square$

## A.4 Gaussian Random Elements

573 Here we recall the definition of a Gaussian process as a Banach-space-valued random variable,
574 following for instance van Zanten and van der Vaart [49].

575 **Definition 17** (Gaussian random element)**.** *Let $(\mathbb{B}, \|\cdot\|_{\mathbb{B}})$ be a Banach space, and $f$ be a Borel*
576 *random variable with values in $\mathbb{B}$ almost surely. We say that $f$ is a Gaussian random element if $b^*(f)$*
577 *is a univariate Gaussian random variable for every bounded linear functional $b^*$ on $\mathbb{B}$.*

Random variables of this kind are also sometimes called *Gaussian in the sense of duality*. One should think of a Gaussian random element as a generalization of a Gaussian process, but which is better-behaved from a function-analytic point of view and in particular does not require the process to be an actual function—as opposed to, for instance, a distribution. Many connections between the usual Gaussian processes and Gaussian random elements exist, see Lifshits [27], Ghosal and van der Vaart [17], Appendix I, van der Vaart and van Zanten [46] for details. The following observation about Gaussian random elements will be useful later.

**Lemma 18.** *A Gaussian process $f$ on the manifold $\mathcal{M}$ with almost surely continuous sample paths is a Gaussian random element in the Banach space $(\mathcal{C}(\mathcal{M}), \|\cdot\|_\infty)$ of continuous functions on $\mathcal{M}$.*

*Proof.* Since $\mathcal{C}(\mathcal{M})$ is separable, this follows from Lemma I.6 in Ghosal and van der Vaart [17]. $\square$

# B  Technical Lemmas

This section contains the lemmas used in Appendix C. In this section the expression $a \lesssim b$ means $a \leq Cb$ for some constant $C > 0$ whose value is irrelevant for our claims. We start by an upper bound on the metric entropy of Sobolev balls on $\mathcal{M}$ with respect to the uniform norm.

**Lemma 19** (Entropy of Sobolev balls)**.** *For all $s > 0$ let $H_1^s = \left\{ f \in H^s(\mathcal{M}) : \|f\|_{H^s(\mathcal{M})} \leq 1 \right\}$. Define the $\varepsilon$-covering number of $H_1^s$ with respect to the norm $\|\cdot\|_{L^\infty(\mathcal{M})}$ by*

$$N\left(\varepsilon, H_1^s, \|\cdot\|_{L^\infty(\mathcal{M})}\right) = \arg\min_{J \in \mathbb{N}} \left\{ \exists h_1, .., h_J \in H_1^s : H_1^s \subset \bigcup_{j=1}^J B\left(h_j, \varepsilon, \|\cdot\|_{L^\infty(\mathcal{M})}\right) \right\} \quad (40)$$

*where $B\left(h_j, \varepsilon, \|\cdot\|_{L^\infty(\mathcal{M})}\right)$ stands for the $\|\cdot\|_{L^\infty(\mathcal{M})}$ ball with center $h_j$ and radius $\varepsilon$.*

*For any $\nu > 0$, there exist $C, \varepsilon_0 > 0$ such that for every $\varepsilon \leq \varepsilon_0$*

$$\ln N\left(\varepsilon, H_1^{\nu+d/2}, \|\cdot\|_{L^\infty(\mathcal{M})}\right) \leq C\varepsilon^{-\frac{d}{\nu+d/2}}, \quad (41)$$

*where the left-hand side of the inequality above, as a function of $\varepsilon$, is called the METRIC ENTROPY of the Sobolev ball $H_1^{\nu+d/2}$ with respect to the uniform norm $\|\cdot\|_{L^\infty(\mathcal{M})}$.*

*Proof.* Using the charts we will reduce the problem to the entropy of the unit ball of the Sobolev space $H^{\nu+d/2}\left([0,1]^d\right)$ for which the upper bound is known. Take $f \in H_1^{\nu+d/2}$ and look for an approximation of $f$ by $\tilde{f}$ of the form

$$\tilde{f} = \sum_{l=1}^L \chi_l(h_l \circ \phi_l) \quad (42)$$

for some functions $h_l : \mathcal{V}_l \to \mathbb{R}$ where $\mathcal{V}_l \subseteq \mathbb{R}^d$. We have

$$\left\|f - \tilde{f}\right\|_{L^\infty(\mathcal{M})} = \left\|\sum_{l=1}^L \chi_l(h_l \circ \phi_l - f)\right\|_{L^\infty(\mathcal{M})} \leq \sum_{l=1}^L \left\|\chi_l(h_l \circ \phi_l - f)\right\|_{L^\infty(\mathcal{U}_l)} \quad (43)$$

$$\leq \sum_{l=1}^L \left\|h_l \circ \phi_l - f\right\|_{L^\infty(\mathcal{U}_l)} \leq \sum_{l=1}^L \left\|h_l - f \circ \phi_l^{-1}\right\|_{L^\infty(\mathcal{V}_l)} \quad (44)$$

$$\leq L \max_{1 \leq l \leq L} \left\|h_l - f \circ \phi_l^{-1}\right\|_{L^\infty([0,1]^d)}. \quad (45)$$

This means that to approximate $f$ by $\tilde{f}$ uniformly on $\mathcal{M}$ we need to choose the functions $h_l$ that approximate $f \circ \phi_l^{-1}$ well with respect to the uniform norm on $[0,1]^d$.

Next, we show that the functions $f \circ \phi_l^{-1}$ are contained in an Euclidean Sobolev ball of radius $R$, with $R$ depending only on $\nu$ and the atlas. We use Große and Schneider [21], Lemma 2.1[6] to get

---

[6]Importantly, also the remark just above Große and Schneider [21], Lemma 2.1, that allows us to consider Besov spaces $B_{2,2}^s$ coinciding with the Sobolev spaces $H^s$ instead of the Besov spaces $B_{2,\infty}^s$.

from the second line to the third, and $R$ is the constant hidden behind the notation $\lesssim$ in the last line.

$$\left\| f \circ \phi_l^{-1} \right\|_{H^s([0,1]^d)} = \left\| \sum_{l'=1}^{L} (\chi_{l'} f) \circ \phi_l^{-1} \right\|_{H^s([0,1]^d)} \leq \sum_{l'=1}^{L} \left\| (\chi_{l'} f) \circ \phi_l^{-1} \right\|_{H^s([0,1]^d)} \tag{46}$$

$$= \sum_{l'=1}^{L} \left\| (\chi_{l'} f) \circ \phi_{l'}^{-1} \circ \phi_{l'} \circ \phi_l^{-1} \right\|_{H^s([0,1]^d)} \tag{47}$$

$$\lesssim \sum_{l'=1}^{L} \left\| (\chi_{l'} f) \circ \phi_{l'}^{-1} \right\|_{H^s([0,1]^d)} \lesssim \|f\|_{H^s(\mathcal{M})}. \tag{48}$$

Without loss of generality we assume $R = 1$. By the Euclidean counterpart [18, Theorem 4.3.36] of the result we are proving, we have

$$\ln N\left( \varepsilon, H_1^{\nu+d/2}([0,1]^d), \|\cdot\|_{L^\infty([0,1]^d)} \right) \lesssim \varepsilon^{-\frac{d}{\nu+d/2}}. \tag{49}$$

Let $h_1, .., h_J \in H_1^{\nu+d/2}$ be such that $H_1^{\nu+d/2}([0,1]^d) \subset \cup_{j=1}^{J} B\left( h_k, \varepsilon/L, \|\cdot\|_{L^\infty([0,1]^d)} \right)$. Then for any $f \in H_1^{\nu+d/2}$ there exists a sequence $\{j_l\}_{l=1}^{L} \subseteq \{1, .., J\}$ such that

$$\left\| f - \sum_{l=1}^{L} \chi_l (h_{j_l} \circ \phi_l) \right\|_{L^\infty(\mathcal{M})} < L\frac{\varepsilon}{L} = \varepsilon. \tag{50}$$

This shows that $N\left( \varepsilon, H_1^s, \|\cdot\|_{L^\infty(\mathcal{M})} \right) \leq LJ$, where $L$ is just the number of charts, proving the claim.

$\square$

For the related *diffusion spaces* [13], the RKHS corresponding to the heat (diffusion) kernels, Castillo et al. [9] uses the results of Coulhon et al. [11] to bound the entropy in terms of a wavelet frame instead of relying on charts. We believe this alternative proof scheme should work in our case as well. However, we could not, to the best of our effort, get a tight enough bound for the Sobolev spaces by directly using the results of Coulhon et al. [11] and therefore we chose to rely on charts instead.

The next two theorems will be useful to characterize the RKHS of the extrinsic Matérn process on $\mathcal{M}$. We start by a lemma relating the RKHS of the restriction of a Gaussian process to the original one.

**Lemma 20.** *Assume that $k$ is a kernel on $\mathbb{R}^d$, $f \sim \mathrm{GP}(0, k)$ with almost surely continuous sample paths and $\widetilde{\mathbb{H}}$ is the RKHS of $k$. If $\mathcal{M} \subseteq \mathbb{R}^d$ is a submanifold, then the RKHS $\mathbb{H}$ corresponding to the restricted process $f_{|\mathcal{M}}$ is the set of all restrictions $g_{|\mathcal{M}}$ of functions $g \in \widetilde{\mathbb{H}}$ equipped with the norm*

$$\|h\|_{\mathbb{H}} = \inf_{g \in \widetilde{\mathbb{H}},\ g_{|\mathcal{M}} = h} \|g\|_{\widetilde{\mathbb{H}}}. \tag{51}$$

*Moreover there always exists an element $g \in \widetilde{\mathbb{H}}$ such that $g_{|\mathcal{M}} = f$ and $\|g\|_{\widetilde{\mathbb{H}}} = \|f\|_{\mathbb{H}}$.*

*Proof.* Lemma 5.1 in Yang and Dunson [53]. $\square$

The last result will be used to characterize the RKHS of the extrinsic Matérn Gaussian processes using trace and extension operators. The second ingredient for this is the following.

**Theorem 21.** *If $s > \frac{D-d}{2}$ then the restriction operator extends to a bounded linear map $\mathrm{Tr}_s :$ $H^s(\mathbb{R}^D) \to H^{s-\frac{D-d}{2}}(\mathcal{M})$. Moreover, for every $u > 0$ there exists a bounded right inverse $\mathrm{Ex}_u : H^u(\mathcal{M}) \to H^{u+\frac{D-d}{2}}(\mathbb{R}^D)$ such that $\mathrm{Tr}_{u+\frac{D-d}{2}} \circ \mathrm{Ex}_u = I_{H^u(\mathcal{M})}$.*

*Proof.* Theorem 4.10 in Große and Schneider [21]. $\square$

The last two results allow us to characterize the RKHS of the extrinsic Matérn process on $\mathcal{M}$.

**Proposition 22.** *The RKHS $\mathbb{H}$ of a restricted extrinsic Matérn process $f$ with smoothness parameter $\nu$ on $\mathcal{M}$ is norm equivalent to the Sobolev space $H^{\nu+d/2}(\mathcal{M})$.*

*Proof.* Using Lemma 20, the RKHS $\mathbb{H}$ can be characterized as the set of functions $f : \mathcal{M} \to \mathbb{R}$ that are the restrictions of some $g \in \widetilde{\mathbb{H}}$, where $\widetilde{\mathbb{H}}$ is the RKHS of the ambient Matérn process $\tilde{f}$, with

$$\|f\|_{\mathbb{H}} = \inf_{g \in \widetilde{\mathbb{H}}, \ g_{|\mathcal{M}} = f} \|g\|_{\widetilde{\mathbb{H}}}. \tag{52}$$

Since $\widetilde{\mathbb{H}}$ is norm-equivalent to the Sobolev space[7] $H^{\nu+D/2}(\mathbb{R}^D)$ (see the appendix in Borovitskiy et al. [8]), by the trace and extension theorem Theorem 21 for every $f \in \mathbb{H}$

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

In order to extend convergence rates results with respect to the empirical $L^2$-norm to convergence rates with respect to the full $L^2$-norm, we need to show regularity properties of the prior process' sample paths. Kolmogorov's continuity criterion is a standard tool in probability theory to show that a given stochastic process has a Hölder continuous version: we re-prove it here because we will need a form of the result which gives explicit control of the Hölder norms, which is not usually included in the statement of the theorem.

In the following, if $h$ is a random variable under the probability measure $\Pi$, we define

$$\Pi[h] = \int h \Pi(dh) \tag{60}$$

for the expectation of $h$ with respect to $\Pi$, assuming integrability.

**Lemma 25** (Kolmogorov's continuity criterion). *If $g \sim \Pi$ is a zero mean Gaussian process on $[0,1]^d$*

$$\Pi\Big[|g(x) - g(y)|^2\Big] \leq C\|x - y\|^{2\rho} \tag{61}$$

*for some $0 < \rho \leq 1$ and $C > 0$, then there exists a version of $g$ with samples paths in $\mathcal{C}^\alpha\left([0,1]^d\right)$ for every $0 < \alpha < \rho$. Moreover for every $\alpha < \rho$ this version satisfies $\Pi\Big[\|g\|^2_{\mathcal{C}^\alpha([0,1]^d)}\Big] \leq C'$ where $C' < +\infty$ depends only on $C$, $\rho$ and $\alpha$.*

*Proof.* Take $x, y \in [0,1]$, $M > 0$ and $q \in \mathbb{N}$. Since the random variable $g(x) - g(y)$ is Gaussian we have

$$\Pi\Big[|g(x) - g(y)|^{2q}\Big] = \frac{(2q)!}{2^q q!}\Pi\Big[|g(x) - g(y)|^2\Big]^q \leq C_q\|x - y\|^{2\rho q} \tag{62}$$

where $C_q := C^q \frac{(2q)!}{2^q q!}$. We consider the $2q$-th power for a reason that will become clear later in the proof. Therefore by Markov's inequality for every $x, y \in [0,1]^d$ we have

$$\Pi[|g(x) - g(y)| > u] \leq C_q u^{-2q}\|x - y\|^{2q\rho} \tag{63}$$

Now take $X = \cup_{k\geq 1}X_k$, $X_k = 2^{-k}\mathbb{Z}^d \cap [0,1]^d$. Then the previous inequality applied to any $x, y \in X_k$ adjacent, where we see $X_k$ as a graph where two vertices are connected if they differ by at most one coordinate, and $u = M2^{-k\alpha}$ implies

$$\Pi\Big[|g(x) - g(y)| > M2^{-k\alpha}\Big] \leq C_q M^{-2q}2^{-2kq(\rho-\alpha)} \tag{64}$$

Summing over $k \geq 1$ and adjacent points in $X$—and there are at most $C2^{kd}$ of them where $C > 0$ is an absolute constant—gives us for $q > \frac{d}{2(\rho-\alpha)}$, where we may take $q = \frac{d}{(\rho-\alpha)}$, that

$$\Pi[\exists x, y \in X, x, y \text{ adjacent}, |g(x) - g(y)| > M\|x - y\|^\alpha] \tag{65}$$

$$\leq \sum_{k\geq 1} \sum_{x,y\in X \text{ adjacent}} \Pi\big[|g(x) - g(y)| > M2^{-k\alpha}\big] \tag{66}$$

$$\leq C \sum_{k\geq 1} 2^{kd} C_q M^{-2q} 2^{-2kq(\rho-\alpha)} = \frac{CC_q}{2^{2q(\rho-\alpha)-d} - 1} M^{-2q}. \tag{67}$$

In particular for all $q > \max\left(1, \frac{d}{2(\rho-\alpha)}\right)$ we have

$$\Pi\left[\left(\sup_{x,y\in X \text{ adjacent}} \frac{|g(x) - g(y)|}{\|x - y\|^\alpha}\right)^2\right] \leq 2 + 2\int_1^\infty M\Pi\left[\sup_{x,y\in X \text{ adjacent}} \frac{|g(x) - g(y)|}{\|x - y\|^\alpha} > M\right] \mathrm{d}M \tag{68}$$

$$\leq C_{C,\alpha,\rho} \tag{69}$$

for some constant $C_{C,\alpha,\rho} < +\infty$. In particular $K = \sup_{x,y\in X \text{ adjacent}} \frac{|g(x)-g(y)|}{\|x-y\|^\alpha}$ is finite almost surely. Since $X$ is dense in $[0,1]^d$ and $g$ is almost surely uniformly continuous on $X$, $g$ admits a unique continuous extension to $[0,1]^d$ on an almost sure event $\mathcal{A}$. Let us define

$$\forall x \in [0,1]^d, \overline{g}(x) = \begin{cases} \lim_{y\to x, y\in X} g(y) \text{ on } \mathcal{A} \\ 0 \text{ otherwise} \end{cases} \tag{70}$$

For any $x, y \in [0,1]^d$ and $x_n \to x, y_n \to y, x_n, y_n \in X$ we have

$$|\overline{g}(x) - \overline{g}(y)| \leq \liminf_{n\to\infty} |\overline{g}(x) - \overline{g}(x_n)| + |\overline{g}(x_n) - \overline{g}(y_n)| + |\overline{g}(y_n) - \overline{g}(y)| \tag{71}$$

$$\leq \liminf_{n\to\infty} |\overline{g}(x) - \overline{g}(x_n)| + K\|x_n - y_n\|^\alpha + |\overline{g}(y_n) - \overline{g}(y)| \tag{72}$$

$$= K\|x - y\|^\alpha \tag{73}$$

Hence $\overline{g}$ is $\alpha$-Hölder continuous on $[0,1]^d$ with the same constant $K$ and, using $(a+b)^2 \leq 2(a^2+b^2)$ that is valid for every $a, b > 0$, we have

$$\Pi\left[\|\overline{g}\|^2_{\mathcal{C}^\alpha([0,1]^d)}\right] \leq 2\Pi\left[\left(\sup_{x\in X} g(x)^2\right)\right] + 2\Pi\left[\left(\sup_{x,y\in X \text{ adjacent}} \frac{|g(x) - g(y)|}{\|x - y\|^\alpha}\right)^2\right] \tag{74}$$

$$\leq 2\Pi\left[(|g(0)| + K)^2\right] + 2\Pi\left[\left(\sup_{x,y\in X \text{ adjacent}} \frac{|g(x) - g(y)|}{\|x - y\|^\alpha}\right)^2\right] \tag{75}$$

$$\leq 4\Pi\left[g(0)^2 + K^2\right] + 2\Pi\left[\left(\sup_{x,y\in X \text{ adjacent}} \frac{|g(x) - g(y)|}{\|x - y\|^\alpha}\right)^2\right] \tag{76}$$

$$\leq C_{C,\alpha,\rho} < +\infty \tag{77}$$

where for the last inequality we have also used $g(0) \in L^2$. Moreover $\overline{g}$ is a version of $g$: for all $x \in [0,1]^d$ we have by definition $\lim_{y\in X, y\to x} g(y) = \overline{g}(y)$ almost surely, and $\Pi\left[|g(x) - g(y)|^2\right] \leq C\|x - y\|^{2\rho} \to 0$ as $y \to x, y \in X$, hence the uniqueness of the limit in probability implies that for all $x \in [0,1]^d$ $\overline{g}(x) = g(x)$ almost surely, ie that $\overline{g}$ is a version of $g(x)$. Finally, if $\alpha < \alpha' < \rho$, then since the two versions corresponding to $\alpha$ and $\alpha'$ are continuous, they must be indistinguishable. $\square$

**Remark 26.** *We see in the last proof that we can replace $\Pi\left[\|g\|^2_{\mathcal{C}^\alpha([0,1]^d)}\right] \leq C_{C,\alpha,\rho}$ in the statement by $\Pi\left[\|g\|^r_{\mathcal{C}^\alpha([0,1]^d)}\right] \leq C'_{C,\alpha,\rho,r}$ for any $r > 0$, even though we will only use $r = 2$ in the following.*

The next lemma applies our version of Kolmogorov's criterion, Lemma 25, to the intrinsic Matérn processes on $\mathcal{M}$ by considering charts. Another idea would be to use Driscoll's Theorem—given in Kanagawa et al. [23], Theorem 4.9—and the Sobolev embedding theorem—De Vito et al. [13], Theorem 4—but that would only give us that the sample paths are almost surely in $\mathcal{C}^\gamma(\mathcal{M})$ for every $0 < \gamma < \nu - d/2, \gamma \notin \mathbb{N}$, whereas here we improve the range of index to $\gamma < \nu$. As we will see in Appendix C, we need to ensure that this property holds somewhat uniformly with respect to the truncation parameter, which is why we tracked the constants in our proof of Kolmogorov's criterion. As we will see, the main difficulty in the proof of the next result will be to tackle the case of regularity strictly larger than 1.

**Lemma 27.** *Let $f \sim \Pi_n$ be an intrinsic Matérn process with smoothness parameter $\nu > 0$ truncated at $J_n \in \mathbb{N} \cup \{\infty\}$. Then for every $\gamma < \nu$ we have*

$$\sup_n \Pi_n \left[ \|f\|^2_{\mathcal{C}^\gamma(\mathcal{M})} \right] < \infty. \tag{78}$$

*Proof.* We start by the case $\nu \leq 1$. Take $1 \leq l \leq L$ and define $h_l = (\chi_l f) \circ \phi_l^{-1}$. Then $h_l$ is a Gaussian process with covariance kernel given by

$$\forall x, y \in \mathcal{V}_l, \tilde{K}(x, y) = \chi_l \circ \phi_l^{-1}(x) K(x, y) \chi_l \circ \phi_l^{-1}(y) \tag{79}$$

where $K(x, y) = \Pi_n \left[ \left( f \circ \phi_l^{-1}(x) \right) \left( f \circ \phi_l^{-1} \right)(y) \right]$ is the covariance kernel of $f$. This has an RKHS that we denote $\widetilde{\mathbb{H}}$. The goal is to apply Lemma 25 to $h_l$. For all $x, y \in \mathcal{V}_l$, where we recall that we can assume that $\mathcal{V}_l = (a_l, b_l), 0 < a_l < b_l < 1$, we have

$$\Pi_n \left[ |h_l(x) - h_l(y)|^2 \right] = \tilde{K}(x, x) + \tilde{K}(y, y) - 2\tilde{K}(x, y) \tag{80}$$

$$= \left\| \tilde{K}(x, \cdot) - \tilde{K}(y, \cdot) \right\|^2_{\widetilde{\mathbb{H}}} \tag{81}$$

$$= \sup_{\|\varphi\|_{\widetilde{\mathbb{H}}}=1} \left| \left\langle \tilde{K}(x, \cdot) - \tilde{K}(y, \cdot), \varphi \right\rangle \right|^2 \tag{82}$$

$$= \sup_{\|\varphi\|_{\widetilde{\mathbb{H}}}=1} |\varphi(x) - \varphi(y)|^2 \tag{83}$$

$$\leq \sup_{\|\varphi\|_{\widetilde{\mathbb{H}}}=1} \|\varphi\|^2_{\mathcal{C}^\nu(\mathcal{V}_l)} \|x - y\|^{2\nu} \tag{84}$$

In order to apply Lemma 25, it suffices to show that we have a continuous embedding $\widetilde{\mathbb{H}} \hookrightarrow \mathcal{C}^\nu(\mathcal{V}_l)$. $\widetilde{\mathbb{H}}$ is by definition the completion of

$$\left\{ \sum_{i=1}^p \alpha_i \tilde{K}(x_i, \cdot) : p \geq 1, \alpha_i \in \mathbb{R}, x_i \in \mathcal{V}_l \right\} \tag{85}$$

$$= \left\{ \sum_{i=1}^p \alpha_i \left( \chi_l \circ \phi_l^{-1} \right)(x_i) \left( \chi_l \circ \phi_l^{-1} \right)(\cdot) K \left( \phi_l^{-1}(x_i), \phi_l^{-1}(\cdot) \right) : p \geq 1, \alpha_i \in \mathbb{R}, x_i \in \mathcal{V}_l \right\} \tag{86}$$

equipped with the RKHS norm

$$\left\| \sum_{i=1}^p \alpha_i \tilde{K}(x_i, \cdot) \right\|^2_{\widetilde{\mathbb{H}}} = \sum_{i,j=1}^p \alpha_i \alpha_j \left( \chi_l \circ \phi_l^{-1} \right)(x_i) \left( \chi_l \circ \phi_l^{-1} \right)(x_j) K \left( \phi_l^{-1}(x_i), \phi_l^{-1}(x_j) \right) \tag{87}$$

Hence by definition of the Sobolev space $H^{\nu+d/2}(\mathcal{M})$ and the equality $\|\cdot\|_{\mathbb{H}} = \|\cdot\|_{H^{\nu+d/2}(\mathcal{M})}$ on $\mathbb{H}$ we have

$$\left\| \sum_{i=1}^{p} \alpha_i \tilde{K}(x_i, \cdot) \right\|_{H^{\nu+d/2}(\mathbb{R}^d)}^2 \tag{88}$$

$$= \left\| \sum_{i=1}^{p} \alpha_i \big(\chi_l \circ \phi_l^{-1}\big)(x_i)\big(\chi_l \circ \phi_l^{-1}\big)(\cdot) K\big(\phi_l^{-1}(x_i), \phi_l^{-1}(\cdot)\big) \right\|_{H^{\nu+d/2}(\mathbb{R}^d)}^2 \tag{89}$$

$$\leq \left\| \sum_{i=1}^{p} \alpha_i \big(\chi_l \circ \phi_l^{-1}\big)(x_i) K\big(\phi_l^{-1}(x_i), \cdot\big) \right\|_{H^{\nu+d/2}(\mathcal{M})}^2 \tag{90}$$

$$= \left\| \sum_{i=1}^{p} \alpha_i \big(\chi_l \circ \phi_l^{-1}\big)(x_i) K\big(\phi_l^{-1}(x_i), \cdot\big) \right\|_{\mathbb{H}}^2 \tag{91}$$

$$= \sum_{i,j=1}^{p} \alpha_i \alpha_j \big(\chi_l \circ \phi_l^{-1}\big)(x_i)\big(\chi_l \circ \phi_l^{-1}\big)(x_j) K\big(\phi_l^{-1}(x_i), \phi_l^{-1}(x_j)\big) \tag{92}$$

$$= \left\| \sum_{i=1}^{p} \alpha_i \tilde{K}(x_i, \cdot) \right\|_{\widetilde{\mathbb{H}}}^2. \tag{93}$$

Therefore by completion we find a continuous embedding $\widetilde{\mathbb{H}} \hookrightarrow H^{\nu+d/2}(\mathbb{R}^d)$ with $\|\cdot\|_{H^{\nu+d/2}(\mathbb{R}^d)} \leq \|\cdot\|_{\widetilde{\mathbb{H}}}$ on $\widetilde{\mathbb{H}}$. By the Sobolev Embedding Theorem in $\mathbb{R}^d$—see for instance Triebel [41], Section 2.7.1, Remark 2—we have $B_{2,2}^{\nu+d/2}(\mathbb{R}^d) = H^{\nu+d/2}(\mathbb{R}^d) \hookrightarrow \mathcal{C}^\nu(\mathbb{R}^d)$, which implies $\widetilde{\mathbb{H}} \hookrightarrow \mathcal{C}^\nu(\mathbb{R}^d)$ by composition. Therefore there exists a constant $C = C_\nu$ such that

$$\forall x, y \in \mathcal{V}_l, \Pi_n\Big[|h_l(x) - h_l(y)|^2\Big] \leq C\|x-y\|^{2\nu} \tag{94}$$

Hence, by applying Lemma 25 there exists a version $\tilde{h}_l$ of $h_l$ with almost surely $\alpha$-Hölder continuous sample paths for every $\alpha < \nu$. Now consider $\tilde{h} := \sum_{l=1}^{L} \tilde{h}_l \circ \phi_l$. Then $\tilde{h}$ is a version of $h$ because, for all $a \in \mathcal{U}_l$

$$\Pi\Big[h(a) \neq \tilde{h}(a)\Big] = \Pi\left[\sum_{l=1}^{L} h_l(\phi_l(a)) \neq \sum_{l=1}^{L} \tilde{h}_l(\phi_l(a))\right] \tag{95}$$

$$\leq \Pi\Big[\cup_{l=1}^{L}\big\{h_l(\phi_l(a)) \neq \tilde{h}_l(\phi_l(a))\big\}\Big] \tag{96}$$

$$\leq \sum_{l=1}^{L} \Pi\Big[h_l(\phi_l(a)) \neq \tilde{h}_l(\phi_l(a))\Big] \tag{97}$$

$$= 0 \tag{98}$$

the last equality being true from the fact the each $\tilde{h}_l$ is a version of $h_l$. Moreover

$$\Pi\Big[\big\|\tilde{h}\big\|_{\mathcal{C}^\alpha(\mathcal{M})}^2\Big] = \sum_{l=1}^{L} \Pi\Big[\big\|\big(\chi_l \tilde{h}\big) \circ \phi_l^{-1}\big\|_{\mathcal{C}^\alpha(\mathbb{R}^d)}^2\Big] \tag{99}$$

$$\lesssim \max_{l=1}^{L} \Pi\Big[\|h_l\|_{\mathcal{C}^\alpha([0,1]^d)}^2\Big] \tag{100}$$

$$\leq C_{C,\alpha,\nu,\mathcal{T}} \tag{101}$$

still using Lemma 25 and fact that the $\chi_l$ and $\phi_l$ are smooth, hence the additional dependence in $\mathcal{T}$ in the last constant.

We now turn to the general case. The proof will be similar to the one of Ghosal and van der Vaart [17], Proposition I.3 although we need to control the Hölder norms, work through charts and precisely show that the kernel is regular. Assume for simplicity that $d = 1, 1 < \nu \le 2$, otherwise it suffices to introduce coordinates and to proceed by induction on $\lfloor \nu \rfloor$. Let $l \in \{1, \ldots, L\}$, and as before define $\tilde{K}(x, y) = (\chi_l \circ \phi_l^{-1})(x)(\chi_l \circ \phi_l^{-1})(y)K(\phi_l^{-1}(x), \phi_l^{-1}(y))$ the RKHS of $h_l = (\chi_l f) \circ \phi_l^{-1}$ as well as $\widetilde{\mathbb{H}}$ its RKHS.

First, let us construct an $L^2$-derivative $\dot{h}_l$ of $h_l$—where here $L^2 = L^2(\Omega, \mathcal{F}, \mathbb{P})$ with $(\Omega, \mathcal{F}, \mathbb{P})$ the underlying probability space—namely a square integrable process on $\mathcal{V}_l$ such that

$$\Pi\left[\left|\frac{h_l(x+h) - h_l(x)}{h} - \dot{h}_l(x)\right|^2\right] \to 0 \tag{102}$$

as $h \to 0$, for all $x \in \mathcal{V}_l$. For this we will first show that $\frac{\partial \tilde{K}}{\partial x}(x, \cdot) \in \widetilde{\mathbb{H}}$ for every $x \in \mathcal{V}_l$ and that

$$\left\|\frac{\partial \tilde{K}}{\partial x}(x, \cdot) - \frac{\partial \tilde{K}}{\partial x}(x', \cdot)\right\|_{\widetilde{\mathbb{H}}} \le C_\nu |x - x'|^{\nu-1} \tag{103}$$

We first show that $\frac{\tilde{K}(x+h, \cdot) - \tilde{K}(x, \cdot)}{h}$ is a Cauchy net in $\widetilde{\mathbb{H}}$. We have

$$\left\|\frac{\tilde{K}(x+h, \cdot) - \tilde{K}(x, \cdot)}{h} - \frac{\tilde{K}(x+h', \cdot) - \tilde{K}(x, \cdot)}{h'}\right\|_{\widetilde{\mathbb{H}}} \tag{104}$$

$$= \sup_{\|\varphi\|_{\tilde{\mathbb{H}}}=1} \left\langle \frac{\tilde{K}(x+h, \cdot) - \tilde{K}(x, \cdot)}{h} - \frac{\tilde{K}(x+h', \cdot) - \tilde{K}(x, \cdot)}{h'}, \varphi \right\rangle_{\widetilde{\mathbb{H}}} \tag{105}$$

$$= \sup_{\|\varphi\|_{\tilde{\mathbb{H}}}=1} \frac{\varphi(x+h) - \varphi(x)}{h} - \frac{\varphi(x+h') - \varphi(x)}{h'} \tag{106}$$

$$= \sup_{\|\varphi\|_{\tilde{\mathbb{H}}}=1} \int_0^1 [\varphi'(x+th) - \varphi'(x+th')]\, dt \tag{107}$$

$$\le \sup_{\|\varphi\|_{\tilde{\mathbb{H}}}=1} \|\varphi'\|_{\mathcal{C}^{\nu-1}(\mathcal{V}_l)} |h - h'|^{\nu-1} \tag{108}$$

$$\le \sup_{\|\varphi\|_{\tilde{\mathbb{H}}}=1} \|\varphi\|_{\mathcal{C}^\nu(\mathcal{V}_l)} |h - h'|^{\nu-1} \tag{109}$$

As in the case $\nu \le 1$, we can show show that $\widetilde{\mathbb{H}} \hookrightarrow \mathcal{C}^\nu(\mathbb{R}^d)$. This implies that for a constant $C = C_\nu$

$$\left\|\frac{\tilde{K}(x+h, \cdot) - \tilde{K}(x, \cdot)}{h} - \frac{\tilde{K}(x+h', \cdot) - \tilde{K}(x, \cdot)}{h'}\right\|_{\widetilde{\mathbb{H}}} \le C|h - h'|^{\nu-1} \tag{110}$$

As $|h - h'|^{\nu-1} \to 0$ when $h, h' \to 0$, because $\nu > 1$, this proves that $\frac{\tilde{K}(x+h, \cdot) - \tilde{K}(x, \cdot)}{h}$ is a Cauchy net in $\widetilde{\mathbb{H}}$: by completeness of $\widetilde{\mathbb{H}}$ it converges in $\widetilde{\mathbb{H}}$ to a limit $g$. Since convergence in $\widetilde{\mathbb{H}}$ implies pointwise convergence by the general properties of RKHSs, the limit $g$ satisfies

$$\forall y, g(y) = \lim_{h \to 0} \frac{\tilde{K}(x+h, y) - \tilde{K}(x, y)}{h} = \frac{\partial \tilde{K}}{\partial x}(x, y) \tag{111}$$

Hence the partial derivative $\frac{\partial \tilde{K}}{\partial x}(x, y)$ exists for all $y$ and $g = \frac{\partial \tilde{K}}{\partial x}(x, \cdot) \in \widetilde{\mathbb{H}}$. Moreover, by the isometry $h_l(x) \in L^2 \mapsto \Pi[h_l(x)h_l(\cdot)] = \tilde{K}(x, \cdot) \in \widetilde{\mathbb{H}}$, we deduce that $h_l$ is actually $L^2$-differentiable, with an $L^2$-derivative denoted as $\dot{h}_l$, and that the derivative process $\dot{h}_l$ is Gaussian, as it is an $L^2$ limit of Gaussian random variables, satisfying $\Pi\left[\dot{h}_l(x)\dot{h}_l(y)\right] = \left\langle \frac{\partial \tilde{K}}{\partial x}(x, \cdot), \frac{\partial \tilde{K}}{\partial x}(y, \cdot)\right\rangle_{\widetilde{\mathbb{H}}}$.

Having established the existence of an $L^2$-derivative $\dot{h}_l$ of the process $h_l$, we would like now to show that $\dot{h}_l$ possesses a $(\gamma - 1)$-regular version for every $\gamma < \nu$. For this, we would like to apply Lemma 25 to $\dot{h}_l$.

758 For this notice that, still by isometry, for all $h > 0$

$$\Pi\left[\left|\dot{h}_l(x) - \dot{h}_l(y)\right|^2\right] = \left\|\frac{\partial \tilde{K}}{\partial x}(x', \cdot) - \frac{\partial \tilde{K}}{\partial x}(x, \cdot)\right\|_{\widetilde{\mathbb{H}}}^2 \tag{112}$$

$$\leq 3\left\|\frac{\tilde{K}(x'+h, \cdot) - \tilde{K}(x', \cdot)}{h} - \frac{\partial \tilde{K}}{\partial x}(x', \cdot)\right\|_{\widetilde{\mathbb{H}}}^2 \tag{113}$$

$$+ 3\left\|\frac{\tilde{K}(x+h, \cdot) - \tilde{K}(x, \cdot)}{h} - \frac{\partial \tilde{K}}{\partial x}(x, \cdot)\right\|_{\widetilde{\mathbb{H}}}^2 \tag{114}$$

$$+ 3\left\|\frac{\tilde{K}(x+h, \cdot) - \tilde{K}(x, \cdot)}{h} - \frac{\tilde{K}(x'+h, \cdot) - \tilde{K}(x', \cdot)}{h}\right\|_{\widetilde{\mathbb{H}}}^2 \tag{115}$$

759 Therefore by the same arguments as above, we have

$$\Pi\left[\left|\dot{h}_l(x) - \dot{h}_l(y)\right|^2\right]^{1/2} = \left\|\frac{\partial \tilde{K}}{\partial x}(x', \cdot) - \frac{\partial \tilde{K}}{\partial x}(x, \cdot)\right\|_{\widetilde{\mathbb{H}}} \tag{116}$$

$$\leq \liminf_{h \to 0}\left\|\frac{\tilde{K}(x+h, \cdot) - \tilde{K}(x, \cdot)}{h} - \frac{\tilde{K}(x'+h, \cdot) - \tilde{K}(x', \cdot)}{h}\right\|_{\widetilde{\mathbb{H}}} \tag{117}$$

$$\leq \liminf_{h \to 0} \sup_{\|\varphi\|_{\widetilde{\mathbb{H}}}=1} \int_0^1 |\varphi'(x+th) - \varphi'(x'+th)|\,\mathrm{d}t \tag{118}$$

$$\leq \liminf_{h \to 0} C_\nu |x - x'|^{\nu-1} \tag{119}$$

$$= C_\nu |x - x'|^{\nu-1} \tag{120}$$

760 Therefore we can now apply Lemma 25 to $\dot{h}_l$ and find a version $\tilde{h}'_l$ of $\dot{h}_l$ with sample paths in
761 $\mathcal{C}^{\alpha-1}(\mathcal{V}_l)$ almost surely for all $\alpha < \nu$ and such that

$$\forall \alpha < \nu, \Pi\left[\left\|\tilde{h}'_l\right\|_{\mathcal{C}^{\alpha-1}(\mathcal{V}_l)}^2\right] \leq C_{\nu,\alpha} < +\infty \tag{121}$$

762 Take any $c_l \in (a_l, b_l)$ and consider $\tilde{h}_l := h_l(c_l) + \int_{c_l}^{\cdot} \tilde{h}'_l(t)\,\mathrm{d}t$. Then since $\tilde{h}'_l$ is almost surely in
763 $\mathcal{C}^{\alpha-1}(\mathcal{V}_l)$, $\tilde{h}_l$ is has almost surely $\mathcal{C}^\alpha(\mathcal{V}_l)$ sample paths. Moreover, it is easy to check using our
764 previous results that $\tilde{h}_l$ has an $L^2$-derivative given by $\tilde{h}'_l$. This implies that $\tilde{h}_l$ is a version of $h_l$:
765 indeed, for any $H \in L^2$, the function $x \mapsto \Pi\left[\left(\tilde{h}_l(x) - h_l(x)\right)H\right]$ can be seen to have a vanishing
766 derivative, and is equal to 0 at $x = c_l$, hence $\Pi\left[\left(\tilde{h}_l(x) - h_l(x)\right)H\right] = 0$ for every $H \in L^2$ and
767 $x \in \mathcal{V}_l$ which implies that for every $x \in \mathcal{V}_l$ $\tilde{h}_l(x) = h_l(x)$ almost surely.

768 Consider now $\tilde{h} = \sum_{l=1}^L \tilde{h}_l \circ \phi_l$. Then, arguing as in the case $\nu \leq 1$, we find that $\tilde{h}$ is a version of
769 $h$ with $\mathcal{C}^\alpha(\mathcal{M})$ sample paths for every $\alpha < \nu$, and that for every $\alpha < \nu$ we have $\Pi\left[\left\|\tilde{h}\right\|_{\mathcal{C}^\alpha(\mathcal{M})}^2\right] \leq$
770 $C_{\alpha,\nu} < +\infty$.

771 $\qquad\qquad\qquad\qquad\qquad\qquad\qquad\qquad\qquad\qquad\qquad\qquad\qquad\qquad\qquad\qquad\qquad\qquad\qquad \square$

772 Using the last result and known properties of the Euclidean Matérn processes, we prove the next
773 lemma that shows in a way that all of the Matérn processes presented in this paper are sub-Gaussian,
774 uniformly with respect to the truncation parameter in the case of the truncated intrinsic Matérn
775 process, and live in Hölder spaces with appropriate exponents. This result will be used to control
776 Hölder norms when going from the empirical $L^2$-norm to the full $L^2$-norm. We use the notation
777 $\Pi_n$ in the next result to emphasize that the prior depends on the sample size when we consider a
778 truncated intrinsic Matérn process.

**Lemma 28.** *For $\Pi_n$ the prior in either Definition 4, Theorem 6 or Definition 7, for every $\nu > 0$ and $\gamma < \nu, \gamma \notin \mathbb{N}$, there exists a constant $\sigma(f) = \sigma_\gamma(f)$ independent of $n$ such that*

$$\forall x > 0, \Pi_n \left[ \|f\|_{\mathcal{C}^\gamma(\mathcal{M})} > (x+1)\sigma(f) \right] \leq 2e^{-x^2/2} \tag{122}$$

*Proof.* We start by the restriction $f$ of an extrinsic Matérn process $\tilde{f}$ to $\mathcal{M}$, as in Definition Definition 7. By section 3.1 in van der Vaart and van Zanten [47], for every $\gamma < \nu$ we have $\tilde{f} \in \mathcal{C}^\gamma([0,1]^D)$ almost surely. By lemma I.7 in Ghosal and van der Vaart [17], for every $\gamma < \nu$ $\tilde{f}$ is a gaussian random element in the Banach space $\mathcal{C}^\gamma([0,1]^D)$. In particular, by the Borell-Sudakov-Tsirelson inequality (proposition I.8 in Ghosal and van der Vaart [17]) we have :

$$\forall x > 0, \Pi \left[ \left\| \tilde{f} \right\|_{\mathcal{C}^\gamma([0,1]^D)} > (x+1)\sigma\left(\tilde{f}\right) \right] \leq 2e^{-x^2/2} \tag{123}$$

where $\sigma\left(\tilde{f}\right) = \Pi \left[ \left\| \tilde{f} \right\|^2_{\mathcal{C}^\gamma([0,1]^D)} \right]^{1/2} < \infty$. Since $\mathcal{M}$ is smooth, the restriction $f$ also satisfies

$$\forall x > 0, \Pi \left[ \|f\|_{\mathcal{C}^\gamma(\mathcal{M})} > (x+1)\sigma(f) \right] \leq 2e^{-x^2/2} \tag{124}$$

perhaps for a possibly larger constant $\sigma(f)$.

The case of the intrinsic Matérn process $f \sim \Pi_n$ truncated at $J_n \in \mathbb{N} \cup \{\infty\}$ follows in the same way, as we have shown in Lemma 27 that $\sup_{n \geq 1} \Pi_n \left[ \|f\|^2_{\mathcal{C}^\alpha(\mathcal{M})} \right] \leq C_{\alpha,\nu}$. $\square$

In order to apply Bernstein's inequality when going from the empirical $L^2$-norm to the full $L^2$-norm, we will also need this following extrapolation lemma.

**Lemma 29.** *For any function $g : \mathcal{M} \to \mathbb{R}$ and $\gamma \notin \mathbb{N}$ we have*

$$\|g\|_\infty \lesssim \|g\|^{\frac{d}{2\gamma+d}}_{\mathcal{C}^\gamma(\mathcal{M})} \|g\|^{\frac{2\gamma}{2\gamma+d}}_2 \tag{125}$$

*Proof.* We use lemma 15 from van der Vaart and van Zanten [47] and push it through charts. More precisely we have, using $B^\gamma_{\infty,\infty}([0,1]^D) = \mathcal{C}^\gamma([0,1]^D)$ for $\gamma \notin \mathbb{N}$, that

$$\|g\|_\infty \leq \sum_l \left\| (\chi_l g) \circ \phi_l^{-1} \right\|_{L^\infty(\mathcal{V}_l)} \tag{126}$$

$$\lesssim \max_l \left\| (\chi_l g) \circ \phi_l^{-1} \right\|^{\frac{d}{2\gamma+d}}_{\mathcal{C}^\gamma(\mathcal{V}_l)} \left\| (\chi_l g) \circ \phi_l^{-1} \right\|^{\frac{2\gamma}{2\gamma+d}}_{L^2(\mathcal{V}_l)} \tag{127}$$

By definition of the the manifold Hölder spaces this gives

$$\|g\|_\infty \lesssim \|g\|^{\frac{d}{2\gamma+d}}_{\mathcal{C}^\gamma(\mathcal{M})} \max_l \left\| (\chi_l g) \circ \phi_l^{-1} \right\|^{\frac{2\gamma}{2\gamma+d}}_{L^2(\mathcal{V}_l)} \tag{128}$$

Finally since the $\chi_l$'s are bounded, the charts are smooth and $p_0$ is lower bounded we have

$$\left\| (\chi_l g) \circ \phi_l^{-1} \right\|^2_{L^2(\mathcal{V}_l)} = \int_{\mathcal{V}_l} \left| (\chi_l g) \circ \phi_l^{-1}(y) \right|^2 dy \lesssim \int_{\mathcal{U}_l} g^2(x) p_0(x) \mu(dx) \lesssim \|g\|^2_2 \tag{129}$$

which gives the result. $\square$

Having established regularity properties for our prior processes, we now turn to the so-called *small ball problem*: we want to find sharp lower bounds on $\Pi[\|f\|_\infty < \varepsilon]$ where $f \sim \Pi$ is our prior process. This will be crucial in order to control the concentration functions. In fact, it is well-known that this problem is closely related to the estimation of the metric entropy of the unit ball of the RKHS of $f$ with respect to the uniform norm: see Li and Linde [26] for details. Since we have already characterized the RKHS of our processes in Proposition 22 and Lemma 23, we are able to lower bound the small-ball probabilities. The technicality here involves getting a bound uniform in the truncation parameter for the truncated intrinsic Matérn process, as the truncated Matérn process is a sequence of priors rather than a fixed prior.

**Lemma 30.** *If $f \sim \Pi_n$ the prior in either Definition 4 and Theorem 6 or Definition 7 with smoothness parameter $\nu > 0$, then there exist two constants $C, \varepsilon_0 > 0$ that do not depend on $n$ such that for all $\varepsilon \leq \varepsilon_0$ we have $-\ln \Pi_n[\|f\|_\infty < \varepsilon] \leq C\varepsilon^{-\frac{d}{\nu}}$.*

*Proof.* Because the processes are Gaussian random elements in $\mathcal{C}(\mathcal{M})$, their stochastic process RKHS given by Proposition 22 coincide with their Gaussian random element RKHS. Hence, for the non-truncated intrinsic and the extrinsic Matérn processes the result is a direct application of Lemma 19 and Li and Linde [26], Theorem 1.2.

For the intrinsic Matérn process truncated at $J_n$ it is not immediately clear that the constants $C, \varepsilon_0$ can be taken independent of $n$, and we go through the proof of Li and Linde [26], Proposition 3.1 to see this. We first need a crude upper bound of the form

$$-\ln \Pi_n[\|f\|_\infty < \varepsilon] \leq c\varepsilon^{-c} \tag{130}$$

for some possibly large constant $c > 0$. To get such a bound, we use Castillo et al. [9], Proposition 3 which shows the existence of a universal constant $C > 0$ such that

$$\forall \varepsilon \leq \min(1, 4\sigma(f)) \qquad -\ln \Pi_n[\|f\|_\infty < \varepsilon] \leq Cn(\varepsilon) \ln\left(\frac{6n(\varepsilon)(1 \vee \sigma(f))}{\varepsilon}\right) \tag{131}$$

where $\sigma(f) = \Pi_n\left[\|f\|_\infty^2\right]^{1/2}$ and $n(\varepsilon)$ is defined in Li and Linde [26] by

$$\max\{j \geq 0 : 4l_j(f) \geq \varepsilon\}, l_j(f) = \inf\left\{\Pi_n\left[\left\|\sum_{j\geq 0} Z_j h_j\right\|_\infty^2\right] : f \stackrel{(d)}{=} \sum_{j\geq 0} Z_j h_j\right\} \tag{132}$$

with $\stackrel{(d)}{=}$ standing for the equality in distributions and the infimum being taken over every possible decomposition $\sum_{j\geq 0} Z_j h_j$ with $h_j \in \mathcal{C}(\mathcal{M})$, $Z_j$ being a sequence of IID $N(0,1)$ random variables as in Definition 4, and the series being required to converge uniformly almost surely.

The function $f = \sum_{j=0}^{J_n} \left(\frac{2\nu}{\kappa^2} + \lambda_j\right)^{-\frac{\nu+d/2}{2}} Z_j f_j$ is a valid decomposition. Therefore

$$l_J(f) \leq \Pi_n\left[\left\|\sum_{j=J}^{J_n}\left(\frac{2\nu}{\kappa^2} + \lambda_j\right)^{-\frac{\nu+d/2}{2}} Z_j f_j\right\|_\infty^2\right]^{1/2}. \tag{133}$$

Still by the Sobolev Embedding Theorem and by Weyl's Law, given in Result 10, for every $\gamma > \max(d/2, \nu)$ there exists a constant $C = C_{\gamma,\mathcal{M}}$ such that for all $J \in \mathbb{N}$, allowing $C$ to change from line to line, we have

$$\Pi_n\left[\left\|\sum_{j=J+1}^{J_n}(1+\lambda_j)^{-\frac{\nu+d/2}{2}} Z_j f_j\right\|_\infty^2\right] \leq C^2 \Pi_n\left[\left\|\sum_{j=J+1}^{J_n}(1+\lambda_j)^{-\frac{\nu+d/2}{2}} Z_j f_j\right\|_{H^\gamma(\mathcal{M})}^2\right] \tag{134}$$

$$= C^2 \sum_{j=J+1}^{J_n}(1+\lambda_j)^{-(\nu+d/2-\gamma)} \tag{135}$$

$$\leq C^2 \sum_{j=J+1}^{J_n}(j+1)^{-(1+2(\nu-\gamma)/d)} \tag{136}$$

$$\leq C^2 \sum_{j>J}(j+1)^{-(1+2(\nu-\gamma)/d)} \tag{137}$$

$$\leq C^2(J+1)^{-2(\nu-\gamma)/d} \tag{138}$$

By choosing $J = 0$ this gives us $\sigma(f) \leq C$ independent of $n$. Moreover, by choosing $J \geq C\varepsilon^{-\frac{d}{2(\nu-\gamma)}}$, again for a comparison constant $C$ independent of $n$, this gives us $n(\varepsilon) \leq C\varepsilon^{-\frac{d}{2(\nu-\gamma)}}$ for $C$ independent of $n$. This implies using Castillo et al. [9], Proposition 3 that

$$-\ln \Pi_n[\|f\|_\infty < \varepsilon] \leq C\varepsilon^{-C} \tag{139}$$

830  for $C > 0$ independent of $n$.

831  With this crude bound we can now continue the proof of Li and Linde [26], Proposition 3.1. For this,
832  we need a metric entropy estimate. For this notice that for all $J \in \mathbb{N} \cup \{\infty\}$ we have $B_{\mathbb{H}^J}(0,1) \subset$
833  $B_{\mathbb{H}^\infty}(0,1) = B_{H^{\nu+d/2}(\mathcal{M})}(0,1)$, and therefore using Lemma 19 we have the metric entropy estimate

$$\ln N(B_{\mathbb{H}^J}(0,1)) \leq C\varepsilon^{-\frac{d}{\nu+d/2}} \tag{140}$$

834  for a constant $C > 0$ independent of $J$. Therefore following the proof of proposition 3.1 in Li and
835  Linde [26] we find $-\ln \Pi_n[\|f\|_\infty < \varepsilon] \leq C\varepsilon^{-\frac{d}{\nu}}$ for every $\varepsilon \leq \varepsilon_0$, where $C, \varepsilon_0 > 0$ are constants
836  independent of $n$.  $\square$

837  This concludes this section and we now turn to the proofs of our main results.

# C  Proofs

839  We recall that in the following the expression $a \lesssim b$ means $a \leq Cb$ for some constant $C > 0$ whose
840  value is irrelevant for our claims. We first define our notation for Gaussian likelihood and probability
841  distribution of the sample.

842  **Definition 31.** *For every $\boldsymbol{x} \in \mathcal{M}^n$ and $f : \mathcal{M} \to \mathbb{R}$ we define $p_{f,\boldsymbol{x},\boldsymbol{y}}$ to be the joint distribution*
843  *corresponding to the marginal $p_{\boldsymbol{x}} = p_0$ and conditional $p_{\boldsymbol{y}|\boldsymbol{x}} = \mathrm{N}(f(\boldsymbol{x}), \sigma_\varepsilon^2 \mathbf{I})$, where $f(\boldsymbol{x})$ is the*
844  *vector with entries $f(x_i)$. Expectations with respect to $p_{f,\boldsymbol{x},\boldsymbol{y}}$ we denote by $\mathbb{E}_{f,\boldsymbol{x},\boldsymbol{y}}$ and to $p_0$ by $\mathbb{E}_{\boldsymbol{x}}$.*

845  Following van der Vaart and van Zanten [47], Theorem 1, which is valid for any compact space hence
846  also for $\mathcal{M}$, we can deduce a posterior contraction rate with respect to the *empirical $L^2$-norm*[8]

$$\|f\|_n = \left(\frac{1}{n}\sum_{i=1}^n f(x_i)^2\right)^{1/2} \tag{141}$$

847  by studying first the so-called *concentration functions* with respect to the uniform norm. This is the
848  object of the following lemma. We again recall that the prior $\Pi_n$ may depend on $n$ if we consider a
849  truncated intrinsic Matérn process.

850  **Theorem 32.** *Let $\Pi_n$ denote the prior in either Theorem 5, Theorem 6 or Theorem 8 with smoothness*
851  *parameter $\nu$. Let $\mathbb{H}_n$ denote the corresponding RKHS. Define the* CONCENTRATION FUNCTION *for*
852  *$f_0 \in C(\mathcal{M})$ and $\varepsilon > 0$ by*

$$\varphi_{f_0}(\varepsilon) = -\ln \Pi_n[\|f\|_\infty < \varepsilon] + \inf_{f \in \mathbb{H}_n : \|f - f_0\|_\infty < \varepsilon} \|f\|_{\mathbb{H}_n}^2. \tag{142}$$

853  *Then if $f_0 \in H^\beta(\mathcal{M}) \cap B_{\infty,\infty}^\beta(\mathcal{M}), \beta > 0$ we have $\varphi_{f_0}(\varepsilon_n) \leq n\varepsilon_n^2$ for $\varepsilon_n$ a multiple of $n^{-\frac{\min(\nu,\beta)}{2\nu+d}}$.*

854  *Proof.*  The first term on the right-hand side of Equation (142) is bounded by $C\varepsilon^{-d/\nu}$ by Lemma 30.
855  To bound the second term, we assume, without loss of generality,[9] that $\nu \geq \beta$. Consider an
856  approximation $f = \Phi_j(\sqrt{\Delta})f_0$ of $f_0$, where $c\varepsilon \leq 2^{-\beta j} \leq \varepsilon$ and $c > 0$ is an absolute constant. Since
857  we assume $f_0 \in B_{\infty,\infty}^\beta(\mathcal{M})$, by definition of $B_{\infty,\infty}^\beta(\mathcal{M})$ we have

$$\|f_0 - f\|_\infty \leq \|f_0\|_{B_{\infty,\infty}^\beta(\mathcal{M})} 2^{-\beta j} \lesssim \varepsilon \tag{143}$$

858  where in the last inequality the $B_{\infty,\infty}^\beta(\mathcal{M})$-norm is the constant implied by notation $\lesssim$. We now
859  show that

$$\|f\|_{\mathbb{H}}^2 \lesssim \varepsilon^{-\frac{2}{\beta}(\nu-\beta+d/2)} \tag{144}$$

---

[8]This is actually a seminorm, but we follow the rest of the literature in referring to it as a norm.

[9]Because $H^\beta(\mathcal{M}) \cap B_{\infty,\infty}^\beta(\mathcal{M}) \subseteq H^{\min(\beta,\nu)}(\mathcal{M}) \cap B_{\infty,\infty}^{\min(\beta,\nu)}(\mathcal{M})$, if $\beta > \nu$ then $f_0 \in H^\beta(\mathcal{M}) \cap$
$B_{\infty,\infty}^\beta \subseteq H^\nu(\mathcal{M}) \cap B_{\infty,\infty}^\nu(\mathcal{M})$ gives a rate of $n^{-\frac{\nu}{2\nu+d}} = n^{-\frac{\min(\beta,\nu)}{2\nu+d}}$.

First notice that by Lemma 23 and Proposition 22, for any prior considered here we have $\mathbb{H} \subseteq H^{\nu+d/2}(\mathcal{M})$ and $\|\cdot\|_{\mathbb{H}} \leq \|\cdot\|_{H^{\nu+d/2}(\mathcal{M})}$ for a constant $C$ that does not depend on $n$. Hence using Result 10 and properties of $\Phi$ we have

$$\|f\|_{\mathbb{H}}^2 \lesssim \|f\|_{H^{\nu+d/2}(\mathcal{M})}^2 \tag{145}$$

$$= \sum_{l \geq 0}(1+\lambda_l)^{\nu+d/2}\Phi^2\left(2^{-j}\sqrt{\lambda_l}\right)|\langle f_l, f_0\rangle|^2 \tag{146}$$

$$\leq \sum_{l:\sqrt{\lambda_l}\leq 2^{j+1}}(1+\lambda_l)^{\nu+d/2-\beta}(1+\lambda_l)^{\beta}|\langle f_l, f_0\rangle|^2 \tag{147}$$

$$\leq 2^{(j+1)(2\nu-2\beta+d)}\sum_{l:\sqrt{\lambda_l}\leq 2^{j+1}}(1+\lambda_l)^{\beta}|\langle f_l, f_0\rangle|^2 \tag{148}$$

$$\leq 2^{(j+1)(2\nu-2\beta+d)}\sum_{l \geq 0}(1+\lambda_l)^{\beta}|\langle f_l, f_0\rangle|^2 \tag{149}$$

$$= 2^{(j+1)(2\nu-2\beta+d)}\|f_0\|_{H^{\beta}(\mathcal{M})}^2 \tag{150}$$

$$\leq 2^{2(\nu-\beta+d/2)}c^{-\frac{2}{\beta}(\nu-\beta+d/2)}\|f_0\|_{H^{\beta}(\mathcal{M})}^2\varepsilon^{-\frac{2}{\beta}(\nu-\beta+d/2)} \tag{151}$$

Our assumption $\nu \geq \beta$ implies that

$$\frac{2}{\beta}(\nu-\beta+d/2) \geq \frac{d}{\beta} \geq \frac{d}{\nu}. \tag{152}$$

Hence we have $\varepsilon^{-d/\nu} \leq \varepsilon^{-\frac{2}{\beta}(\nu-\beta+d/2)}$ which gives us $\varphi_{f_0}(\varepsilon) \lesssim \varepsilon^{-\frac{2}{\beta}(\nu-\beta+d/2)}$. It is then easy to check that $\varepsilon_n = Mn^{-\frac{\beta}{2\nu+d}}$ satisfies $\varphi_{f_0}(\varepsilon_n) \leq n\varepsilon_n^2$ for $M > 0$ large enough. $\qquad\square$

From this we deduce an upper bound on the error in *the empirical $L^2$ norm* $\|\cdot\|_n$, i.e. on the Euclidean distance between the posterior Gaussian process $f$ and the ground truth function $f_0$ evaluated at data locations $x_i$.

**Lemma 33.** *Let $\Pi_n$ denote the prior in either Theorem 5, Theorem 6 or Theorem 8 with smoothness parameter $\nu > 0$. Fix $f_0 \in H^{\beta}(\mathcal{M}) \cap B_{\infty,\infty}^{\beta}(\mathcal{M})$ with $\beta > 0$. Then*

$$\mathbb{E}_{f \sim \Pi_n(\cdot|\boldsymbol{x},\boldsymbol{y})}\|f-f_0\|_n^q \leq \varepsilon_n^q \tag{153}$$

*for all $q \geq 1$ and $\varepsilon_n$ a constant multiple of $n^{-\frac{\min(\nu,\beta)}{2\nu+d}}$ with constant depending on $f_0, q, \nu$.*

*Proof.* By Theorem 32 for $\varepsilon_n$ a multiple of $n^{-\frac{\min(\beta,\nu)}{2\nu+d}}$, we have $\varphi_{f_0}(\varepsilon_n) \leq n\varepsilon_n^2$. By virtue of this, the proof of Theorem 1 and Proposition 11 of van der Vaart and van Zanten [47] imply the result. Indeed, the proof of Theorem 1 relies solely on the fact that $\varphi_{f_0}(\varepsilon_n/2) \leq n\varepsilon_n^2$ and an application of van der Vaart and van Zanten [47], Proposition 11. We have $\varphi_{f_0}(\varepsilon_n) \leq n\varepsilon_n^2 \leq n(2\varepsilon_n)^2$ and hence the condition is satisfies with $\varepsilon_n$ replaced by $2\varepsilon_n$. Moreover, even if van der Vaart and van Zanten [47], Theorem 1 is formulated for $q = 2$, van der Vaart and van Zanten [47], Proposition 11 gives a result for all $q \geq 1$. $\qquad\square$

Notice that for the last result we only assumed $\nu, \beta > 0$, and therefore require no constraints on the smoothness parameters. We now turn to the proofs of our main results, Theorems 5, 6 and 8. For them, the extra assumption $\min(\beta,\nu) > d/2$ is needed in order to go from the empirical $L^2$ norm to the true $L^2(p_0)$ norm, leveraging regularity of the ground truth function and the Gaussian process. The value $d/2$ in this assumption is not surprising, as by the Sobolev embedding theorem this is the minimal natural requirement to guarantee that $f_0$ and functions in the support of the prior are at least continuous.

*Proof of Theorems 5, 6 and 8.* Given the technical lemmas from Appendix B and Lemma 33, the proof is similar to the one of Theorem 2 in van der Vaart and van Zanten [47]. We include it for completeness and to point out the differences in our context.

Take $\varepsilon_n \propto n^{-\frac{\min(\beta,\nu)}{2\nu+d}}$ satisfying $\varphi_{f_0}(\varepsilon_n/2) \le n\varepsilon_n^2$ (such a rate exists by Theorem 32). Then for each $n$ there exists an element $f_n \in \mathbb{H}_n$, where this notation refers to the RKHS corresponding to $\Pi_n$, satisfying

$$\|f_n\|_{\mathbb{H}}^2 \le n\varepsilon_n^2 \qquad\qquad \|f_n - f_0\|_\infty \le \varepsilon_n/2. \tag{154}$$

Hence for any $\gamma$ such that $d/2 < \gamma < \nu, \gamma \notin \mathbb{N}$, any $s > 0, \tau > 0$ and an indexed family of events $\mathcal{A}_r$ that is to be chosen in the future we have

$$\varepsilon_n^{-q}\, \mathbb{E}_{\boldsymbol{x},\boldsymbol{y}}\, \mathbb{E}_{f\sim\Pi_n(\cdot|\boldsymbol{x},\boldsymbol{y})}\|f - f_0\|_{L^2(p_0)}^q \lesssim \varepsilon_n^{-q}\, \mathbb{E}_{\boldsymbol{x},\boldsymbol{y}}\, \mathbb{E}_{f\sim\Pi_n(\cdot|\boldsymbol{x},\boldsymbol{y})}\|f_n - f_0\|_{L^2(p_0)}^q \tag{155}$$

$$+ \varepsilon_n^{-q}\, \mathbb{E}_{\boldsymbol{x},\boldsymbol{y}}\, \mathbb{E}_{f\sim\Pi_n(\cdot|\boldsymbol{x},\boldsymbol{y})}\|f - f_n\|_{L^2(p_0)}^q \tag{156}$$

$$\lesssim 1 + \varepsilon_n^{-q}\, \mathbb{E}_{\boldsymbol{x},\boldsymbol{y}}\, \mathbb{E}_{f\sim\Pi_n(\cdot|\boldsymbol{x},\boldsymbol{y})}\|f - f_n\|_{L^2(p_0)}^q \tag{157}$$

$$= 1 + \mathbb{E}_{\boldsymbol{x},\boldsymbol{y}} \int_0^\infty qr^{q-1}\Pi_n(\mathcal{B}(r) \mid \boldsymbol{x},\boldsymbol{y})\, \mathrm{d}r \tag{158}$$

where the events $\mathcal{B}(r)$ are defined by $\mathcal{B}(r) = \left\{ \|f - f_n\|_{L^2(p_0)} > \varepsilon_n r \right\}$. Denote

$$\mathcal{B}^{(\mathrm{I})}(r) = \{2\|f - f_n\|_n > \varepsilon_n r\} \tag{159}$$

$$\mathcal{B}^{(\mathrm{II})}(r) = \left\{ \|f\|_{\mathcal{C}^\gamma(\mathcal{M})} > \tau\sqrt{n}\varepsilon_n r^s \right\} \tag{160}$$

$$\mathcal{B}^{(\mathrm{III})}(r) = \left\{ \|f\|_{\mathcal{C}^\gamma(\mathcal{M})} \le \tau\sqrt{n}\varepsilon_n r^s,\ 2\|f - f_n\|_n \le \varepsilon_n r < \|f - f_n\|_{L^2(p_0)} \right\}. \tag{161}$$

Then $\mathcal{B}(r) \subseteq \mathcal{B}^{(\mathrm{I})}(r) \cup \mathcal{B}^{(\mathrm{II})}(r) \cup \mathcal{B}^{(\mathrm{III})}(r)$ and thus

$$\varepsilon_n^{-q}\, \mathbb{E}_{\boldsymbol{x},\boldsymbol{y}}\, \mathbb{E}_{f\sim\Pi_n(\cdot|\boldsymbol{x},\boldsymbol{y})}\|f - f_0\|_{L^2(p_0)}^q \lesssim 1 + \mathbb{E}_{\boldsymbol{x},\boldsymbol{y}} \int_0^\infty r^{q-1}\Pi_n\Big(\mathcal{B}^{(\mathrm{I})}(r) \mid \boldsymbol{x},\boldsymbol{y}\Big)\, \mathrm{d}r \tag{162}$$

$$+ \mathbb{E}_{\boldsymbol{x},\boldsymbol{y}} \int_0^\infty r^{q-1}\mathbb{1}_{\mathcal{A}_r^c}\, \mathrm{d}r \tag{163}$$

$$+ \mathbb{E}_{\boldsymbol{x},\boldsymbol{y}} \int_0^\infty r^{q-1}\mathbb{1}_{\mathcal{A}_r}\Pi_n\Big(\mathcal{B}^{(\mathrm{II})}(r) \mid \boldsymbol{x},\boldsymbol{y}\Big)\, \mathrm{d}r \tag{164}$$

$$+ \mathbb{E}_{\boldsymbol{x},\boldsymbol{y}} \int_0^\infty r^{q-1}\mathbb{1}_{\mathcal{A}_r}\Pi_n\Big(\mathcal{B}^{(\mathrm{III})}(r) \mid \boldsymbol{x},\boldsymbol{y}\Big)\, \mathrm{d}r. \tag{165}$$

For the first term, by Lemma 33 applied conditionally on the $x_i$-values, for which we got a bound on the integrated empirical $L^2$ norm uniformly on the design points, we have

$$\mathbb{E}_{\boldsymbol{x},\boldsymbol{y}} \int_0^\infty r^{q-1}\Pi_n\Big(\mathcal{B}^{(\mathrm{I})}(r) \mid \boldsymbol{x},\boldsymbol{y}\Big)\, \mathrm{d}r \lesssim \mathbb{E}_{\boldsymbol{x},\boldsymbol{y}}\, \mathbb{E}_{f\sim\Pi_n(\cdot|\boldsymbol{x},\boldsymbol{y})}\|f - f_0\|_n^q \tag{166}$$

$$\lesssim \mathbb{E}_{\boldsymbol{x},\boldsymbol{y}}\, \mathbb{E}_{f\sim\Pi_n(\cdot|\boldsymbol{x},\boldsymbol{y})}\|f - f_n\|_n^q + \|f_0 - f_n\|_\infty^q \tag{167}$$

$$\lesssim \varepsilon_n^q \tag{168}$$

Moreover, by Lemma 14 in van der Vaart and van Zanten [47] applied with $r$ in the notation of the reference being equal to $\sqrt{n}\varepsilon_n r^s$, for each $r > 0$ the event

$$\mathcal{A}_r(\boldsymbol{x}) = \left\{ \int \frac{p_{\boldsymbol{y}|\boldsymbol{x}}^{(f)}(\boldsymbol{y})}{p_{\boldsymbol{y}|\boldsymbol{x}}^{(f_0)}(\boldsymbol{y})}\Pi_n(df) \ge e^{-n\varepsilon_n^2 r^{2s}}\Pi_n[\|f - f_0\|_\infty < \varepsilon_n r^s] \right\} \tag{169}$$

is such that

$$p_{\boldsymbol{y}|\boldsymbol{x}}^{(f_0)}[\mathcal{A}_r^c(\boldsymbol{x})] \le e^{-n\varepsilon_n^2 r^{2s}/8} \tag{170}$$

Therefore, by Fubini's Theorem, since $n\varepsilon_n^2 \ge n^{\frac{d}{2\nu+d}} \ge 1$ the second term is bounded by

$$\mathbb{E}_{\boldsymbol{x},\boldsymbol{y}}^{(f_0)} \int_0^\infty r^{q-1}\mathbb{1}_{\mathcal{A}_r^c(\boldsymbol{x})}\, \mathrm{d}r = \int_0^\infty r^{q-1}\mathbb{E}_{\boldsymbol{x}}\Big[\mathbb{E}_{\boldsymbol{y}|\boldsymbol{x}}^{(f_0)}[\mathcal{A}_r^c(\boldsymbol{x})]\Big]\, \mathrm{d}r \tag{171}$$

$$\le \int_0^\infty r^{q-1}e^{-n\varepsilon_n^2 r^{2s}/8}\, \mathrm{d}r \tag{172}$$

$$\le \int_0^\infty r^{q-1}e^{-r^{2s}/8}\, \mathrm{d}r \tag{173}$$

$$\le C \tag{174}$$

where $C = C_{s,q} < \infty$. It remains to bound the last two terms. By Bayes' Rule, we have the equality

$$\Pi_n\left[\|f\|_{\mathcal{C}^\gamma(\mathcal{M})} > \tau\sqrt{n}\varepsilon_n r^s | \boldsymbol{y}\right] = \frac{\int_{\|f\|_{\mathcal{C}^\gamma(\mathcal{M})} > \tau\sqrt{n}\varepsilon_n r^s} \Pi_{i=1}^n \frac{dp_{f,\boldsymbol{x}}^{(n)}}{dp_{f_0,\boldsymbol{x}}^{(n)}} \Pi_n(df)}{\int \Pi_{i=1}^n \frac{dp_{f,\boldsymbol{x}}^{(n)}}{dp_{f_0,\boldsymbol{x}}^{(n)}} \Pi_n(df)} \tag{175}$$

therefore on $\mathcal{A}_r(\boldsymbol{x})$ we have

$$\Pi_n\left[\|f\|_{\mathcal{C}^\gamma(\mathcal{M})} > \tau\sqrt{n}\varepsilon_n r^s | \boldsymbol{y}\right] \tag{176}$$

$$\leq \frac{e^{n\varepsilon_n^2 r^{2s}}}{\Pi_n[\|f - f_0\|_\infty < \varepsilon_n r^s]} \int_{\|f\|_{\mathcal{C}^\gamma(\mathcal{M})} > \tau\sqrt{n}\varepsilon_n r^s} \Pi_{i=1}^n \frac{dp_{f,\boldsymbol{x}}^{(n)}}{dp_{f_0,\boldsymbol{x}}^{(n)}} \Pi_n(df) \tag{177}$$

Hence taking expectation and using Fubini–Tonelli's Theorem gives

$$\mathbb{E}_{\boldsymbol{xy}}^{(f_0)}\left[\mathbb{1}_{\mathcal{A}_r(\boldsymbol{x})}\Pi_n\left[\|f\|_{\mathcal{C}^\gamma(\mathcal{M})} > \tau\sqrt{n}\varepsilon_n r^s | \boldsymbol{y}\right]\right] \tag{178}$$

$$\leq \frac{e^{n\varepsilon_n^2 r^{2s}}}{\Pi_n[\|f - f_0\|_\infty < \varepsilon_n r^s]} \mathbb{E}_{\boldsymbol{xy}}^{(f_0)}\left[\int_{\|f\|_{\mathcal{C}^\gamma(\mathcal{M})} > \tau\sqrt{n}\varepsilon_n r^s} \Pi_{i=1}^n \frac{dp_{f,\boldsymbol{x}}^{(n)}}{dp_{f_0,\boldsymbol{x}}^{(n)}} \Pi_n(df)\right] \tag{179}$$

$$= \frac{e^{n\varepsilon_n^2 r^{2s}}}{\Pi_n[\|f - f_0\|_\infty < \varepsilon_n r^s]} \Pi_n\left[\|f\|_{\mathcal{C}^\gamma(\mathcal{M})} > \tau\sqrt{n}\varepsilon_n r^s\right] \tag{180}$$

Therefore the third term can be bounded by

$$\mathbb{E}_{\boldsymbol{x},\boldsymbol{y}}^{(f_0)} \int_0^\infty r^{q-1}\mathbb{1}_{\mathcal{A}_r}\Pi_n\left(\mathcal{B}^{(\mathrm{II})}(r) \mid \boldsymbol{x},\boldsymbol{y}\right) dr \tag{181}$$

$$\leq \int_0^\infty r^{q-1} \frac{e^{n\varepsilon_n^2 r^{2s}}}{\Pi_n[\|f - f_0\|_\infty < \varepsilon_n r^s]} \Pi_n\left[\|f\|_{\mathcal{C}^\gamma(\mathcal{M})} > \tau\sqrt{n}\varepsilon_n r^s\right] dr \tag{182}$$

Now using Lemma 28, for a possibly small constant $c > 0$ independent of $n$, we have

$$\Pi_n\left[\|f\|_{\mathcal{C}^\gamma(\mathcal{M})} > \tau\sqrt{n}\varepsilon_n r^s | \boldsymbol{y}\right] \leq e^{-cn\tau^2\varepsilon_n^2 r^{2s}} \tag{183}$$

Moreover, by using the bound on the concentration function in Theorem 32 and Ghosal and van der Vaart [17], Proposition 11.19, we can assume that

$$\Pi_n\left[\|f - f_0\|_\infty < \sqrt{n}\varepsilon_n r^s\right] \geq e^{-c^{-1}n\varepsilon_n^2 r^{2s}}. \tag{184}$$

Therefore the third term is bounded by

$$\mathbb{E}_{\boldsymbol{x},\boldsymbol{y}}^{(f_0)} \int_0^\infty r^{q-1}\mathbb{1}_{\mathcal{A}_r}\Pi_n\left(\mathcal{B}^{(\mathrm{II})}(r) \mid \boldsymbol{x},\boldsymbol{y}\right) dr \leq \int_0^\infty r^{q-1}e^{-cn\tau^2\varepsilon_n^2 r^{2s}}e^{c^{-1}n\varepsilon_n^2 r^{2s}} dr \tag{185}$$

$$\leq \int_0^\infty r^{q-1}e^{-r^{2s}} dr < \infty \tag{186}$$

if $\tau^2 c > 1 + c^{-1}0$. It remains to bound the last term.

We have by the same arguments as above that

$$\mathbb{E}_{\boldsymbol{x},\boldsymbol{y}} \int_0^\infty r^{q-1} \mathbb{1}_{\mathcal{A}_r} \Pi_n\Big(\mathcal{B}^{(\mathrm{III})}(r) \mid \boldsymbol{x},\boldsymbol{y}\Big) \,\mathrm{d}r \tag{187}$$

$$= \mathbb{E}_{\boldsymbol{x},\boldsymbol{y}} \int_0^\infty r^{q-1} \mathbb{1}_{\mathcal{A}_r(\boldsymbol{x})} \tag{188}$$

$$\times \Pi_n\Big[\|f\|_{\mathcal{C}^\gamma(\mathcal{M})} \leq \tau\sqrt{n}\varepsilon_n r^s, 2\|f-f_n\|_n \leq \varepsilon_n r \leq \|f-f_n\|_2 \big| \boldsymbol{y}\Big] \,\mathrm{d}r \tag{189}$$

$$\leq \int_0^\infty r^{q-1} \frac{e^{n\varepsilon_n^2 r^{2s}}}{\Pi_n[\|f-f_0\|_\infty < \varepsilon_n r^s]} \tag{190}$$

$$\times \mathbb{E}_{\boldsymbol{x}} \, \Pi_n\Big[\|f\|_{\mathcal{C}^\gamma(\mathcal{M})} \leq \tau\sqrt{n}\varepsilon_n r^s, 2\|f-f_n\|_n \leq \varepsilon_n r \leq \|f-f_n\|_2\Big] \,\mathrm{d}r \tag{191}$$

$$\leq \int_0^\infty r^{q-1} e^{(c+1)n\varepsilon_n^2 r^{2s}} \tag{192}$$

$$\times \int_{\|f\|_{\mathcal{C}^\gamma(\mathcal{M})} \leq \tau\sqrt{n}\varepsilon_n r^s, \varepsilon_n r \leq \|f-f_n\|_2} p_0[\|f-f_n\|_2 \geq 2\|f-f_n\|_n] \Pi_n(df) \,\mathrm{d}r. \tag{193}$$

As the squared empirical $L^2$-norm is a sample average of the true $L^2$-norm, the probability in the integrand can be controlled easily via a concentration inequality. As in van der Vaart and van Zanten [47], we use Bernstein's inequality—van der Vaart and Wellner [48], Lemma 2.2.9—to find that

$$p_0[\|f-f_n\|_2 \geq 2\|f-f_n\|_n] = p_0\Big[\|f-f_n\|_n^2 - \|f-f_n\|_2^2 \leq -\frac{3}{4}\|f-f_n\|_n^2\Big] \tag{194}$$

$$\leq \exp\left(-\frac{9n}{16}\frac{\|f-f_n\|_2^2}{\|f-f_n\|_\infty^2}\right) \tag{195}$$

Moreover, by Lemma 29, since $\gamma \notin \mathbb{N}$ we have

$$\|f-f_n\|_\infty \lesssim \|f-f_n\|_{\mathcal{C}^\gamma(\mathcal{M})}^{\frac{d}{2\gamma+d}} \|f-f_n\|_2^{\frac{2\gamma}{2\gamma+d}} \tag{196}$$

Using the Sobolev Embedding Theorem—De Vito et al. [13], Theorem 4—$\|f-f_n\|_{\mathcal{C}^\gamma(\mathcal{M})} \lesssim \|f_n\|_{\mathbb{H}} + \|f\|_{\mathcal{C}^\gamma(\mathcal{M})} \lesssim \tau\sqrt{n}\varepsilon_n r^s$ whenever $\|f\|_{\mathcal{C}^\gamma(\mathcal{M})} \leq \tau\sqrt{n}\varepsilon_n r^s$. Therefore, for a constant $c > 0$ we have

$$p_0[\|f-f_n\|_2 \geq 2\|f-f_n\|_n] \leq \exp\left(-cn\frac{\|f-f_n\|_2^2}{\|f-f_n\|_{\mathcal{C}^\gamma(\mathcal{M})}^{\frac{2d}{2\gamma+d}} \|f-f_n\|_2^{\frac{4\gamma}{2\gamma+d}}}\right) \tag{197}$$

$$\leq e^{-c\tau^{-\frac{2d}{2\gamma+d}} n^{\frac{2\gamma}{2\gamma+d}} r^{\frac{2d}{2\gamma+d}(1-s)}} \tag{198}$$

Hence, we can bound the last term by

$$\mathbb{E}_{\boldsymbol{x},\boldsymbol{y}} \int_0^\infty r^{q-1} \mathbb{1}_{\mathcal{A}_r} \Pi_n\Big(\mathcal{B}^{(\mathrm{III})}(r) \mid \boldsymbol{x},\boldsymbol{y}\