# OpenReview forum: "Posterior Contraction Rates for Matérn Gaussian Processes on Riemannian Manifolds"
_NeurIPS.cc/2023/Conference — NeurIPS 2023 spotlight_

### Official Review · Reviewer_dGg7 · 2023-06-28

**Soundness:** 4 excellent
**Presentation:** 4 excellent
**Contribution:** 4 excellent
**Rating:** 8
**Confidence:** 3

**Summary:**

The paper concerns Gaussian processes on manifolds. The authors present theorems for contraction rates for Matern Gaussian processes defined intrinsically and extrinsically through and embedding in a higher-dimensional Euclidean space. The authors shows that the rates are asymptotically equal in the two settings. Additionally, they treat the case of finitely truncated series expansion of the kernels to get a similar rate. Finally, it is shown experimentally that it can still be beneficial to work with the intrinsic geometry in the small sample domain.

**Strengths:**

- extremely well-written paper. I found the exposition very clear
- novel theoretical results
- interesting findings (in line with what the authors state, I would not have expected the manifold dimension to show up in the extrinsic case)
- empirical study to cover the small sample case

**Weaknesses:**

- since the theorems are not in the main paper, one could perhaps consider if a longer format than NeurIPS (a journal paper) would be more suitable for the paper from a presentation point of view

**Questions:**

no questions

**Limitations:**

yes

---

> ### Author Rebuttal · Authors · 2023-08-08
>
> Thank you very much for your review, and especially for the comment that you found our exposition "very clear" in spite of the technical nature of the subject! Below we comment on one of the points.
>
> ---
>
> *"since the theorems are not in the main paper, one could perhaps consider if a longer format than NeurIPS (a journal paper) would be more suitable for the paper from a presentation point of view"*
> * While it certainly would have also been possible for us to write up our work in a longer format compared to NeurIPS such as JMLR, we believe that the NeurIPS format is also appropriate here, because the relatively short page limit means that a paper like ours needs to focus its main body on summarizing the main theoretical results, rather than on technical details in proofs. It is valuable to have papers like this, because they provides a picture of the state of affairs which are also readable by non-experts - we think this is particularly important for posterior contraction analysis, which tends to involve harder-to-parse mathematics than other areas. Therefore, we believe the NeurIPS format used by this work effectively complements other, much longer and more technical-detail-focused papers on related questions which can be found in the literature.

---

> > ### Comment · Reviewer_dGg7 · 2023-08-12
> >
> > Thanks for the rebuttal. My scoring has not changed.

---

### Official Review · Reviewer_woEb · 2023-07-02

**Soundness:** 3 good
**Presentation:** 4 excellent
**Contribution:** 3 good
**Rating:** 8
**Confidence:** 5

**Summary:**

This paper studies the contraction rate(s) for both the intrinsic and extrinsic Mat\'ern Gaussian process in compact Riemannian manifold. The authors proved that the (optimal) rate in both cases is $\frac{2 \min(\beta, \nu)}{2 \nu +d}$, where $\nu$ is the smooth parameter of the Mat\'ern process, $\beta$ is the regression function smoothness class, and $d$ is the dimension. The authors also showed with examples that the geometric models outperform the non-geometric ones through empirical error analysis.


**Strengths:**

The results in this paper are novel and enlightening. Up to minor typos, I enjoy reading the paper. It is concerned with the fine topic of Gaussian processes on manifolds, and on why (and how) this kind of modeling is valuable through quantitative posterior contraction analysis. The main contribution is the (optimal) contraction rate of both intrinsic and ambient Mat\'ern processes on compact manifolds in the nonparametric setting, the analysis of which is different from that in the Euclidean setting as the definition of the Mat\'ern processes on manifolds is subtle (though the rates are the same). It is also shown how the underlying geometric analysis outperforms by numerical experiments.

**Weaknesses:**

As for every (good) paper, there are always plenty of things remained to be done. For instance,

(1)  In the current analysis it is assumed that the "nugget" $\sigma_\epsilon$ is given. In Bayesian setting, it would be interesting to know what happens if we put a prior on it (and more importantly, what prior to put).

(2) The results, implicitly, compute the interpolation errors (as $L^2$ error based on $p_0$). What about extrapolation errors (that is, to predict an "outside" the domain)? This is also important in most geostatistics problems.

(3) For the numerical experiments, the authors only consider the synthetic examples, e.g. dragon, sphere. It is interesting to see if the intrinsic and extrinsic modeling can bring a big difference for some real data. One such example is provided in [14].

**Questions:**

While the paper is generally well written, I feel the authors need to address the following points:

(1) I catch the idea of the intrinsic vs extrinsic modeling quickly. But the authors may point out a reference for this terminology (is this inspired from [15])?

(2) p.3, line 80: I guess the authors mean $f \sim GP(m, k)$ instead of $GP(0,k)$.

(3) p.3, line 115-116: the notation $C^{\beta}$ for the H\"older class is nonstandard. Does this mean $\mathcal{C}^{0, \beta}$? I guess not since $\beta > \frac{d}{2}$ meaning that it can be larger than $1$. The authors may need to clarify this.

(4) In Result 1, it is assumed that $f$ is mean zero. Is this also assumed in Theorems 5 and 6? It is known that for the Mat\'ern processes in the Euclidean space, the mean function may raise the identifiability issue (see Stein's Interpolation of Spatial Data, or Tang, Zhang and Banerjee's paper On identifiability and consistency of the nugget in Gaussian spatial process models). There are also related discussions in [24]. The authors may need to clarify, and add a few sentences in the paper.

(5) p.4, Assumption 3: the authors may "indicate" $\sigma_\epsilon$ is known earlier, as this is important for the Bayesian workers.

(6) Regarding the use of Theorems 5 and 6: it is clear from the rates that one should take $\nu = \beta$ (i.e. if one can identify the function class, and set the same smoothness parameter in the Mat\'ern process). However, we never know exactly $\beta$. The question is whether there is an adaptive way to select $\nu$. (Of course, this may be discussed in another paper. I only want to bring this question to the authors.)

(7) Regarding Theorem 8: the rate is $\frac{2 \min(\beta , \nu)}{2 \nu +d} = \frac{2 \nu}{2 \nu +d}$ if $\beta = \nu$. This rate also appears in the prediction based on BLUE (best linear unbiased predictor) on the context of MLE. See e.g. Tang, Zhang and Banerjee's paper On identifiability and consistency of the nugget in Gaussian spatial process models, JRSS-B, 2021, page 1055. The authors may want to mention this as well.

(8) There are a few more references that the authors may want to add.

(a) Stein's book Interpolation of Spatial Data is one of the main references on the Mat\'ern Gaussian processes.

(b) Tang, Zhang and Banerjee's paper On identifiability and consistency of the nugget in Gaussian spatial process models studies the Mat\'ern process (with nugget) in the Euclidean space, where the identifiability issue occurs as pointed out in (7). This is related to the setting of Theorem 8. Arafat, Porcu, Bevilacqua and Mateu's paper Equivalence and orthogonality of Gaussian measures on spheres, Journal of Multivariate Analysis, 2018 is also relevant.

(c) Regarding the prior on $\sigma^2_\epsilon$, one such model is the conjugate Bayesian linear model in Banerjee's paper Modeling massive spatial datasets using a conjugate bayesian linear modeling framework, Spatial Statistics, 2020; this model was analyzed in Zhang, Tang and Banerjee's Exact Bayesian geostatistics using predictive stacking, arXiv:2304.12414, Section 2. I believe it should be possible to generalize the results in this paper to the conjugate bayesian linear model.

---

> ### Author Rebuttal · Authors · 2023-08-08
>
> Thank you very much for your thorough review of our work and for the very encouraging comments! Below we address key questions:
>
> **Further work:**
>
> (1) *Nugget and prior on $\sigma_\epsilon$*
> * Thank you for this question! In our work, the main reason we assume $\sigma_\epsilon$ is fixed is because in the Euclidean case other work has studied conditions on priors that ensure minimax optimality - at least up to ln n factor, as is done in [17] - **we expect these to be similar in both the Euclidean and geometric settings**. We therefore opted to not do this to avoid the paper becoming too long, but will amend the manuscript to add references so that readers interested in this case can find the relevant papers.
>
> (2) *Interpolation vs. extrapolation, $p_0$-norm vs empirical norm*
> * Thank you for the comment! We would argue that **our results do, in an appropriate technical sense, extrapolate** outside of the design points, since $p_0$ is an absolutely continuous measure - as opposed to, for instance, to the empirical measure of the data. Our proof technique uses the assumption that $p_0$ is lower-bounded, which implies that we can control convergence over the whole manifold, by moving between different distributions by changing the constants in the bound. Note that, if one considers regions where $p_0$ takes small values, the constant may degenerate, so this assumption does have limitations - but, we expect these properties are similar to the Euclidean case and not particularly specific to the manifold setting.
>
> (3) *Real data examples*
> * Thanks - this is a great point! We opted to focus on synthetic examples for simplicity, since this allows us to better control the moving parts in the experiment. However, we also wanted to note that in addition to [14] a similar performance difference was also observed in [12] in the context of medical data with a slightly-different prior. We will add a few remarks mentioning this to the experimental section to point reviewers toward these papers.
>
> **Questions:**
>
> (1) *Intrinsic vs. extrinsic terminology*
> * Thanks for this question! We called the processes this because the concepts “extrinsic” and “intrinsic” mirror the distinction between intrinsic and extrinsic properties/quantities **in the sense of differential geometry** (and mathematics more generally) where the former refers to concepts not needing any kind of embedding to be defined, while the latter refers to ones that need to be expressed through an embedding in a higher dimensional space/object. This is exactly the difference in how the two Matérn processes are constructed.
>
> (2) *Thanks for spotting the typo!*
>
> (3) *Notation for Hölder spaces*
> * This is a great point - thank you for spotting this! For $\gamma = k + \alpha$ with integer $k >= 0$ and $0 < \alpha <= 1$, define $CH^\gamma$ to be the space of k times differentiable functions with $k$th derivative being $\alpha$-Hölder. We've **changed the notation to $CH$** to avoid confusion with ordinary smooth functions.
>
> (4) *Prior mean and identifiability*
> * Thank you for this observation! The mean of the prior processes is indeed kept fixed at 0, as is commonly done for proving contraction rates for Gaussian processes. This choice actually leads to optimal contraction rates, but could more generally be relaxed. Regarding identifiability, note that we are not trying to identify covariance parameters given a sample of a Matérn process in an infill asymptotic regime, but rather to show contraction of the posterior towards a fixed regression function. **Our nonparametric regression model's parameter is therefore identifiable**: the probability distribution is only indexed by the regression function, and our results imply the existence of a consistent estimator for it, by taking for instance the posterior mean of the process. We will add a few sentences and references on this particular point.
>
> (5) *Emphasis on $\sigma_\epsilon$*
> * This is a good idea - we'll add a sentence on this!
>
> (6) *Adaptive selection of $\nu$*
> * This is an *excellent* point! We agree adaptivity to the smoothness of f_0 is a natural next step. For the intrinsic Matérn process adaptivity can be achieved by standard techniques that are not specific to the geometric setting. One way would be to follow the approach of Kirichenko & Van Zanten in “Estimating a smooth function on a large graph by Bayesian Laplacian regularization” where a multiplicative scaling parameter is introduced in the definition, and on which another prior is placed. Another solution could be to consider a truncated intrinsic Matérn process with a prior on the truncation, similar to Waaij & Van Zanten in “Full adaptation to smoothness using randomly truncated series priors with Gaussian coefficients and inverse gamma scaling”. For the extrinsic Matérn process, the problem is more complicated, but is tackled for the RBF kernel by Yang & Dunson in “Bayesian Manifold Regression” where they show that they can achieve adaptivity by placing a prior on the length scale of the prior process: **we believe a result of the same flavor could be shown in our context, although our proof technique is fundamentally different**. Studying adaptivity is, however, difficult and we believe should be the focus of a follow-up paper, but we are happy to add more comments on this in the revised version.
>
> (7) *Connection with other asymptotic rates*
> * For $\beta = \nu$, namely the less-realistic case where the right smoothness has been chosen in advance, we indeed recover the same contraction rate as the one you mention. This makes sense, as this is the minimax optimal rate of estimation of $f_0$ in this model. **It's a great idea to point this out with appropriate references** - thanks for bringing this to our attention!
>
> (8) *References*
> * Thank you very much for bringing these references to our attention! We will add them to help the reader frame our results with respect to identifiability and current research on Bayesian models.

---

> > ### Comment · Reviewer_woEb · 2023-08-11
> >
> > Thanks for the detailed explanations. The score remains unchanged.

---

### Official Review · Reviewer_qZfm · 2023-07-31

**Soundness:** 4 excellent
**Presentation:** 4 excellent
**Contribution:** 3 good
**Rating:** 7
**Confidence:** 3

**Summary:**

This paper establishes bounds on the contraction rate of matern processes on Riemannian manifolds. The authors study three variants:
1) intrinsic matern process
2) truncated intrinsic matern process
3) extrinsic matern process
and show that in each case, the optimal contraction rate can be achieved, which matches the Euclidean case.



**Strengths:**

This is a fundamental problem, and it is remarkable that the authors are able to prove the same optimal contraction rates for both the intrinsic and extrinsic Riemannian matern processes. Furthermore, the manifold hypothesis has been receiving increasing attention recently, and so the analysis of Gaussian processes on manifolds is well motivated.

I also appreciate the examples that the authors provided for illustrating the difference between intrinsic and extrinsic processes.

**Weaknesses:**

see questions below

**Questions:**

Regarding the smoothness parameter nu:
1) Can the authors provide intuition for why nu (as used in (7) and (10)) is the smoothness parameter (i.e. smoothness in what sense)?
2) The Riemannian matern process (7) and the Euclidean matern process (10) appear quite different, can the authors explain the analogy between these two processes, and specifically, why is nu, as used in (7), comparable to nu as used in (10)? In particular, can the authors explain the comment from line 275 "the intrinsic Matérn process, its truncated version and the extrinsic Matérn process all possess the same posterior contraction rates", and how these two rates are comparable under the different assumptions?
3) All theorems assume nu > d/2. Is this a standard assumption? How necessary is this assumption, and the theorems not work when nu is small? (and I have the a similar question for the beta > d/2 assumption in assumption 3 as well)

Regarding the intrinsic matern process:
4) expression (7) involves an sum over eigenfunctions of the laplace beltrami operator. The authors do mention that this sum can be truncated, but for a general manifold, it seems like even computing a single eigenfunction can be quite expensive. Can the authors comment on how this is done computationally (and what is the cost), when the manifold is an arbitrary one, e.g. the dragon?

5) In figure 2, the authors give a dumbbell example which highlights the difference between intrinsic and extrinsic kernels -- one important difference seems to be that two points can be far away in manifold distance, but close in euclidean distance (under the embedding). Consequently, a function may have a small lipschitz constant wrt manifold distance, but a huge lipschitz constant wrt euclidean embedding distance. Intuitively, why is this not reflected in the contraction rates? Is it because the assumptions made do not care about things like lipschitz smoothness? (related to my earlier question of what is the meaning of nu?)


Regarding the bound:

6) What is contained in the constant C in the theorems, is this a universal constant? Or does it depend polynomially/exponentially on any problem parameters?

---

> ### Author Rebuttal · Authors · 2023-08-08
>
> Thank you very much for taking the time to review our work! Thank you for the encouraging comment that work addresses a fundamental research problem! Below we address the questions, some of which we had to partially quote due to character limits:
>
> ---
>
> *".. intuition for why nu is the smoothness parameter .."*
> * Thank you for the question! Indeed, it may not appear obvious that $\nu$ in the definitions of the priors can be interpreted as a smoothness parameter. This stems from the sample path regularity of the processes: it is shown in [44] that the samples of the Euclidean Matérn process (and therefore also those of the extrinsic process) are $\alpha$-Holder for every $\alpha<\nu$, whereas this property for the intrinsic Matern process is **precisely the content of our Lemma 27**.
>
> *".. explain the analogy between these two processes  (Riemannian vs. Euclidean) ..", ".. why is nu, as used in (7), comparable to nu as used in (10)"*
> * Thank you for bringing this important point of confusion, which is likely to be shared by other readers, to our attention. To compare the two processes, note that both can be represented as **(weak) solutions of the same stochastic partial differential equation**, defined over Euclidean space and the manifold, respectively. This is the viewpoint from which the intrinsic Matérn process was studied in [8] and other prior works, and is why both are called Matérn processes.
> * Regarding $\nu$: see the remark above, but also note that the Euclidean Matérn process on R^d has an RKHS norm equivalent to the Sobolev space $H^{nu+d/2}(R^d)$, and the same is true for the intrinsic Matern by our Lemma 23 where the Sobolev space $H^{nu+d/2}(M)$ is defined using the Bessel potential space formulation.
>
> *".. how these two (contraction) rates are comparable under the different assumptions?"*
> * Thank you for the comment! Let us clarify this: we fix a **single, common data generating process**, namely nonparametric regression with random design and a fixed unknown regression function $f_0$, and, for **three different Bayesian models**, we compare the asymptotic contraction/convergence rate of the posterior distribution towards the true regression function $f_0$. The rates all depend on the intrinsic dimensionality of the data $d$, the smoothness parameter of our prior processes $nu$, and the smoothness of the true regression function $f_0$, and for the models considered end up equal up to constants. To improve clarity, we will update the manuscript to further emphasize this.
>
> *".. nu > d/2. Is this a standard assumption? .. *
> * This is a good question! **Yes, this is relatively standard:** $\nu>d/2$ is also used in the prior work [44], where the Euclidean counterparts of our results are proved. From a technical standpoint, this is needed in order to get a convergence rate under the $L^2(p_0)$ norm from a convergence rate at the input locations. Removing this assumption would be an interesting research question, but we suspect that it does not particularly involve geometry specific to the manifold setting which is our focus here.
>
> *".. comment on how this (obtaining Laplace-Beltrami eigenfunctions) is done computationally .."*
> * Thank you for the comment - this is actually the main computational challenge in working with kernels on manifolds. In practice one can rely on two different sets of techniques. The first one is to **discretize/mesh the manifold and solve for the eigenpairs of a large sparse matrix** as is done in [8], although this inevitably introduces numerical errors on top of the asymptotic contraction rates that we present here. The second way is to rely on **algebraic techniques based on symmetries which makes an exact computation possible for a large class of manifolds** of interest - see [2,3].
>
> *".. a function may have a small lipschitz constant wrt manifold distance, but a huge lipschitz constant wrt euclidean embedding distance. Intuitively, why is this not reflected in the contraction rates? .."*
> * This is an *extremely relevant point*: curvature and more general distortions between geodesic and Euclidean distances introduce undesirable behaviors of the extrinsic processes when compared to the intrinsic ones. In our analysis we are however incorporating the **same smoothness assumptions** on the regression function in both the intrinsic and the extrinsic case and nonetheless find **equivalent contraction rates** for both priors: the crucial fact is that the contraction rates presented here are all asymptotic. Hence, we highly suspect a **very bad dependence of the constant on the curvature and embedding** in the extrinsic case which would explain the differences in performance that we can witness in the small data regime - in fact, we believe our work provides an **excellent motivation to initiate future work on non-asymptotic contraction analysis** that we believe may be needed to capture these differences.
>
> *".. constant C in the theorems, is this a universal constant? Or does it depend polynomially/exponentially on any problem parameters?"*
> * Thank you for pointing that out! As discussed above, the value of $C$ is precisely what makes the extrinsic and intrinsic processes non-equivalent in practice when it comes to the performances. An inspection of the proof shows that $C$ depends on (1) $d,D$ (for the extrinsic prior), (2) the prior process hyperparameters, (3) $\beta$, (4) $M$, (5) the distribtuion $p_0$ over $x$-values, (6) $\sigma_\epsilon^2$, and (7) the Sobolev/Holder norms of the true regression function $f_0$. As mentioned in line 295 the constant in the case of the extrinsic process is expected to be bigger than the one for the intrinsic process, especially when the distortion between geodesic and Euclidean distance is high. We suspect in particular that the operator norm of the trace and extension operators between Sobolev spaces to be big in this case. We will add further clarifications on the constant in the revised version.

---

> > ### Comment · Reviewer_qZfm · 2023-08-13
> >
> > Thank you for the rebuttal. My questions are adequately addressed, and I have increased my score to 7.

---

### Official Review · Reviewer_yf8T · 2023-08-01

**Soundness:** 3 good
**Presentation:** 3 good
**Contribution:** 3 good
**Rating:** 8
**Confidence:** 3

**Summary:**

This paper investigates the theoretical properties and performance of Gaussian processes in machine learning, particularly when applied in geometric settings such as Riemannian manifolds. It compares intrinsic and extrinsic methods, with the former directly formulated on the manifold of interest and the latter requiring a higher-dimensional Euclidean space embedding. The research derives posterior contraction rates for three primary geometric model classes and shows that all three can lead to optimal procedures, given certain conditions. Empirical experiments support these theoretical findings, demonstrating better performance by intrinsic models in small-data regimes.

**Strengths:**

NA

**Weaknesses:**

NA

**Questions:**

NA

---

> ### Author Rebuttal · Authors · 2023-08-08
>
> Thank you very much for the accurate summary of our work and for your review! If you have any further questions or comments, please feel free to post them - we are happy to provide further information or clarification where needed. If not, thank you very much for your time in reading our work!

---

### Author Rebuttal · Authors · 2023-08-08

We would like to thank the referees for their summaries and pertinent observations that will help us to improve the final version of the paper. Most of the comments were about the format, additional references, identifiability and insights regarding the definitions of the different processes; we have responded to each reviewer in detail.

---

### Decision · Program_Chairs · 2023-09-21

**Decision:**

Accept (spotlight)

**Comment:**

The authors analyze Gaussian processes on Riemannian manifolds. This question is of significant interest in many areas of machine learning. They bound the posterior contraction rate (i.e. rate at which posterior converges to true data distribution) of intrinsic, truncated intrinsic, and extrinsic matern processes. (Interestingly the optimal rate is the same for intrinsic and extrinsic processes)

The reviewers all agree that the technical result is novel and interesting. Empirical results complement their results nicely, highlighting potential future directions for sharper bounds which will reveal the reveal the difference between intrinsic and extrinsic methods in certain settings.